# Genetic variants associated with platelet count are predictive of human disease and physiological markers

Evgenia Mikaelsdottir [1✉], Gudmar Thorleifsson[1], Lilja Stefansdottir[1], Gisli Halldorsson [1], Jon K. Sigurdsson[1], Sigrun H. Lund [1], Vinicius Tragante [1], Pall Melsted[1,2], Solvi Rognvaldsson [1], Kristjan Norland[1], Anna Helgadottir [1], Magnus K. Magnusson [1,3], Gunnar B. Ragnarsson[4], Sigurdur Y. Kristinsson[3,5], Sigrun Reykdal[5], Brynjar Vidarsson[5], Ingibjorg J. Gudmundsdottir [6], Isleifur Olafsson[7], Pall T. Onundarson[3,8], Olof Sigurdardottir[9], Emil L. Sigurdsson[10], Gerdur Grondal[11], Arni J. Geirsson[11], Gudmundur Geirsson[12], Julius Gudmundsson[1], Hilma Holm [1], Saedis Saevarsdottir [1,3,11], Ingileif Jonsdottir [1,3], Gudmundur Thorgeirsson[1,6], Daniel F. Gudbjartsson [1,2], Unnur Thorsteinsdottir[1,3], Thorunn Rafnar [1] & Kari Stefansson [1,3✉]

Platelets play an important role in hemostasis and other aspects of vascular biology. We conducted a meta-analysis of platelet count GWAS using data on 536,974 Europeans and identified 577 independent associations. To search for mechanisms through which these variants affect platelets, we applied *cis*-expression quantitative trait locus, DEPICT and IPA analyses and assessed genetic sharing between platelet count and various traits using polygenic risk scoring. We found genetic sharing between platelet count and counts of other blood cells (except red blood cells), in addition to several other quantitative traits, including markers of cardiovascular, liver and kidney functions, height, and weight. Platelet count polygenic risk score was predictive of myeloproliferative neoplasms, rheumatoid arthritis, ankylosing spondylitis, hypertension, and benign prostate hyperplasia. Taken together, these results advance understanding of diverse aspects of platelet biology and how they affect biological processes in health and disease.

[1] deCODE Genetics/Amgen, Sturlugata 8, 101 Reykjavik, Iceland. [2] School of Engineering and Natural Sciences, University of Iceland, Reykjavik, Iceland. [3] Faculty of Medicine, University of Iceland, 101 Reykjavik, Iceland. [4] Department of Oncology, Landspitali-University Hospital, 101 Reykjavik, Iceland. [5] Department of Hematology, Landspitali-University Hospital, 101 Reykjavik, Iceland. [6] Department of Cardiology, Landspitali-University Hospital, 101 Reykjavik, Iceland. [7] Department of Clinical Biochemistry, Landspitali-University Hospital, 101 Reykjavik, Iceland. [8] Laboratory Hematology, Landspitali-University Hospital, 101 Reykjavik, Iceland. [9] Department of Clinical Biochemistry, Akureyri Hospital, 600 Akureyri, Iceland. [10] Development Centre for the Primary Care, 109 Reykjavik, Iceland. [11] Department of Rheumatology, Landspitali-University Hospital, 101 Reykjavik, Iceland. [12] Department of Urology, Landspitali-University Hospital, 101 Reykjavik, Iceland. ✉email: evgenia.mikaelsdottir@gmail.com; kstefans@decode.is

Platelets are small anucleate cell fragments generated from megakaryocytes in the bone marrow in the process of thrombopoiesis and released through proplatelets into the bloodstream, where they play a major role in primary hemostasis and numerous other aspects of vascular biology and homeostasis[1]. Platelet count (PLT) in humans ranges between 150,000 and 400,000/μl and is regulated through a balance between thrombopoiesis and platelet turnover[2,3]. PLT deviating from the normal range, either a decrease (thrombocytopenia) or an increase (thrombocythemia), may be indicative of a disease state[2].

PLT is highly heritable, with reported heritability estimates ranging from 54 to 87%, suggesting an essential role of genetics in shaping this trait[2]. Several genes affecting PLT have been identified through studies of congenital platelet-related disorders[4–6]. Further discoveries have been made through genome-wide association studies (GWAS)[7–15], yielding 1,810 variants that associate with PLT with $P$ value $\leq 5 \times 10^{-8}$ (GWAS catalog[16], last access May 26, 2021).

To explore PLT and its connections to diseases and other traits, we conducted a meta-GWAS analysis of PLT using data from 536,974 Europeans from Iceland and the UK and identified 577 independent signals associated with PLT. In an attempt to determine how these variants affect PLT, we performed a search for candidate causal genes and pathways, employing coding variant and cis-expression quantitative trait locus (cis-eQTL) analyses. In addition, we analyzed our data with Data-driven Expression Prioritized Integration for Complex Traits (DEPICT)[17] and employed the Ingenuity Pathway Analysis (IPA) (QIAGEN Inc., https://www.qiagenbioinformatics.com/products/ingenuitypathway-analysis)[18]. We also assessed genetic sharing by PLT and various diseases and other traits using a PLT polygenic risk score.

## Results

**Variants associating with PLT.** We conducted meta-analysis of PLT GWAS from Iceland and the UK ("Methods"). In the Icelandic study, we analyzed about 32.6 million variants identified through whole-genome sequencing of 28,075 Icelanders and imputed into 139,479 chip-typed individuals and their untyped 1st and 2nd degree relatives[19], or a total of 270,211 individuals with PLT measurements ("Methods"). In the UK Biobank GWAS, we used the haplotype reference panel for imputation and analyzed about 33 million variants. The UK Biobank study contained 397,495 individuals of European British ancestry with PLT measurements. For genome-wide significance thresholds, we used the weighted Holm–Bonferroni method[20] to account for all 50,177,681 variants being tested in the combined dataset and defined $P$ value thresholds that take into account the prior probability of functional impact of the variants[21] ("Methods" and Supplementary Data 1).

Under the additive model, the meta-analysis and subsequent conditional analyses yielded 577 independent genome-wide significant signals (Supplementary Fig. 1 and Supplementary Data 2). Apart from rs190391173, we did not observe substantial heterogeneity in the effect estimates between the two populations (Supplementary Fig. 2 and Supplementary Data 2). While the majority of the variants are common (minor allele frequency (MAF) ≥ 5%), 28 are rare (MAF ≤ 1%) in at least one of the populations (Supplementary Data 2). Forty-four of the 577 variants are predicted to affect the coding sequence of the corresponding genes, with 40 annotated as missense or splice-region variants and 4 annotated as loss-of-function variants (stop-gained, splice donor, or splice acceptor) (Supplementary Data 2).

Three of the signals are located ≥1 Mb away from the nearest known PLT variant reported in the GWAS catalog[16] as of May 26, 2021 (Note § in Supplementary Data 2), indicating novel PLT loci. rs77542162 on chr17 is a missense variant in *ABCA6*. The variant has been reported to associate with blood lipids[22,23], and some other variants in the area have been associated with height. However, there are no previously reported associations with platelet traits in the 2-MB window surrounding the SNP. The gene belongs to the ATP-binding cassette transporter family, it is cholesterol-responsive and potentially involved in intracellular lipid transport processes[24,25]. While the exact mechanisms of ABCA6 involvement in the control of these traits are yet to be uncovered, there is an example of a related transporter, ABCG4, that regulates cholesterol efflux and platelet production/number[26]. rs2118446 and rs7808461 are intergenic variants on chr2 and chr7, respectively. Although variants nearby have been reported in association with mean platelet volume (MPV), the only trait besides PLT that these two variants associate with in our study, there have been no previous reports of association with PLT at these loci. It should also be noted that while PLT and MPV are genetically related, association with one does not automatically result in association with the other. Moreover, variants, located near rs7808461 and reported to associate with MPV[12,14,15], correlate with rs7808461 with $R^2 < 0.2$, thus very likely representing a different signal. For rs2118446-T, we found one cis-eQTL that involves gene expression of *GCC2* in adipose tissue (effect = −0.95, $P$ value = $2.1 \times 10^{-80}$ in the Icelandic RNA-sequencing data generated from the adipose tissue, see "Methods"). GCC2 (GRIP and coiled-coil domain containing 2) is a peripheral membrane protein localized to the *trans*-Golgi network and is required for endosome-to-Golgi transport and maintenance of Golgi structure[27,28]. According to the expression data[29,30], the gene is also expressed in platelets and megakaryocytes, but no significant cis-eQTLs involving rs2118446 and *GCC2* were detected in these cell types, and the exact role of the gene in platelets and/or megakaryocytes is not known. We did not find any message in the proximity to rs7808461 that was affected by the SNP in the tissues analyzed.

**PLT polygenic risk score.** Platelet indices have been reported to associate with various diseases and quantitative traits (QTs)[31–35]. In order to search for diseases and other traits that share a genetic basis with PLT, we derived a polygenic risk score for both Icelandic and UK PLT datasets (PLT PRS) using about 600,000 variants[36] and subsequently analyzed the score for association with all available diseases and other traits across the two populations. We set meta $P$ value $\leq 1 \times 10^{-5}$ (0.05/5,000 available main phenotypes) as a significance threshold. This analysis yielded 24 traits associated with the PLT PRS (Table 1 and Supplementary Data 3), including five diseases: myeloproliferative neoplasms (MNP), ankylosing spondylitis (AS), rheumatoid arthritis (RA), hypertension, and benign prostate hyperplasia (Table 1). Interestingly, the PLT PRS associated only with MPN among hematologic diseases (Supplementary Data 4).

Of the QTs tested, the PLT PRS associated most significantly with MPV and all tested blood cell counts, except red blood cell count. In addition, the PLT PRS associated with several QTs related to cardiovascular health, anthropometric traits and with QTs linked to inflammation, and kidney and liver functions (Table 1).

To analyze the effect of the MHC region on the association of the PLT PRS with these traits, we removed the variants, representing the MHC in the polygenic risk core calculations, and recalculated the association of the PLT PRS with these traits (Supplementary Data 5). Exclusion of the MHC revealed that the

**Table 1 Platelet count polygenic risk score and correlation with human diseases and quantitative traits.**

| Trait | Iceland | | | UK | | | Combined | | |
|---|---|---|---|---|---|---|---|---|---|
| | E | P | nAff/nCon | E | P | nAff/nCon | E (95% CI) | P | $P_{het}$ |
| AS | 0.179 | 0.002 | 373/139,426 | 0.322 | $5.2 \times 10^{-15}$ | 612/407,963 | 0.274 (0.208, 0.340) | $3.2 \times 10^{-16}$ | 0.044 |
| Hypertension | 0.021 | 0.017 | 32,026/110,632 | 0.020 | $4.4 \times 10^{-6}$ | 77,566/331,087 | 0.020 (0.013, 0.028) | $2.3 \times 10^{-7}$ | 0.92 |
| RA | 0.129 | $4.3 \times 10^{-7}$ | 1772/136,368 | 0.098 | $5.7 \times 10^{-10}$ | 4001/404,652 | 0.107 (0.080, 0.133) | $2.1 \times 10^{-15}$ | 0.32 |
| MPN | 0.237 | 0.0017 | 180/141,889 | 0.209 | 0.00011 | 348/406,981 | 0.218 (0.132, 0.305) | $6.6 \times 10^{-7}$ | 0.76 |
| BPH | 0.018 | 0.2 | 9346/42,591 | 0.041 | $5.6 \times 10^{-8}$ | 21,067/166,609 | 0.036 (0.023, 0.049) | $7.0 \times 10^{-8}$ | 0.15 |
| **Trait** | **Beta** | **P** | **n** | **Beta** | **P** | **n** | **Beta (95% CI)** | **P** | **$P_{het}$** |
| PLT | 0.228 | $2.9 \times 10^{-1558}$ | 139,489 | 0.326 | $3.8 \times 10^{-6978}$ | 397,495 | 0.295 (0.292, 0.298) | $3.9 \times 10^{-8334}$ | $5.1 \times 10^{-531}$ |
| MPV | −0.201 | $1.2 \times 10^{-675}$ | 136,560 | −0.192 | $1.2 \times 10^{-1865}$ | 397,490 | −0.194 (−0.198, −0.190) | $2.4 \times 10^{-2538}$ | $1.9 \times 10^{-5}$ |
| WBC | 0.059 | $7.6 \times 10^{-116}$ | 140,123 | 0.082 | $1.2 \times 10^{-463}$ | 397,495 | 0.075 (0.072, 0.078) | $2.3 \times 10^{-566}$ | $1.0 \times 10^{-31}$ |
| Lymphocyte count | 0.047 | $3.4 \times 10^{-73}$ | 132,215 | 0.066 | $3.3 \times 10^{-300}$ | 396,820 | 0.060 (0.057, 0.063) | $1.3 \times 10^{-363}$ | $1.6 \times 10^{-9}$ |
| Basophil count | 0.027 | $4.2 \times 10^{-64}$ | 131,881 | 0.026 | $3.2 \times 10^{-60}$ | 396,820 | 0.026 (0.024, 0.029) | $2.2 \times 10^{-122}$ | 0.66 |
| Eosinophil count | 0.043 | $6.6 \times 10^{-61}$ | 131,889 | 0.061 | $7.2 \times 10^{-251}$ | 396,820 | 0.055 (0.052, 0.058) | $8.7 \times 10^{-303}$ | $1.4 \times 10^{-8}$ |
| Monocyte count | 0.055 | $2.2 \times 10^{-87}$ | 132,212 | 0.063 | $5.9 \times 10^{-243}$ | 396,820 | 0.060 (0.057, 0.064) | $4.4 \times 10^{-327}$ | 0.00012 |
| Neutrophil count | 0.042 | $8.2 \times 10^{-66}$ | 132,200 | 0.062 | $2.9 \times 10^{-270}$ | 396,820 | 0.055 (0.052, 0.058) | $1.4 \times 10^{-324}$ | $3.9 \times 10^{-25}$ |
| Triglycerides | 0.025 | $9.7 \times 10^{-9}$ | 96,081 | 0.020 | $2.4 \times 10^{-29}$ | 390,346 | 0.021 (0.017, 0.024) | $2.7 \times 10^{-36}$ | 0.29 |
| Total cholesterol | 0.023 | $2.2 \times 10^{-7}$ | 102,393 | 0.012 | $2.1 \times 10^{-9}$ | 390,652 | 0.014 (0.010, 0.017) | $3.2 \times 10^{-14}$ | 0.024 |
| Non-HDL cholesterol | 0.021 | $4.2 \times 10^{-6}$ | 96,881 | 0.013 | $3.5 \times 10^{-9}$ | 358,461 | 0.015 (0.011, 0.018) | $2.5 \times 10^{-13}$ | 0.11 |
| Heart rate | 0.022 | $2.1 \times 10^{-6}$ | 55,042 | 0.015 | $1.9 \times 10^{-17}$ | 385,917 | 0.016 (0.013, 0.019) | $5.9 \times 10^{-22}$ | 0.16 |
| Mean arterial pressure | 0.009 | 0.041 | 76,003 | 0.013 | $1.2 \times 10^{-12}$ | 385,914 | 0.012 (0.009, 0.016) | $2.0 \times 10^{-13}$ | 0.4 |
| GGTP | 0.031 | $2.1 \times 10^{-14}$ | 101,537 | 0.031 | $7.4 \times 10^{-60}$ | 390,457 | 0.031 (0.028, 0.034) | $1.4 \times 10^{-72}$ | 1 |
| AP | 0.020 | $2.8 \times 10^{-6}$ | 102,035 | 0.018 | $2.3 \times 10^{-21}$ | 390,671 | 0.018 (0.015, 0.022) | $3.9 \times 10^{-26}$ | 0.67 |
| Bilirubin | −0.029 | $3.1 \times 10^{-13}$ | 73,535 | −0.013 | $4.5 \times 10^{-17}$ | 389,085 | −0.015 (−0.018, −0.012) | $1.2 \times 10^{-25}$ | 0.00018 |
| CRP | 0.013 | $5.7 \times 10^{-5}$ | 115,557 | 0.019 | $2.2 \times 10^{-24}$ | 389,840 | 0.018 (0.014, 0.021) | $2.3 \times 10^{-27}$ | 0.11 |
| Creatinine | −0.019 | $1.3 \times 10^{-6}$ | 136,328 | −0.014 | $4.8 \times 10^{-15}$ | 390,463 | −0.015 (−0.018, −0.012) | $6.6 \times 10^{-20}$ | 0.25 |
| Height | −0.055 | $6.6 \times 10^{-20}$ | 74,992 | −0.036 | $1.8 \times 10^{-67}$ | 407,823 | −0.038 (−0.042, −0.034) | $1.1 \times 10^{-83}$ | 0.0029 |
| Weight | −0.027 | $2.9 \times 10^{-8}$ | 76,546 | −0.020 | $1.9 \times 10^{-29}$ | 407,548 | −0.021 (−0.024, −0.018) | $8.9 \times 10^{-36}$ | 0.18 |
| RBC | −0.005 | 0.062 | 139,114 | 0.002 | 0.28 | 397,499 | 0.000 (−0.003, 0.003) | 0.85 | 0.032 |

*AP alkaline phosphatase, AS ankylosing spondylitis, BPH benign prostate hyperplasia, CRP C-reactive protein, E effect, GGTP gamma-glutamyl transpeptidase, MPV mean platelet volume, n number of individuals, nAff number of cases, nCon number of controls, P P value, P$_{het}$ P value for heterogeneity in the effect estimate between the Icelandic and UK Biobank data, PLT platelet count, RA rheumatoid arthritis, RBC red blood cell count, WBC white blood cell count. The applied significance threshold was a combined P value of $1 \times 10^{-5}$ (see "Methods"). Only statistically significant associations are presented.*

PLT PRS association with ankylosing spondylitis, benign prostate hyperplasia, and cholesterol is dependent on the MHC region. While association with rheumatoid arthritis is strongly affected by the MHC region, the MHC does not fully explain the association of the PLT PRS with rheumatoid arthritis. At the same time, exclusion of the MHC strengthened the PLT PRS association with hypertension, MPV, heart rate, alkaline phosphatase, and bilirubin (Supplementary Data 5).

**Association of the PLT variants with phenotypes identified in the PLT PRS analyses.** Next, we tested the 577 variants from the PLT meta-analysis for association with the phenotypes identified in the PLT PRS analyses, combining available trait data from Iceland and the UK. The significance threshold for the meta $P$ value was set at $3.6 \times 10^{-6}$, or 0.05 corrected with the number of variants (577) and phenotypes tested (24, excluding the discovery phenotype PLT). All significant associations were subjected to further testing by conditional analyses to determine whether the variant itself is responsible for the observed effect ("Methods").

Of the 577 variants, 356 associated with QTs other than PLT. As expected based on the results of the PLT PRS analyses, they associated most frequently with hematologic QTs, with MPV in the leading position (Fig. 1 and Supplementary Data 6). The direction of effects was most consistent with the PLT PRS association results for MPV, blood cell counts, and C-reactive protein (Table 1, Fig. 1, and Supplementary Data 6). Variants associating with decreased PLT associated with larger platelet size and vice versa, consistent with previously reported shared genetic influence on PLT and MPV and an inverse correlation between the two traits[37,38]. However, 17 out of 191 variants associated with PLT and MPV with the same direction of effects (Supplementary Data 6). This indicates that while the inverse

relationship between PLT and MPV is generally a rule, there are some exceptions[39]. Only 7 of 577 variants associated with the tested diseases with $P$ value $\leq 3.6 \times 10^{-6}$, with five variants associating with hypertension, one associating with RA, and one associating with MPN (Supplementary Data 7).

**Pathway and tissue enrichment analysis and search for causal genes.** Only 38 out of the 577 variants are located in or near genes implicated in platelet disorders (Supplementary Data 2 and 8), defined based on information from the Online Mendelian Inheritance in Man database (OMIM[40], www.omim.org) and confirmed through literature search. However, mechanisms by which the remaining variants affect PLT are unclear. In order to shed light on these mechanisms, we employed several different approaches.

We searched for candidate causal genes at loci harboring the identified variants by screening for correlated coding sequence variants, which could account for the signal. Forty-four of the 577 signals are represented by an index SNP that affects a protein sequence (missense, splice region, stop-gained, splice donor, or splice acceptor variants) (Supplementary Data 2). Additional 71 index SNPs were found to correlate strongly with coding variants and represent the same signal as confirmed by conditional analyses (Supplementary Data 9 and 10). In other words, coding sequence variants could be responsible for the association with PLT for 115 of the 577 variants. Of these 115 signals, 85 are located in genes expressed in megakaryocytes and/or platelets, based on expression data from platelets and megakaryocytes[29,30], with 18 of them mapping to genes implicated in platelet-related disorders (Supplementary Data 2, 8 and 9). The remaining 30 variants are located in 29 genes with no record of expression in these two cell types. However, several of them are noteworthy for their role in cholesterol/lipid homeostasis: *ABCA6*, *APOH*, *GCKR*,

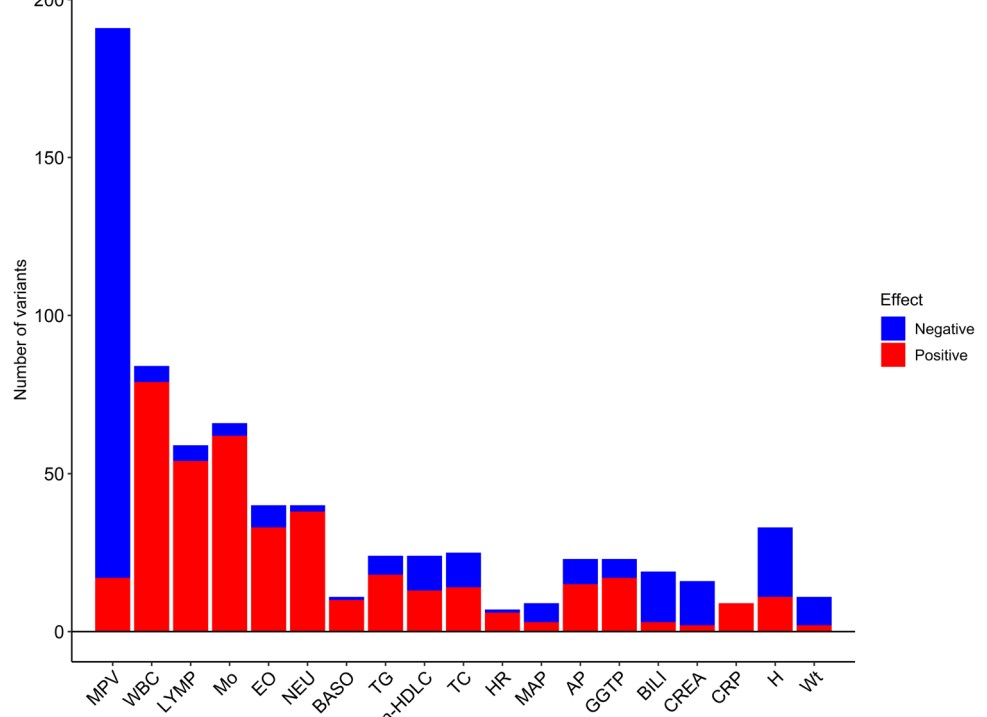

**Fig. 1 Overview of association of the PLT variants with other quantitative traits.** Data are presented with respect to the PLT increasing allele. Significance criteria: $P$ value $\leq 3.6 \times 10^{-6}$ ("Methods"). For details on associations, see Supplementary Data 6. AP alkaline phosphatase, BASO basophil count, BILI total bilirubin, CREA serum creatinine, CRP C-reactive protein, EO eosinophil count, GGTP gamma-glutamyl transpeptidase, H height, HR heart rate, LYMP lymphocyte count, MAP mean arterial pressure, Mo monocyte count, MPV mean platelet volume, n-HDLC non-HDL cholesterol, NEU neutrophil count, TC total cholesterol, TG triglycerides, WBC white blood cell count, Wt weight.

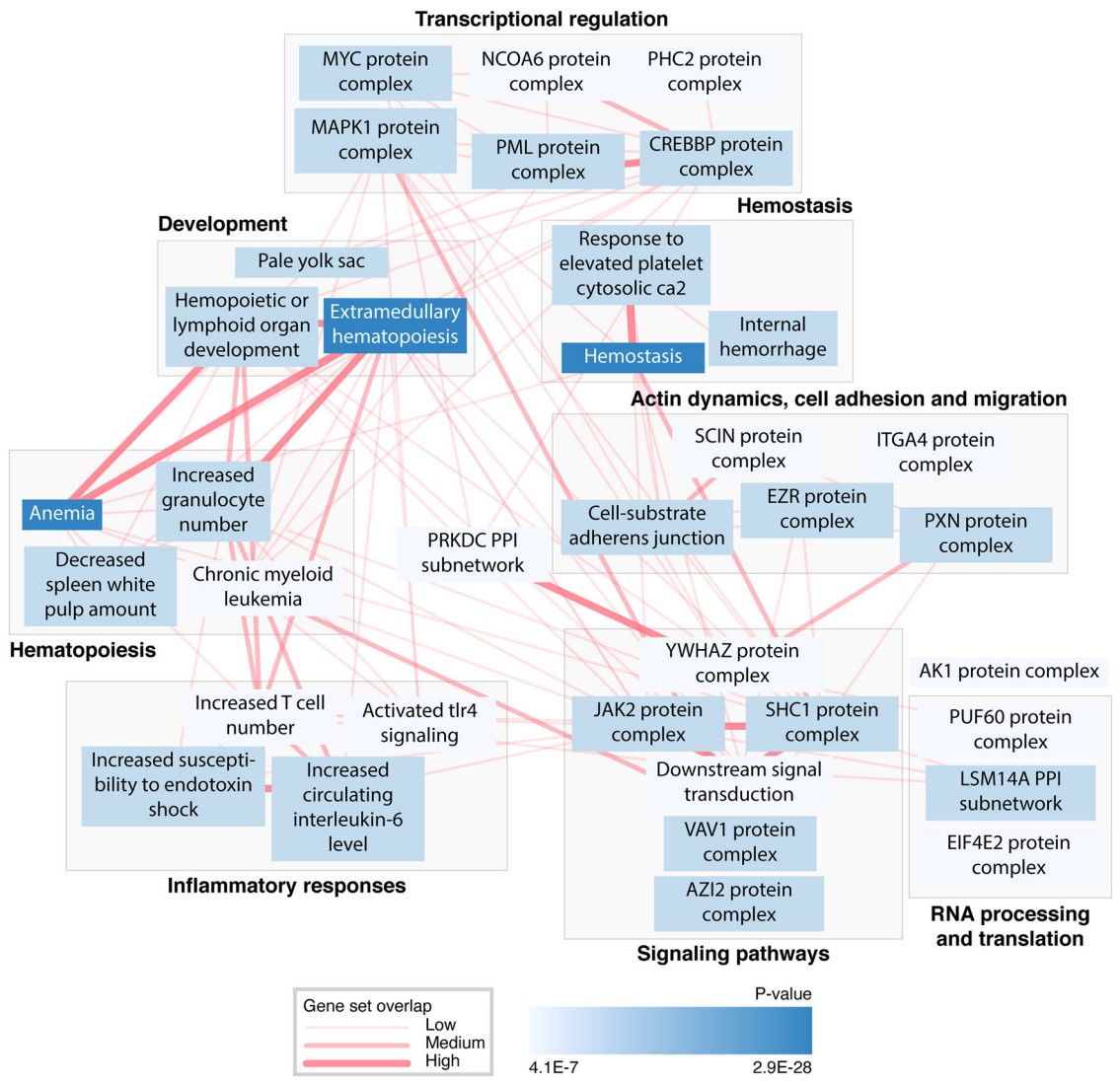

**Fig. 2 A network of gene sets identified in the DEPICT analyses.** Gene sets from the gene set enrichment analysis were clustered according to their most relevant biological functions. Only the topmost gene sets of the clusters with eight or more significant gene sets are presented (significance criteria: *P* value $\leq 3.46 \times 10^{-6}$). Connecting lines represent gene set overlap if Pearson correlation > 0.3, with thicker lines indicating higher correlation. For detailed information on gene sets, clusters, and Pearson correlation between the sets, see Supplementary Data 13 and 14.

*TM6SF2*, *IRF1*, and *PNPLA3* (Supplementary Data 11). By analogy with *ABCG4*[26], they could function both in platelet count regulation and control of cholesterol/lipid homeostasis. Alternatively, they might modify cholesterol/lipid homeostasis and thus create a pro-inflammatory environment, leading to an increase in platelet production. Interestingly, all these genes are expressed in the liver, with three being linked to nonalcoholic fatty liver disease (Supplementary Data 11). Further experimental evidence is required to elucidate their role in both regulation of PLT and control of lipid metabolism/homeostasis.

Physiological system, cell type, and tissue enrichment analysis with DEPICT[17] revealed the strongest enrichment for spleen (*P* value = $4.8 \times 10^{-14}$), followed by enrichment for blood cells and synovial fluid (*P* values ranging from $9.4 \times 10^{-14}$ to $3.4 \times 10^{-9}$) (Supplementary Data 12). In gene set enrichment analysis, we tested whether genes in associated regions were enriched for reconstituted versions of gene sets[17] and identified 758 significantly enriched gene sets grouped into 139 clusters (Supplementary Data 13, significance criteria set at *P* value $\leq 3.5 \times 10^{-6}$, or 0.05 corrected with the number of tested gene sets (14,461)). The analysis revealed that the most significant gene sets are the ones

implicated in hematopoiesis, inflammatory responses, signal transduction, transcriptional regulation, RNA processing and translation, hemostasis, development, actin dynamics, cell adhesion, and cell migration (Fig. 2, Supplementary Data 13 and 14).

To search for genes whose expression is affected by the 577 variants in cells and tissues that were identified as significantly enriched by DEPICT (Supplementary Data 12), we performed *cis*-eQTL analysis using the Icelandic whole blood and adipose tissue datasets, the Genotype-Tissue Expression (GTEx) project data[41], as well as recently reported platelet and megakaryocyte datasets[29] (see "Methods"). We found 579 significant *cis*-eQTLs, where 158 variants associated with altered expression of 235 genes in 19 tissues and cell types, all represented by the top eQTLs (Fig. 3 and Supplementary Data 15).

Taken together, these approaches yielded a list of 284 annotated genes that can be classified as candidate causal genes for PLT regulation (Supplementary Data 16). While some of these genes (e.g., *GP1BA*, *GP6*, *ITGA2B*, and *TUBB1*) are already well-known in platelet biology, mechanisms by which other genes may affect PLT are yet to be elucidated. The list of these genes was analyzed with IPA for association with diseases, physiological

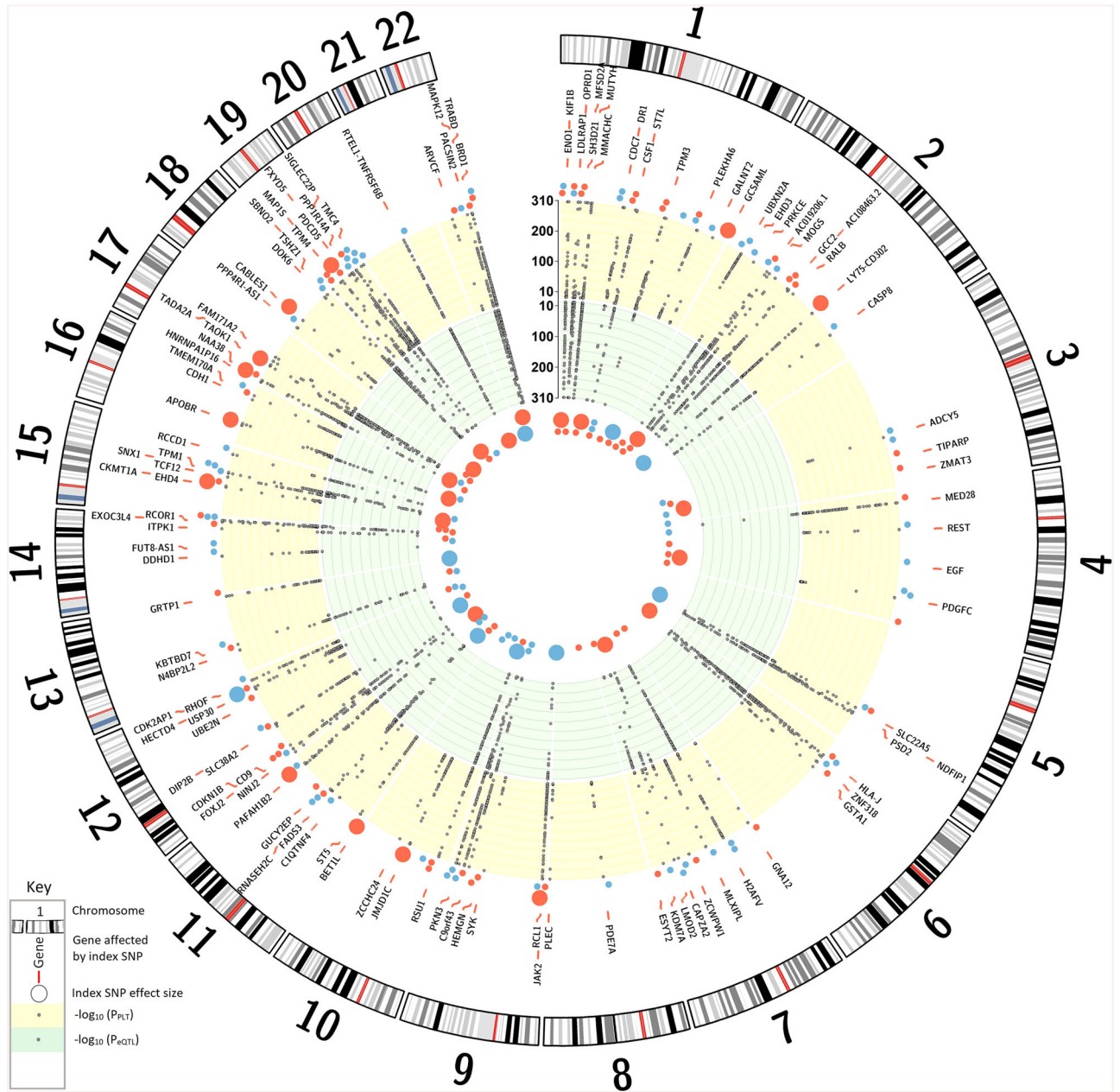

**Fig. 3 The PLT variants affecting gene expression.** The Circular Manhattan plot shows genes identified in the *cis*-eQTL analyses whose expression is affected by the PLT variants (Supplementary Data 15). Only PLT variants affecting the expression of these genes are presented in the plot (Supplementary Data 2, Note 1). For variants, which affect expression of more than one gene, only the gene representing the strongest *cis*-eQTL (the largest effect size) is shown. Effector alleles are the same as in Supplementary Data 2. The yellow band: $-\log_{10}$ of P value for association of the variants with PLT. Dots outside the yellow band represent the genetic effect sizes of the index PLT SNPs, with the blue dot color indicating PLT decrease and the red representing PLT increase (see Supplementary Data 2 for details). The green band: $-\log_{10}$ of P value for association of the variants with *cis*-eQTLs. eQTL effects are shown as colored dots inside the green band. The dot size represents the effect size, and the color indicates the effect direction, where red is increase and blue is a decrease of gene expression. For scaling purposes, both PLT and *cis*-eQTL effects are expressed in standard deviation.

systems, and cellular functions, revealing strong involvement of the listed genes in cancers and hematological diseases, including hereditary platelet disorders and myeloproliferative neoplasms, as well as platelet development, morphology, and function, hematological system development and function, and cellular functions related to cell-to-cell signaling, cell development, cell death, and proliferation (Fig. 4 and Supplementary Data 17).

**IPA core analysis of *cis*-eQTL data.** To gain deeper insight into possible mechanisms of PLT regulation by the sequence variants,

we performed a Core Analysis of our *cis*-eQTL data (Supplementary Data 15) with IPA[18], focusing on the identification of master regulators and changes in canonical pathways and toxicological functions. For consistency with the PLT PRS results, the effects in the *cis*-eQTL data were expressed in terms of the PLT increasing allele.

In the IPA Core Analysis, 279 endogenous molecules were identified as master regulators, i.e., molecules that can act on the genes in the dataset either directly or through intermediate regulators, with Benjamini–Hochberg corrected P value ≤ 0.05 (Supplementary Data 18). Fourteen of these master regulators

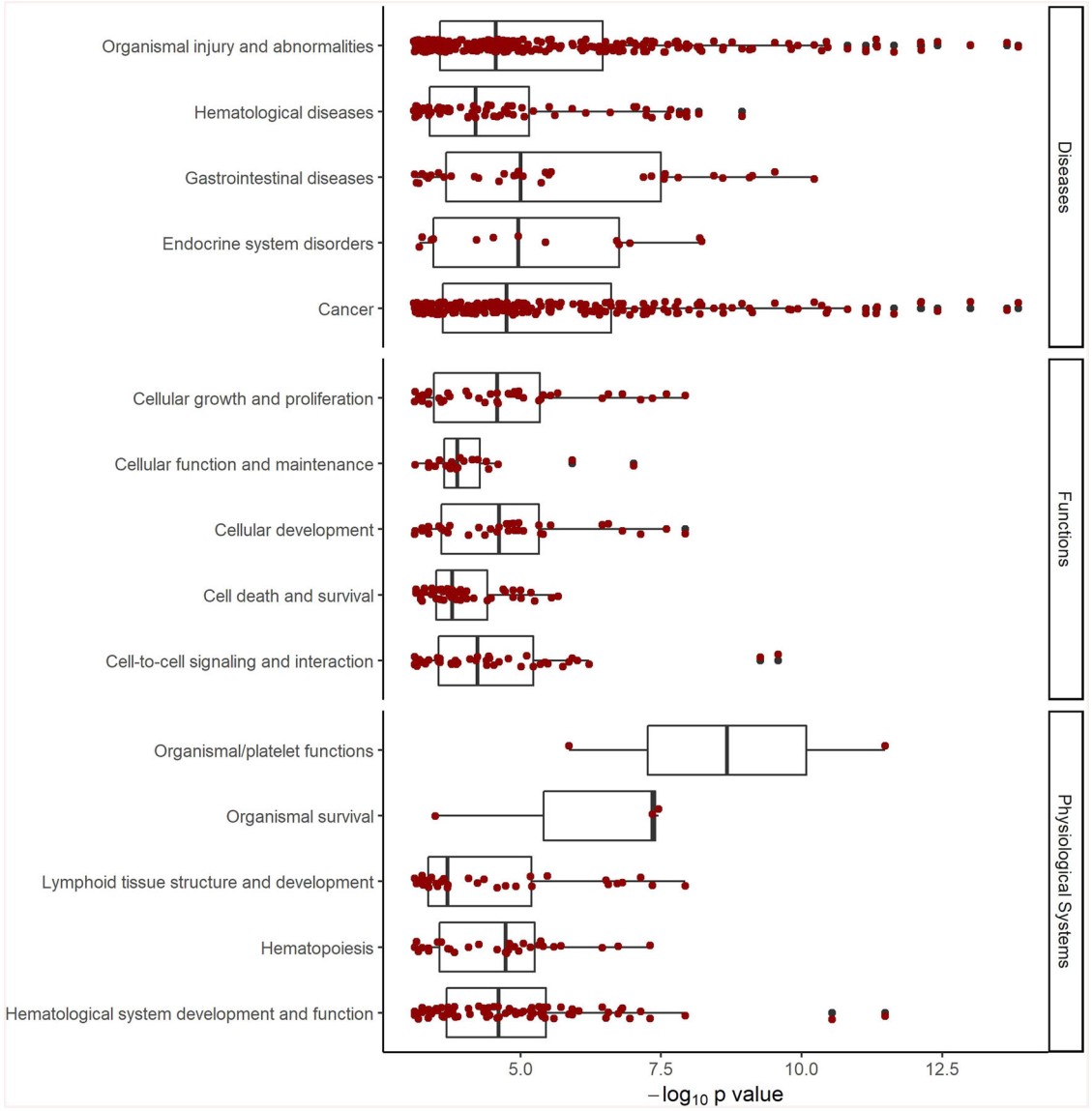

**Fig. 4 Association with diseases, molecular functions, and physiologic systems.** The 284 candidate causal PLT genes (Supplementary Data 16) were analyzed for association with diseases, molecular functions, and physiologic systems, using the Ingenuity Pathway Analysis (see "Methods"). Shown are *P* values of the identified associations along with median and interquartile ranges for each group (boxplots). Dark red dots represent individual associations, and black dots indicate outliers. For details on associations, see Supplementary Data 17.

associated with a significant increase or decrease in activity, that is, the absolute value of activation *Z* score ≥ 2 ("Methods"), and were investigated in more detail in terms of diseases, functions, and canonical pathways that they control, using data from the IPA's Ingenuity Knowledge Base (Supplementary Data 19). This analysis revealed that some of the top master regulators could be involved in regulatory mechanisms linked to the diseases and traits associated with the PLT PRS. Examples of such master regulators are FOXO3, which controls mechanisms linked to MPN, RA, hypercholesterolemia, and hypertriglyceridemia, and PRKCG that controls canonical pathways related to regulation of blood pressure, platelet count, and function and linked to RA (Table 1 and Supplementary Data 19). In addition, the IPA Core Analysis revealed that some aspects of kidney and liver functions (e.g., renal necrosis/cell death and liver hyperplasia/hyperproliferation) could be affected by the differential gene expression observed in our *cis*-eQTL data (Supplementary Data 20).

The IPA Core Analysis also identified eight significantly altered canonical pathways (Benjamini–Hochberg corrected *P* values

≤ 0.05, Supplementary Data 21). Some of these eight canonical pathways might play a role in regulation of molecular mechanisms relevant to the diseases associated with the PLT PRS, e.g., Ephrin Receptor Signaling and contraction of vascular tissue or Role of Tissue Factor in Cancer and myelopoiesis (Supplementary Data 22).

## Discussion
In this study, we identified 577 variants that associate with PLT in the Icelandic and UK populations. While most of the variants are found at loci already reported[7–15], three of these variants are located 1 MB away from variants reported to associate with PLT in previous GWAS, indicating novel PLT loci. The loci, harboring two of them, rs2118446 and rs7808461, have been reported to associate with MPV. However, it should be noted that although PLT and MPV are genetically related, association with one trait does not automatically mean association with the other one. For example, of 577 variants associated with PLT in our study only

33% (191 variants) associate also with MPV. The third one is a missense variant in *ABCA6* that has been associated with blood lipids, but not with PLT or other platelet-related traits. Only 38 of the 577 variants reside in or near genes that are implicated in platelet-related disorders. To search for potential causal genes for the remaining variants and link our findings with biological functions, we performed *cis*-eQTL and coding variant analyses, pathway/tissue and gene set enrichment analyses with DEPICT and Core Analyses with IPA. These analyses pointed to potentially causal genes, many of which do not currently have a known role in platelet biology.

Analysis of association between the PLT PRS and hematologic traits revealed interesting results. According to the classical view of hematopoiesis, the separation of the myeloid and lymphoid lineages is followed by a further split of the myeloid lineage into megakaryocyte–erythrocyte bipotent progenitors and progenitors of other myeloid lineages[42]. Given this close developmental relationship between platelets and red blood cells, a lack of association between the PLT PRS and red blood cell count is somewhat unexpected, albeit consistent with results of recent reports[14,43,44]. Of the specific blood cell types tested, the PLT PRS associated most significantly with monocyte, lymphocyte, and neutrophil counts. Moreover, of 356 variants associated with QTs other than PLT, 119 associated with one or more of these cell-type counts. However, despite this very strong association with blood cell counts, the PLT PRS associated with only one of the tested hematologic diseases, i.e. MPN.

We observed the association of the PLT PRS with 12 non-hematologic QTs, including traits related to cardiovascular, liver, and kidney functions. The genetic links were further supported by findings of the IPA Core Analysis, which looked at changes in relevant canonical pathways and toxicological functions and searched for potential master regulators of these events. The association of the PLT PRS with blood lipids is not unexpected given reported molecular links (e.g., refs. 26,45), although a recent study did not observe a significant genetic correlation between PLT and blood lipids[44]. While platelets have been implicated in kidney and liver diseases[46–48], genetic sharing by PLT and markers of liver and kidney functions is not well established. A recent study reported a negative genetic correlation between PLT and total bilirubin, which is consistent with our findings. In contrast to our study demonstrating an association of the PLT PRS with alkaline phosphatase, gamma-glutamyl transpeptidase, and serum creatinine, they did not find the association with these traits[44]. Our study is, therefore, one of the first to provide evidence for genetic links between PLT and these traits.

We tested the PLT PRS for association with a large number of phenotypes representing the majority of common human diseases and observed a significant association with RA and AS. Additional support for the link between PLT and RA came from the DEPICT analysis, which identified synovial fluid, a relevant RA site, as one of the most significant target cell types and tissues. The IPA assessment of the *cis*-eQTL data also suggested that there are molecular mechanisms, such as those regulated by FOXO3, that could connect PLT and RA. While a link between PLT and AS has not been established, a positive correlation between platelets and severity of RA has been known for a long time in the clinical setting[49,50]. Clinical and animal studies have shown that platelet numbers are increased within the synovium and synovial fluid in RA, thus contributing to the pro-inflammatory environment of the synovium and potentially playing a role in the thrombus formation, destruction of cartilage, and alteration of synovial microcirculation observed in RA patients[51–55]. Of note, the previous studies[12,38,44] did not observe a significant genetic correlation/sharing between PLT and RA. This could be attributed to differences in the study design. Compared to those

studies, we derived the polygenic risk score using a larger number of variants and including the MHC region, and we had larger RA cohorts.

Our data also indicated a shared genetic basis for PLT and hypertension. Platelets may be an important cell type in development and consequences of hypertension: usually, platelets are necessary to maintain vascular integrity, but their enhanced activation may play a key role in the pathogenesis of hypertensive vascular disease[56–60]. In addition to hypertension, the PLT PRS was significantly correlated with a blood pressure/hypertension-related QT, i.e. mean arterial pressure. This association was further supported by findings from the DEPICT analysis, which indicated significant enrichment of arteries and endothelial cells in the data, and by results of the IPA Core Analysis that identified master regulators and molecular pathways relevant to the control of blood pressure. At the same time, genetic sharing by PLT and cardiovascular diseases, such as coronary artery disease (CAD), venous thromboembolism, and intracerebral hemorrhage did not reach the applied significance threshold of $1 \times 10^{-5}$, although each of these cohorts included from about 1,000 to 28,110 cases. Of note, although our findings for CAD did not reach the set significance threshold ($P_{meta} = 1.1 \times 10^{-5}$, Effect$_{meta} = 0.024$), they are consistent with the results presented by Astle et al.[12] who reported a weak inverse correlation between risk of CAD and MPV, opposite to what is expected (see ref. 61). This inverse relationship between the risk of CAD and MPV implies that the risk of CAD is positively correlated with PLT, which is in agreement with the observations in our study.

Two impressive reports of PLT (and other blood traits) meta-analyses and genetics of blood traits and diseases have recently been published[14,15]. While most signals that we identified in our meta-analysis are within 1 Mb range from those reported in these and earlier genome-wide association studies of PLT, three signals represent potentially novel PLT loci (discussed above). Apart from the identification of genome-wide significant signals associated with PLT, the major results of the two papers on the one hand and our study, on the other hand, are not overlapping, contradicting, or redundant. The main focus of those reports was on hematopoietic cells and hematologic disorders, whereas we did not restrict our analyses to hematopoietic cell types and disorders associated with them, acknowledging the relevance of multiple non-hematopoietic tissues to regulation of PLT. Another difference concerns the polygenic risk score analysis. In these two papers, the polygenic risk/trait score was constructed with 135–689 variants depending on the trait or was restricted to variant–trait associations that reached genome-wide significance in *trans*-ethnic MR-MEGA meta-analysis. However, we constructed our PRS based on 600,000 variants representing the whole genome without exclusion of any specific region. The authors tested their score in the prediction of hematological disorders or used them to evaluate portability of the PRS across European populations and association with rare blood disorders, whereas we tested ours in terms of genetic sharing between PLT and a wide range of diseases and traits, not only hematopoietic ones.

In summary, we identified multiple sequence variants associated with PLT and potentially new genes with functions in platelet biology. Our PRS analysis indicates that genetic sharing by PLT and red blood cell count is less than could be expected given their close developmental relationship. At the same time, the PLT PRS is much more predictive of counts of other blood cell types tested, albeit of only one of the tested hematologic diseases. Finally, we observed an association between the PLT PRS and MPN, RA, AS, hypertension, and benign prostate hyperplasia, as well as cardiovascular, anthropometric, inflammation, liver, and kidney function-related QTs, with further support provided by DEPICT and IPA analyses. These results

help shed light on diverse aspects of platelet biology and how it interacts with multiple biological processes in health and disease.

## Methods

**Datasets**. The meta-analysis combined genome-wide association studies of platelet count (PLT) in Iceland and UK Biobank datasets. Information on the PLT datasets is summarized in Supplementary Data 3. The Icelandic study included 139,479 chip-typed individuals and their 130,732 familially imputed 1st and 2nd degree relatives with available PLT measurements. PLT measurements used in GWAS were obtained from three health care centers: the National University Hospital of Iceland (LSH), the Icelandic Medical Center (Laeknasetrid) Laboratory in Mjodd (RAM) in Reykjavik, Iceland, and Akureyri Hospital (SAK) in Akureyri, Iceland. The data included all measurements made in these laboratories in the period from 1993 to 2015. The year of birth ranged from 1893 to 2015 (median year of birth 1969), and 52.2% were females. Mean PLT ± standard deviation (SD) was 253,900/µl ± 101,200/µl, with average 12.7 measurements per person. The National Bioethics Committee approved the study (reference number: VSN-15-023), including the protocol, methodology, and all documents presented to the participants. All individuals who donated samples provided written informed consent. All sample identifiers were encrypted in accordance with the regulations of the Icelandic Data Protection Authority. Personal identities of the participants and biological samples were encrypted by a third-party system approved and monitored by the Icelandic Data Protection Authority. The UK Biobank dataset included PLT data for 397,495 chip-typed individuals. Mean PLT ± SD was 252,400/µl ± 59,900/µl, with 1.05 measurements per individual. These individuals are participants in a large prospective cohort study of ~500,000 volunteer participants, who were recruited between the ages of 40 and 69 years in 2006–2010 across the UK[62,63]. All participants gave informed consent, and the UK Biobank's scientific protocol and operational procedures were reviewed and approved by the North West Research Ethics Committee (REC reference number 06/MRE08/65). Individuals whose data were used in the study were all of the genetically confirmed white British ancestry. This research has been conducted using the UK Biobank Resource under application numbers 24711 and 24898. In summary, the only inclusion criteria in either population were: (1) available platelet count measurements, (2) available genetic data, (3) written consent, and (4) for UK participants, confirmed white British ancestry. No other inclusion or exclusion criteria were applied with regards to study participants, and individuals with Mendelian disorders were not excluded from the analysis.

Most measurements, which were used to define various Icelandic QTs relevant to this study, were obtained from the three largest clinical laboratories in Iceland: (i) LSH (hospitalized and ambulatory patients); (ii) RAM (ambulatory patients), and (iii) SAK (hospitalized and ambulatory patients). Numerous cohorts representing the Icelandic QTs and case-control phenotypes, which were tested for association with the platelet count polygenic risk score (PLT PRS), have been described in our previous studies (e.g., refs. [64–80]). See Supplementary Data 3 for details on cohorts that represent traits associated with the PLT polygenic risk score.

Various health records and health-related information and genetic data are available for the UK Biobank and Icelandic cohorts. Disease definitions were primarily based on hospital electronic health records of ICD10 codes and self-reported data, as well as data from the Cancer Registry.

The UK cohort representing rheumatoid arthritis (RA) was defined based on electronic health records of the ICD10 codes M069, M059, and M060, and the definition of the cohort representing ankylosing spondylitis (AS) was based on ICD10 code M45. This matched the definition of the respective Icelandic cohorts. The UK Biobank study included 4001 RA cases with 404,652 controls, and 612 AS cases with 407,953 controls. The Icelandic RA cohort consisted of 1772 chip-typed and 537 familially imputed cases and 136,368 chip-typed and 203,503 familially imputed controls. The Icelandic AS cohort included 373 chip-typed and 60 familially imputed cases and 139,426 chip-typed and 131,349 familially imputed controls.

The Icelandic benign prostatic hyperplasia and associated lower urinary tract symptoms (BPH/LUTS) study population consisted of 13,928 men with symptomatic BPH/LUTS (9346 chip-typed and 4582 familially imputed individuals) and 104,000 controls (42,591 chip-typed and 59,630 familially imputed individuals). Controls were males not known to have symptomatic BPH/LUTS. The UK Biobank BPH/LUTS dataset consisted of 21,067 men with symptomatic BPH/LUTS, according to hospital-based diagnosis (ICD10 code N40), as well as 166,609 male controls not known to have been diagnosed with BPH/LUTS.

The myeloproliferative neoplasm cohorts were defined based on records from the Icelandic and British Cancer Registries. The Icelandic cohort included 333 cases (180 chip-typed and 153 familially imputed individuals) and 328,796 controls (141,889 chip-typed and 186,907 familially imputed individuals). The UK cohort consisted of 348 cases and 406,981 controls.

The Icelandic study included 32,026 chip-typed and 12,566 familially imputed individuals diagnosed with hypertension and 110,566 chip-typed and 220,357 familially imputed controls. The hypertension case definition originated from electronic health records and clinical evaluations. The control samples match the cases in age, sex, and county of origin. The UK Biobank study contained 77,566 individuals with hypertension and 331,087 controls. Hypertension was defined based on electronic health records of the ICD10 code I10.

**Genotyping**. Genotyping and imputation of the Icelandic samples were performed as described in Gudbjartsson et al.[81] and Jónsson et al.[82]. In short, we sequenced the whole genomes of 28,075 Icelanders using Illumina technology to a mean depth of at least 10× (median 32×). SNPs, deletions, and insertions were identified and their genotypes called using joint calling with the Genome Analysis Toolkit HaplotypeCaller (GATK version 3.4.07)[83]. Genotype calls were improved by using information about haplotype sharing, taking advantage of the fact that all sequenced individuals had also been chip-typed and long-range phased. About 32.64 million variants that passed the quality threshold were then imputed into 139,479 Icelanders, who had been genotyped with various Illumina SNP chips and their genotypes phased using long-range phasing[19,84] and for whom PLT measurements were available. Using genealogic information, the sequence variants were imputed into 130,732 untyped relatives of the chip-typed individuals to further increase the sample size for association analysis and increase the power to detect associations. For further information regarding genotyping and imputation we refer to Gudbjartsson et al.[81]. For comparison of data including both genotyped and familially imputed Icelanders vs. only genotyped Icelanders, see Supplementary Fig. 3 and Supplementary Data 23.

The UK Biobank genotyping was performed using a custom-made Affymetrix chip, UK BiLEVE Axiom[85], and with Affymetrix UK Biobank Axiom array[86]. In this study, we only used 33.9 million variants that were imputed based on the HRC reference panel[63].

**GWAS and meta-analysis**. Logistic regression assuming an additive model was used to test for association between variants and diseases, treating the disease status as the response and expected genotype counts from imputation as covariates, and using a likelihood ratio test to compute $P$ values. Prior to association analysis of quantitative traits, measurements were adjusted for sex, age, year of birth, measurement site, and population structure. Average of multiple measurements for an individual was used, and the measurements were normalized to a standard normal distribution using quantile normalization. Since there was a higher standard deviation in the Icelandic PLT data than in the UK PLT cohort (see above), which could be explained by more diversity within the Icelandic cohort with most measurements coming from the tertiary hospital (LSH), the normalization of PLT was done preserving the standard deviation in the original data. Quantitative traits were tested for association with genotypes using a linear mixed model implemented in BOLT-LMM[87]. All variants, which were tested, had imputation information over 0.8 in Iceland and over 0.7 in the UK. The association analysis for both the Icelandic and UK datasets was done using software developed at deCODE genetics[81]. For the Icelandic study group, patients and controls were matched on gender and age at diagnosis or age at inclusion. Information on the county of origin within Iceland was included as covariates to adjust for possible population stratification. For the UK datasets, cases and controls were restricted to individuals of genetically confirmed white British origin. Forty principal components were included in the analysis to adjust for population substructure. To account for inflation in test statistics due to cryptic relatedness and stratification, we applied the method of linkage disequilibrium (LD) score regression[88] to estimate the inflation in the test statistics and adjusted all $P$ values accordingly. The estimated correction factors for the phenotypes, which were found to correlate with the PLT PRS, are shown in Supplementary Data 24. The Q–Q plots are presented in Supplementary Fig. 4.

Variants in the UK imputation dataset were mapped to NCBI Build38 positions and matched to the variants in the Icelandic dataset based on allele variation. Results from the two study cohorts were combined using a Mantel–Haenszel model[89], in which the groups were allowed to have different population frequencies for alleles and genotypes but were assumed to have a common effect. Heterogeneity was tested by comparing the null hypothesis of the effect being the same in all populations to the alternative hypothesis of each population having a different effect using a likelihood ratio test. I2 lies between 0 and 100% and describes the proportion of total variation in study estimates that is due to heterogeneity.

We selected a threshold of imputation info > 0.8 and minor allele frequency (MAF) > 0.01% in Iceland and imputation info > 0.7 and MAF > 0.01% in the UK for variants available in both datasets. A more stringent quality threshold was used for the Icelandic dataset, since the imputation in that dataset is of higher quality, especially for rare variants, because the imputation is based on a large set of whole-genome sequenced Icelanders. Furthermore, the phasing of the Icelandic genetic data is much more reliable due to the large fraction of the population included and the use of long-range phasing. A total of 50,177,681 variants met these criteria. For detailed information, see Supplementary Data 1.

We used the weighted Holm–Bonferroni method[20] to account for all 50,177,681 variants being tested ($P$ value < (0.05*weight)/50,177,681). Using the weights given in Sveinbjornsson et al.[21], this procedure controls the family-wise error rate at 0.05. The following five variant classes were defined and the following significance thresholds were applied: (1) $P$ value $\leq 1.58 \times 10^{-7}$ for high-impact variants, including stop-gained and stop-loss, frameshift, splice acceptor, or donor and initiator codon variants ($n = 13,762$); (2) $P$ value $\leq 3.17 \times 10^{-8}$ for moderate-impact variants, including missense, splice-region variants, inframe deletions and insertions ($n = 268,662$); (3) $P$ value $\leq 2.88 \times 10^{-9}$ for low-impact variants, including synonymous, 3′ and 5′ UTR, and upstream and downstream variants ($n = 3,723,529$); (4) $P$ value $\leq 1.44 \times 10^{-9}$ for lowest-impact deep intronic and

intergenic variants in DNase I hypersensitivity sites (DHS) ($n = 6,728,845$); (5) $P$ value $\leq 4.8 \times 10^{-10}$ for other lowest-impact non-DHS deep intronic and intergenic variants ($n = 39,442,883$) (Supplementary Data 1).

**Conditional analysis**. We applied approximate conditional analyses, implemented in the GCTA software[90] to the meta-analysis summary statistics to identify independent association signals, using a stepwise model selection procedure. Variants were concluded to belong to an independent signal if their adjusted $P$ value was class-specific genome-wide significant (for applied genome-wide significance thresholds, see Supplementary Data 1). LD between variants was estimated using a set of 8700 whole-genome sequenced Icelandic individuals, assuming that variants more than 10 Mb away from each other are in complete linkage equilibrium. Due to the complexity and population differences in LD in the MHC region, the conditional analysis there was done directly on genotype data in the Icelandic and UK Biobank cohorts separately, and the results meta-analyzed to find the set of variants that explain the signal in that region. Only 16 secondary signals (Supplementary Data 2 and Note ‡) were identified in these analyses; all of them are located <1 Mb away from the primary signals.

In testing of the PLT variants for association with phenotypes identified in the PLT PRS analyses (see later), the variants were considered significant if their $P$ value was $\leq 3.6 \times 10^{-6}$, or 0.05 corrected with the number of tested variants (577) and tested phenotypes (24, excluding the discovery phenotype PLT). All significant associations were further tested in conditional analyses, using the methodology described above, to determine whether the variant itself is responsible for the observed effect.

**Polygenic risk score (PRS) and phenotype correlation analysis**. We used PRS analysis of the GWAS results for PLT to investigate its predictive power for other traits. We derived PLT PRSs both using Icelandic and UK datasets, as described in Kong et al.[36]. Briefly, the PRSs were calculated using genotypes for about 600,000 autosomal markers included on the Illumina SNP chips to avoid uncertainty due to imputation quality. We estimated linkage disequilibrium (LD) between markers using 14,938 phased Icelandic samples and used this LD information to calculate adjusted effect estimates using LDpred[36,91]. The effect estimates calculated using the Icelandic data were used as weights to generate the weighted PRS (PLT PRS$_{Ice}$) for testing in the UK, and the effect estimates generated from the UK data were used to derive the weighted PRS (PLT PRS$_{UK}$) for testing in Iceland. We created several PRSs assuming different fractions of causal variants (the $P$ parameter in LDpred). Subsequently, we selected one PLT PRS$_{UK}$ that was the most predictive of PLT in Iceland explaining 11.9% of the variance, and one PLT PRS$_{Ice}$ that was the most predictive of PLT in the UK explaining 10.9% of the variance (Supplementary Data 25). The most predictive PLT PRS$_{Ice}$ was used to analyze correlation with various phenotypes in the UK data, and the most predictive PLT PRS$_{UK}$ was tested for correlation with various phenotypes in Iceland. The correlation between the PLT PRS and phenotypes was calculated using logistic regression in R (v3.5) (http://www.R-project.org) adjusting for year of birth and principal components by including them as covariates in the analysis.

More than 3300 quantitative traits were available for the UK cohort presenting various lab measurements, various data for brain and its specific areas, ECG, measurements of body size and its different parts, metabolic rates, speech reception, birth weight, and so on, >3400 presenting various ICD codes, 39 lists from Cancer Registry, as well as data based on CISR and NCISR codes. Thousands of phenotypes (QTs, diseases defined based on e.g. the ICD codes, data from the Cancer Registry) were also available for Icelanders. Five thousand traits were considered to have a matching phenotype in the other cohort and were therefore included in the analysis, where we searched for phenotypes, which associate with the PLT PRS with the $P$ value threshold set at $1 \times 10^{-5}$ (0.05/5000). With regards to hematologic diseases, we performed the analysis of only those four hematologic diseases (Supplementary Data 4) where cohorts were defined in the same way, that is, data coming from the Cancer Registry of Iceland and the UK and diagnosis confirmed by pathologists.

**Enrichment analyses with DEPICT**. We performed Data-driven Expression Prioritized Integration for Complex Traits (DEPICT)[17] analysis of gene set enrichment, as well as analysis of the physiological system, tissue, and cell-type enrichment. We analyzed all 577 variants identified in our meta-analysis of PLT GWAS. We applied default settings in DEPICT, where all SNPs with LD > 0.5 with respect to the PLT lead SNPs in each locus are included in the analysis, with no additional adjustments or modifications of gene mapping. In the physiological system, tissue- and cell-type enrichment analysis, we tested whether genes in the PLT-associated regions, which were identified in our study, were highly expressed in any of the 209 Medical Subject Heading (MeSH) tissue and cell-type annotation categories that include 37,427 human Affymetrix HGU133a2.0 platform microarrays[17]. In the gene set enrichment analysis, we tested whether genes in the identified PLT-associated regions were enriched for reconstituted versions of the 14,461 gene sets used by DEPICT[17], and identified 758 significantly enriched gene sets ($P$ value $\leq 3.46 \times 10^{-6}$, or 0.05 corrected with 14,461 tested gene sets). DEPICT was also used to compute pairwise Pearson correlations between all reconstituted gene sets that were subsequently clustered by similarity using the affinity propagation method[92]. For each cluster with eight or more

significant gene sets, the member gene set with the lowest $P$ value was used as a representative in the pathway interaction network constructed based on the clusters' potential roles in hematopoiesis, inflammatory responses, signaling pathways, transcriptional regulation, RNA processing and translation, hemostasis, development and regulation of actin dynamics, cell adhesion and migration (Fig. 2). Interactions between the clusters were visualized with Cytoscape (https://cytoscape.org/).

**Cis-expression quantitative trait locus (cis-eQTL) analysis**. We performed cis-eQTL analysis to search for genes whose expression is affected by the 577 variants in cells and tissues that were identified as significantly enriched by the DEPICT analyses. In search for cis-eQTLs in blood and adipose tissues, we analyzed RNA-sequencing data from the whole blood of 13,206 Icelanders and adipose tissue from 750 Icelanders. cis-eQTLs in other tissues were analyzed in the corresponding GTEx datasets[41] (https://www.gtexportal.org/home/) and platelet and megakaryocyte datasets[29] (http://www.biostat.jhsph.edu/~kkammers/GeneSTAR/).

The Icelandic RNA-sequencing data were prepared as follows. For RNA preparation from blood, 2.5 mL of blood was drawn in Paxgene Blood RNA Tubes (PreAnalytiX). RNA was subsequently isolated using the Chemagic Total RNA Kit special (Perkin-Elmer, CMG-1084) on a Chemagic360 instrument following the manufacturer's protocol. The quality (RIN score) and quantity of isolated total RNA samples was assessed using the DNA 5 K/RNA chip for the LabChip GX (Perkin-Elmer).

For total RNA isolation from adipose tissue, adipose tissue samples were sectioned to appropriate size on dry ice and transferred to a pre-chilled 2 mL screw-cap tube containing a 5-mm stainless steel bead (Qiagen, Cat no. 69989). Samples were stored at $-80\,°C$ until isolation. RNAZol RT (Molecular Reasecrch Center, Inc. Cat no. RN190) was added to the frozen samples and incubated for 2 min at room temperature. The adipose samples were homogenized on a Tissue Lyser LT (Qiagen, Cat no 856000) according to the instrument manufacturer's instructions, centrifuged at $12,000 \times g$ for 5 min and the excess lipids removed from the top. Samples were then processed further following the manufacturer's instructions for total RNA isolation using RNAzol RT.

cDNA libraries derived from Poly-A mRNA were generated using Illumina's TruSeq RNA v2 Sample Prep Kit. Briefly, Poly-A mRNA was isolated from total RNA samples (0.2–1 μg input) using hybridization to Poly-T beads. The Poly-A mRNA was fragmented at $94\,°C$, and first-strand cDNA was prepared using random hexamers and the SuperScript II reverse transcriptase (Invitrogen). Following second-strand cDNA synthesis, end repair, the addition of a single A base, indexed adaptor ligation, AMPure bead purification, and PCR amplification, the resulting cDNA sequencing libraries were measured on the LabChip GX, diluted to 3 nM and stored at $-20\,°C$. Samples were pooled, clustered on to flowcells using either Illumina's cBot and the TruSeq PE cluster kits v4 (4 samples/ pool), or on NovaSeq S4 flowcells (24 samples/pool) using on-board clustering, respectively. Paired-end sequencing ($2 \times 125$ cycles) was performed with either HiSeq2500 instruments using the TruSeq SBS kits v5 from Illumina or NovaSeq instruments using S4 flowcells. Approximately 54–68 million (Q1–Q3) fragments were sequenced per sample.

RNA-sequencing integrity was inspected using parameters generated by FastQC[93]. Sequenced reads were aligned with STAR[94] for quality assurance of the RNA. Alignment files were processed with RNA-SeQC[95] and Picard CollectRnaSeqMetrics (https://broadinstitute.github.io/picard/) to estimate sequencing artifacts. Genotype concordance was determined by comparing imputed genotypes to those derived from genome alignment of RNA-sequencing reads. Transcript abundance was estimated with kallisto v0.43.1[96] using personalized transcriptome reference. Gene expression was computed by aggregating transcripts abundance. Association between variant and gene expression was estimated using a generalized linear regression assuming an additive genetic effect and normal quantile-transformed gene expression estimates, adjusting for measurements of sequencing artifacts, demography variables, blood composition and hidden covariates[97]. All variants within 5 Mb of each gene were tested. Top independent eQTL signals were produced by running iterative conditional association, each iteration adding genotypes of the eQTL with lowest $P$ value as a covariate into the model.

Only associations with $P$ value $\leq 1.52 \times 10^{-6}$ for blood (0.05 corrected with the number of tested variant-gene associations (32,778)) and $P$ value $\leq 8.8 \times 10^{-7}$ for adipose tissue (0.05 corrected with the number of tested variant-gene associations (56,868)) were considered significant and underwent further analyses. To address the question whether the variant itself is directly responsible for the observed effect on gene expression, we identified the topmost cis-eQTL signals and asked if our PLT variants represented these topmost cis-eQTLs or correlated with the variants representing the topmost cis-eQTLs. In case of correlation, the PLT variant and the topmost cis-eQTL variant were subjected to conditional cis-eQTL analyses to further confirm their effect on the gene expression.

To investigate the effect of the PLT variants on gene expression in other tissues and cell types identified as significant by DEPICT, we analyzed expression data for corresponding cells and tissues using two sources, namely the GTEx datasets[41], and eQTL data from platelets and megakaryocytes derived from iPSCs[29]. We analyzed all tissues, defined as significant by DEPICT, including non-hematopoietic ones, since they can also be relevant to control of PLT (e.g. liver and kidney, major production sites of thrombopoietin). In these analyses, we asked whether the PLT

variants presented the topmost cis-eQTL or were highly correlated with the variants presenting the topmost cis-eQTL ($R^2 \geq 0.95$). For eQTLs identified in the GTEx data, the significance thresholds applied were the same as defined in the GTEx datasets (see Supplementary Data 15). For eQTLs identified in the platelet and megakaryocyte data, the significance threshold was set at the calculated $q$ value $\leq 0.05$[29].

**Ingenuity pathway analysis (IPA)**. IPA (https://www.qiagenbioinformatics.com/products/ingenuitypathway-analysis, QIAGEN Inc.) is a bioinformatics tool that connects molecules into networks based on information on biomolecules and their relationships, which is gathered in the Ingenuity Knowledge Base[18]. The Core Analysis incorporated in IPA was used to interpret our data, in the context of biological processes, pathways, and networks.

Two types of statistical analysis are applied in IPA[18]. The first one is a $P$ value of overlap that estimates the overlap between the molecules in the dataset and a disease, biological function, or pathway. It is calculated using the right-tailed Fisher's exact test, with $P$ value $\leq 0.05$ indicating a statistically significant, non-random association. The second one is an activation $Z$ score that makes predictions about activation or inhibition of a molecule or a pathway by using the information on the direction of expression of the genes in the dataset and assessing the match between the observed and the predicted up- and downregulation patterns of gene expression. The positive $Z$ score means that the molecule, biological process or disease is trending towards an increase, and the negative $Z$ score indicates a trend towards a decrease, with activation $Z$ scores $\geq 2$ or $\leq -2$ indicating that the disease or function is statistically significantly increased or decreased, respectively.

To gain a comprehensive insight into diseases, molecular mechanisms, and physiological systems potentially affected by the genes at the PLT loci, we ran the IPA Core Analysis with the combined list of candidate genes identified in the cis-eQTL and coding variant analyses (Supplementary Data 16). It returned a list of 500 diseases and functions associated with the genes on our list with $P$ values ranging from $7.95 \times 10^{-4}$ to $1.42 \times 10^{-14}$ (Fig. 4). However, since these data (a list of gene names) do not provide information on the effect direction, which limits full exploitation of the other parts of the Core Analysis, we continued this investigation using the cis-eQTL data (Supplementary Data 15) instead of the gene list.

For consistency with the results of the PLT PRS analysis where associations were presented with respect to PLT increase, we used our cis-eQTL data expressed in terms of the PLT increasing alleles as an input into the IPA Core Analysis. To correct for multiple testing, the significance threshold in the analysis of canonical pathways and toxicological functions was set at Benjamini–Hochberg corrected $P$ value $\leq 0.05$. Fifteen toxicological functions (Supplementary Data 20) and eight canonical pathways (Supplementary Data 21) met this requirement.

The Causal Network Analysis within the IPA Core Analysis was applied to identify upstream molecules, or master regulators, that are predicted to orchestrate causal networks and control the expression of the genes in the dataset. The significance threshold was set at Benjamini–Hochberg corrected $P$ value $\leq 0.05$, with 279 endogenous molecules meeting this requirement (Supplementary Data 18). Fourteen of these 279 endogenous master regulators with Benjamini–Hochberg corrected $P$ value $\leq 0.05$ had the absolute values of the activation $Z$ score $\geq 2$ and were further investigated with respect to canonical pathways and diseases they affect, using the IPA's Ingenuity Knowledge Base (Supplementary Data 19). The eight significantly altered canonical pathways were also analyzed in terms of molecular mechanisms, diseases, and functions that they regulate, using the IPA's Ingenuity Knowledge Base (Supplementary Data 22).

**Reporting summary**. Further information on research design is available in the Nature Research Reporting Summary linked to this article.

## Data availability

The sequence variants from the Icelandic population whole-genome sequence data have been deposited at the European Variant Archive under accession code PRJEB15197. The GWAS summary statistics and data on the polygenic risk score are available at https://www.decode.com/summarydata/. The UK Biobank data can be obtained upon application (https://www.ukbiobank.ac.uk/). The authors declare that the data supporting the findings of this study are available within the article, its Supplementary Information file, and upon reasonable request. Data presented in Fig.1, showing the overview of association of the PLT variants with other quantitative traits with respect to the PLT increasing allele, are provided in Supplementary Data 6. Data presented in Fig. 2, which shows a network of gene sets identified in the DEPICT analyses, are provided in Supplementary Data 13 and 14. Data points presented in Fig. 3, showing the PLT variants that affect gene expression, are provided in Supplementary Data 15 and Supplementary Data 2 Note ł. Data points presented in Fig. 4 that shows the association of the 284 candidate causal PLT genes with diseases, molecular functions, and physiologic systems are provided in Supplementary Data 17. Data presented in Supplementary Fig. 1, which shows the Manhattan plot, are provided in the GWAS Summary Statistics. Data points presented in Supplementary Fig. 2, which shows the effects of the 577 PLT variants in Iceland vs. the UK, are provided in Supplementary Data 2. Data points presented in Supplementary Fig. 3, which compares effects of the 577 PLT variants in data including both genotyped and familially imputed Icelanders and genotyped only Icelanders, are provided in Supplementary Data 23.

## Code availability

In the study, we used publicly available software and datasets in conjunction with the above-described algorithms in the sequencing processing pipeline (whole-genome sequencing, association testing, RNA-sequencing mapping and analysis): Bedtools v2.25.0-76-g5e7c696z, https://github.com/arq5x/bedtools2/; BOLT-LMM, https://data.broadinstitute.org/alkesgroup/BOLT-LMM/downloads/; BWA 0.7.10 mem, https://github.com/lh3/bwa; Cytoscape, https://cytoscape.org/; DEPICT, https://data.broadinstitute.org/mpg/depict/; Genome browser, https://genome.ucsc.edu/; GenomeAnalysisTKLite 2.3.9, https://github.com/broadgsa/gatk/; OMIM, https://www.omim.org/; GTEx, https://www.gtexportal.org/home/; GWAS catalog, https://www.ebi.ac.uk/gwas/; LD link, https://ldlink.nci.nih.gov/?tab=ldpair; LDSC (LD Score), https://github.com/bulik/ldsc; Picard tools, https://broadinstitute.github.io/picard/; SAMtools 1.3, http://samtools.github.io/; Platelet/ megakaryocyte cis-eQTLs, http://www.biostat.jhsph.edu/~kkammers/GeneSTAR/; Variant Effect Predictor, https://github.com/Ensembl/ensembl-vep. Ingenuity Pathway Analysis (IPA) was performed using commercially available software from QIAGEN Inc. (https://www.qiagenbioinformatics.com/products/ingenuitypathway-analysis). We used R extensively to analyze data and create plots. No custom codes were created for this project.

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

## Author contributions

E.K.M., G. Thorleifsson, M.K.M., T.R., U.T., and K.S. designed the study and interpreted the results. G.B.R., S.Y.K., S. Reykdal, B.V., I.J.G., I.O., P.T.O., O.S., E.L.S., G. Grondal, A.J.G., and G. Geirsson carried out the subject ascertainment, recruitment, and collection of clinical data. E.K.M., M.K.M., T.R., A.H., J.G., H.H., S.S., I.J., and G. Thorgeirsson collected, processed, and analyzed the genotype and phenotype data. G. Thorleifsson, L.S., G.H., J.K.S., S. Rognvaldsson, P.M., K.N., S.H.L., V.T., and D.F.G. performed the statistical and bioinformatics analyses. E.K.M., G. Thorleifsson, T.R., U.T., and K.S., drafted the manuscript. All authors contributed to the final version of the paper.

## Competing interests

E. Mikaelsdottir, G. Thorleifsson, L. Stefansdottir, G. Halldorsson, J.K. Sigurdsson, S.H. Lund, V. Tragante, P. Melsted, S. Rognvaldsson, K. Norland, A. Helgadottir, M. K. Magnusson, J. Gudmundsson, H. Holm, S. Saevarsdottir, I. Jonsdottir, G. Thorgeirsson, D.F. Gudbjartsson, U. Thorsteinsdottir, T. Rafnar, and K. Stefansson, who are affiliated with deCODE genetics/Amgen, declare competing financial interests as employees. The remaining authors declare no competing financial interests. There are no non-financial competing interests to declare.
