## [Transparent Peer Review File · Communications Biology]

Reviewers' comments:

Reviewer #1 (Remarks to the Author):

Overall, an interesting and thorough analysis of platelet count genetics. Some questions remain, however. In general, I still had questions on several parts of the methods and think the authors need to better contextualize their results with the existing GWAS literature in a few places.

1. How was hematological trait filtering performed? Would those with Mendelian disorders have been removed? More description is needed for what was done, and how this is likely to impact GWAS results/implication of Mendelian disease gene variants.

2. Are the traits in Supplementary Table 4 the only hematological diseases tested? What about anemia, thrombocytopenia, etc? Could modest associations observed be in part due to exclusion of phenotypic outliers pre GWAS?

3. How many variants are included as significant that don't meet a conventional GWS threshold like 1×10^{-9} , and could only be identified with your functional variant upweighting scheme? Are these all variants in known loci at least? Why not use a more conservative threshold?

4. How does this list of loci compare to the Vuckovic et al loci identified? When is known variant list from? This isn't clear in Supplemental Table 2.

5. Re Supplementary Table 3... it would also be helpful to know what traits were tested and were NOT significant. Could a list be provided?

6. For this sentence... "with five variants associating with hypertension, one associating with RA and one associating with MPN (Supplementary Table 6)." Are these highly pleiotropic variants, for example in ABO locus? If so I'm not sure how useful this analysis is...

7. How many of the PRS associations are robust to exclusion of MHC region? What causal % parameter was used for the selected score? Could all variants and their weights included in the PRS used be presented in Supplemental Table? How about a table comparing the variance in PLT explained by different score parameter choices (this would be helpful for other people assessing PRS for other quantitative traits)?

8. When you say a gene is implicated in PLT biology or PLT disorders... how defined? More details needed in text. I see supplementary table 7, but am not clear what sources were used. Also what does the "?" mean?

9. How is mapping to genes accomplished for DEPICT and gene set enrichment analysis? It doesn't seem like you are limiting this to genes mapped either via a lead protein coding variant or cis eQTL. Are you? Also, how relevant are cis eQTLs mapped only in non hematopoietic tissues? Do you think these are likely leading you to the right target gene? I think more details/justification are needed here.

10. Any interesting genes in the ones implicated by coding variants, other than those known in PLT disorders? What about genes with high expression in megakaryocytes? Could you assess this systematically using gene expression data from BLUEPRINT or other sources?

11. Minor points on the figures- I would list significance threshold used for plotted variants in Fig 1. legend, and there are some issues with image quality in Fig. 2.

Reviewer #2 (Remarks to the Author):

Mikaelsdottir and colleagues report 577 independent associations in a large well-conducted analysis of 2 populations (Iceland and UK Biobank) along with downstream analyses in regards to polygenic risk scores (PRS), disease associations, eQTLs and pathway enrichments. Multiple testing was adjusted based on functional category. Due to recent large GWAS in the UK Biobank alone for PLT/other cell counts (Vukovic et al) and trans-ethnic GWAS in $\sim 746K$, most of the signals in the current paper are not novel, with 4 being reported as novel for PLT. There are some major and a few minor comments for the authors to consider.

1) Discussion should acknowledge that many of these variants are known in the most recent large work and cite Chen et al. Overall there is a lack of discussion of Chen et al. and Vukovic et al. as they covered many of the topics in the current work from different approaches (PRS, genetic correlation with traits, Mendelian variants/bleeding associated variants, etc). Chen et al. is cited but not really discussed and Vukovic et al. is not cited.

2) One of my biggest concerns is that the expression QTL analysis does not include megakaryocytes or platelets. By focusing on traditional GTEx and data from the Icelanders there is a major flaw. The importance of eQTL data in specific cell types is known, and demonstrated in platelets perhaps best by the characterization of a functional SNP in GRK5 that is highly platelet specific as supported by multiple experiments, eQTL analysis and colocalization in different tissues (see Rodriguez et al. PMID 32649586 for more details). There are now several platelet cis-eQTL datasets available, recognizing there may not be perfect overlap in typed/imputed variants to the current study. Most important to check is the eQTL dataset of Kammers et al. (PMID 33094331) which encompassed n=290 platelet samples (and also included megakaryocyte eQTL data on a smaller number of subjects). I suggest you expand the scope of your eQTL analyses to include platelet/MK eQTL data. As demonstrated by the example of GRK5 this approach could be a highly powerful approach, and if some of your findings seem platelet-specific you may also explore megakaryocyte epigenetics data (as Chen et al. showed the enrichment of PLT/MPV signals in MEP lineage cells). I recognize the Kammers et al. paper may have emerged while the authors completed or submitted their work (though there are a few smaller earlier works on platelet cis-eQTL as well).

3) The authors briefly mention there are 4 novel signals in the study (annotated in TableS2). It may be worth describing those briefly (two intergenic, 1 intronic to CAMTA1, one coding variant in ABCA6. On a quick check of Chen et al. sentinel variants (accounting for different genome builds), 3 of these signals had a GWAS significant MPV signal relatively nearby to the current PLT signals. Thus, these may reflect similar signals as noted by others and the authors here due to the PLT/MPV consistent relationship at most loci. The 4th locus an ABCA6 coding variant had nearby associations (3kb) with red cell traits in Chen et al. but interestingly not PLT/MPV.

4) There is not much discussion of the MPN association. It is interesting to note the TERT SNP association here. However, other genes traditionally affecting PLT like JAK2 and MPL are also known to be associated with MPN in clinical cases. I believe there may be more to discuss and unpack here. Were there rare/functional variants in JAK2 and MPL in the dataset and were they not associated with MPN?

5) Hematologic disorders mentioned here seem to involve blood cancers (white cell cancers would not necessarily be expected). What is the scope of disorders examined? (perhaps a list of codes?) Were bleeding and platelet disorders included? What about venous/arterial/cerebro vascular CVD? There is a long-running hypothesis about MPV increase being associated with increased CVD risk (and to some extent high PLT being associated with CVD) though the evidence is mixed. Astle et al. found CVD MR association in the opposite direction to that expected. It is unclear to me if these other disease etiologies would have been addressed in the current analysis. If so, the lack of positive findings may be worth some discussion.

6) The association of PLT with MPV is expected as the authors acknowledge. This is long known as the authors cite to several older papers. It is also illuminated by more recent genetic correlation analyses (i.e., Eicher et al. and Chen et al.). It is unclear how/if red cell correlations were examined. It is mentioned in the Discussion as nothing found but then why are red cell traits not listed in Table S3? Two papers are cited as recent – however, the most up to date and critical is Chen et al. 2020 – see their genetic correlation matrices (their Figure S6B) between blood cell traits where they show stronger PLT to WBC correlation and weak PLT to RBC traits.

7) Table S4 – Inherited platelet disorders – what is your reference? (ISTH Tier 1 Platelet/bleeding

genes/UK PanelApp as rather current) To my knowledge SH2B3 does not belong in inherited platelet disorders. CD36 arguably does not as well since most reported mutations are mild with no effect on bleeding diatheses or hemostasis.

8) Table S7 for platelet biology and platelet disorders seems somewhat incomplete. What is the source? (as above some sources suggested). Further, for platelet function/aggregation you may require an updated search. For instance the very same GRK5 SNP mentioned above associated here with PLT is not given which has evidence of strength similar to PEAR1.

9) Caution is needed in interpreting the PLT PRS as potentially suggesting causal links. Given the numerous loci involved and their complex biology these PRS associations may be driven by non-causal relationships as well as chance and other factors. In particular I find the Hypertension relationship to be one I am skeptical of given the modest effect size and modest association in Icelanders ($p=0.017$). The OR are rather weak in Table S6 by contrast with those of RA/MPN.

Minor comments

1) A lot of the later analyses are exploratory data analyses (enrichment, IPA, master regulators). While these can be illuminating there is also some circularity in these types of analyses (i.e., biases of existing databases/literature), and the fact that gene expression levels in specific cell types may drive many of these enrichments, they may be non-specific and categories with a high degree of overlap may not provide granular detail or insights. E.g., the enrichment of spleen associations is not surprising given that the spleen is the primary site of platelet destruction and any spleen or platelet expression dataset will nearly always show a high degree of overlap. At the end of the day I find myself somewhat lost in very large tables of gene lists and there is not a lot of surety about what is truly meaningful (without further experimentation or support). However, as mentioned above and below a few areas may be more interesting (steatosis, cancer, MPN) while others may not be (HTN)

2) Table S5 the note that this is with respect to PLT increasing allele is somewhat buried. The table might be easier to interpret if you also included in the Effects of the PLT trait as well (reinforces the increasing aspect and also allows side by side comparison of effect sizes).

3) In regards to Hepatic Steatosis is interesting to note that some of the predictive models that have been built in the literature include PLT as a covariate. It would be worthwhile to cite these. Likewise, you cannot wholly claim that is a novel link to this paper.

4) In re: to cancer and platelets beyond MPN, it may be worthwhile to read some of the works of Thomas Wurdinger on platelet transcriptomes as biomarkers for cancer types and consider citing some of that work

5) There are some attempts being made to capture published Polygenic Risk Scores to make them more transparent and reproducible due to known issues transferring these PRS across populations and ongoing research in that area. I suggest the authors consider if accepted depositing their PRS in such a resource (or making a clear Supplemental table of the PRS included and PRS calculations)

Reviewer #3 (Remarks to the Author):

Review: Platelet count: Sequence variants, its genetic associations with other traits and potential regulatory mechanisms

This manuscript by Mikaelsson et al. presents the results of a large-scale meta-analysis of platelet

count including summary data from GWAS of more than half a million European people. After identifying a large number of independent variants (4 are claimed as novel), the authors characterized potential mechanisms of action via coding variant and cis- expression quantitative trait locus (cis-eQTL) analyses and Data- driven Expression Prioritized Integration for Complex Traits (DEPICT). In addition they employed the Ingenuity Pathway Analysis (IPA) to look for enrichment of effects by pathway. Finally, they compared the predictive value of a polygenic risk score for platelet count among an array of additional available phenotypes. Overall, this paper represents a well-powered analysis coupled with hypothesis-generating downstream functional analyses that are likely to be of interest to the field.

Major comments:

1. This article needs a "table 1" summarizing the demographic information and platelet related measurements in each dataset for all included individuals.
2. The authors do not mention inclusion and exclusion criteria of subjects. Platelet counts could be greatly affected by variety of disease conditions and medications.
3. The effect of relatedness in the association tests from the genotypically imputed 1st and 2nd degree Icelanders should be explored more- it would be helpful to have one more column in supplemental table 2 to list the p-value from genotyped-Icelander-only, as a sensitivity analysis.
4. The author use the meta-analysis results to calculate the PLT PRS- additional details are warranted here. How good is the PLT PRS? AUC? What is the correlation between PLT and PLT PRS?
5. The authors identified 24 traits that are correlated with PLT PRS, how does this compare with the phenotypic correlation with the original measured PLT?
6. Given the likely high correlation of phenotypes, this portion of the study would be better powered if significance was achieved through permutation, rather than using bonferonni for multiple test corrections.
7. The authors also ran the single variant association test for the 577 PLT SNPs on the identified 24 traits. It would be interesting to also run the regression adjusting the original PLT measures in order to see the genetic effect separately from the phenotypic correlation.
8. Although authors state there is not significant heterogeneity in Icelander and UK Biobank, some p values are large in Icelanders but small in UK Biobank (UKB). For example, rs964184 (supplementary table 2) has p value of $1.9E-3$ in Icelander and $1.8E-25$ in UKB. Are such significant meta-analysis results driven by UKB data?
9. The authors should provide figures demonstrating that their test statistics are behaving appropriately given the complex relatedness structures and genotype imputation procedures which may introduce bias- the correction factor for case-control phenotypes which were found to coorelate with the PLT PRS are given, but it would be useful to have more information about any inflation seen in the primary GWAS/Meta analyses.

Minor comments:

1. Filtering criteria of imputation information over 0.8 and 0.7 were applied to Iceland and UK Biobank dataset respectively. What is the effect of these differing thresholds on low MAF ($< 1\sim 2\%$) in Iceland data?
2. The 4 novel PLT loci, should be better highlighted and specifically described. Did any of these show compelling associations with related traits?
3. IPA method is a bit of a black box (disclosure- I'm not a fan of this method, but am trying to set aside my bias here) and provides a network association rather than real causality. A formal causality test would be appropriate and potentially informative for the highlighted networks of interest.

Reviewers' comments:

Reviewer #1 (Remarks to the Author):

Overall, an interesting and thorough analysis of platelet count genetics. Some questions remain, however. In general, I still had questions on several parts of the methods and think the authors need to better contextualize their results with the existing GWAS literature in a few places.

1. How was hematological trait filtering performed? Would those with Mendelian disorders have been removed? More description is needed for what was done, and how this is likely to impact GWAS results/implication of Mendelian disease gene variants.

A: As stated in Methods, the Icelandic data included all measurements made in laboratories of the three main health care centers of Iceland in the period from 1993 to 2015. The UK data included all participants of genetically confirmed white British ancestry. All participants gave their written consent. In other words, the only inclusion criteria were: 1. Available platelet count measurements, 2. Available genetic data, 3. Written consent, and 4. For UK participants, confirmed white British ancestry. No other inclusion or exclusion criteria were applied with regards to study participants, and individuals with Mendelian disorders were not excluded from the analysis. They would however only be a very small fraction of the number of individuals included in the analysis and their impact on the association analysis would thus be very small if any. Especially as the trait values were inverse normal transformed prior to the analysis to diminish the impact of outliers.

This has now been stated in the revised version of the manuscript in the Methods section where the datasets are described (lines 371-375). The addition sounds as following: In summary, the only inclusion criteria in either population were: 1. Available platelet count measurements, 2. Available genetic data, 3. Written consent, and 4. For UK participants, confirmed white British ancestry. No other inclusion or exclusion criteria were applied with regards to study participants, and individuals with Mendelian disorders were not excluded from the analysis.

2. Are the traits in Supplementary Table 4 the only hematological diseases tested? What about anemia, thrombocytopenia, etc? Could modest associations observed be in part due to exclusion of phenotypic outliers pre GWAS?

A: Anemia, thrombocytopenia and hematologic diseases other than those listed Supplementary Table 4 were not analyzed for several reasons. First of all, some of the cohorts were too small to allow robust analysis. Secondly, some other cohorts were defined using different criteria in the two populations and therefore were not comparable. In some other cases, such as thrombocytopenia, there was no information on whether diagnosis was primary or secondary thrombocytopenia or both. We performed analysis of only those hematologic diseases where cohorts were defined in the same way, that is, data coming from the Cancer Registry of Iceland and the UK and diagnosis confirmed by pathologists. This has now been clarified in the Methods section, in the part where we describe Polygenic risk score and phenotype correlation analyses (lines 520-522). The addition sounds as following: With regards to hematologic diseases, we performed analysis of only those four hematologic diseases (Supplementary Table 4) where cohorts were defined in the same way, that is, data coming from the Cancer Registry of Iceland and the UK and diagnosis confirmed by pathologists.

Since the outliers/Mendelian disorder cases were not excluded, this is not likely to explain the observation of modest association.

3. How many variants are included as significant that don't meet a conventional GWS threshold like 1×10^{-9} , and could only be identified with your functional variant upweighting scheme? Are these all variants in known loci at least? Why not use a more conservative threshold?

A: The most commonly used GWS threshold is 5×10^{-8} , a threshold derived 15 years ago when much fewer variants were tested. Only recently some studies have started to use a more reasonable threshold like 5×10^{-9} (e.g. studies by Chen *et al* (PMID: 32888493) and Vuckovic *et al* (PMID: 32888494)). We actually use an even more stringent threshold, a Bonferroni adjustment for the number of tests performed, which in the case of our study is $p < 0.05/50,177,681$. The functional variant reweighting, which we apply, preserves that Bonferroni family wise error rate, i.e while functional variants are more likely to be causal and have to meet a less stringent significance threshold, most variants tested have to meet an even more stringent threshold of $p < 4.8 \times 10^{-10}$. Detailed justification for use of class-specific genome-wide significance thresholds is provided in Sveinbjornsson *et al* (PMID: 26854916).

In our study there are 19 variants whose P value is $> 1 \times 10^{-9}$, one of which is rs2315552, highly correlated with previously reported variant with ($r^2 > 0.9$) and represents the only signal in the region. The variants are all in or near known loci. As an example, one of these 19 variants is rs12731981, a missense in *MPL*. For details, see the table below.

If we consider significance thresholds applied in recent studies by Chen *et al* (PMID: 32888493) and Vuckovic *et al* (PMID: 32888494), where the applied significance threshold in meta-analyses was $p < 5 \times 10^{-9}$, there are only four variants (labeled with * in the table below) with adjusted p-value $> 5 \times 10^{-9}$. Three of these four variants are missense and one is splice acceptor.

Variants with adjusted P > 1E-09. Associations with adjusted P > 5E-09 are labeled with *. All the variants have adjusted p-value < 5E-08, the commonly used GWS threshold.

SNP*	Comment	Reported SNP(s)	Reference(s) (PMID)	Explanatory Note
rs12731981	Previously reported	rs12731981	32888494	
rs138843544	Previously reported	rs138843544	32888494	
rs17580	Previously reported	rs17580	27863252, 32888493	
rs3841260	Previously reported	rs3841260	27863252	
rs8050500	Previously reported	rs8050500	32888493, 32888494	
rs1143671*	Correlated with reported, R ² > 0.8	rs9815756	32888494	
rs8565*	Correlated with reported, R ² > 0.8	rs17685	32888494	
rs2315552	Correlated with reported, R ² > 0.8	rs1476835, rs917665	32888493, 32888494	
rs480243	Correlated with reported, R ² > 0.7	rs7990306	32888493	
rs3114409*	Correlated with reported, R ² > 0.6	rs3049697	32888494	
rs667555	Correlated with reported, R ² > 0.5	rs139892606	32888494	
rs562344339	Correlated with reported, R ² ≈ 0.2	rs572476245	32888494	It is a rare variant (MAF < 0.5% in either population) near the THPO gene, unadjusted P= 6.1E-35/adjusted P= 1.4E-09 .
rs1047440	No correlation with previously reported	rs6892249	32888493	Unadjusted P= 1.3E-09/adjusted P = 1.8E-09 .
rs3735485*	No correlation with previously reported	rs2331174	27863252, 32888493	rs3735485 is a missense variant representing a conditionally independent signal at the locus with unadjusted P=2.7E-09/adjusted P= 6.3E-09 . At this locus, we have also identified a signal with the lead SNP rs11762008 that is highly correlated with the reported rs2331174 (R ² =1).
rs35858667	No correlation with previously reported	rs7833924	27863252, 32888493, 32888494	rs35858667 is a missense variant representing a conditionally independent signal at the locus with unadjusted P=4.4E-25/adjusted P = 1.4E-09 . At this locus, we have also identified a signal with the lead SNP rs11993233 that is highly correlated with the reported rs7833924 (R ² =1).
rs35741412	No correlation with previously reported	rs368770339, rs7950696, rs7124681	27863252, 32888493, 32888494	rs35741412 is a missense variant representing a conditionally independent signal at the locus with unadjusted P= 4.3E-14/adjusted P = 2.6E-09 . At this locus, we have also identified a signal with the lead SNP rs3817334 that is highly correlated with the reported rs7124681 (R ² =1).

rs149678861	No correlation with previously reported	rs8035677, rs12911407, rs61009920, rs1719271, rs1522744, rs1002311, rs1631677	22139419, 27863252, 29403010, 32888493, 32888494	rs149678861 represents a rare (MAF < 1% in either population) conditionally independent signal at the locus with unadjusted P=3.7E-10/ adjusted P= 1.1E-09 . The reported SNPs are highly correlated (R ² ranging from 0.84 to 1) with rs141601939 and rs1719263, two other signals identified in this region in our study.
rs375677	No correlation with previously reported	rs708382, rs850736, rs34603233, rs150568286	22139419, 27863252, 32888493, 32888494	rs375677 represents a conditionally independent signal at the locus with unadjusted P=2E-13/adjusted P = 1.1E-09 . In this region, near the ITGA1B gene, we have also identified signals with the same lead SNPs as reported in the region (rs34603233 and rs150568286).
rs113906245	No correlation with previously reported	rs34536443, rs553314025, rs1865065, rs2360742	27863252, 29403010, 32888493, 32888494	rs113906245 represents a conditionally independent signal at the locus with unadjusted P=4.1E-10/ adjusted P = 2.4E-09 . At this locus, we have also identified a signal with the same lead SNP as reported in the region (rs34536443).

4. How does this list of loci compare to the Vuckovic et al loci identified? When is known variant list from? This isn't clear in Supplemental Table 2.

A: Of the 577 identified signals, 192 are represented by the same lead SNP as in the study by Vuckovic *et al* (PMID: 32888494). In addition, 235 variants are located within the 25 kb range from the variants identified in that study. Only 72 of 577 are located in the range 100kb or more away from the variants identified by Vuckovic *et al*.

The list of known variants, which our results were compared to, includes all variants reported in GWAS of platelet count, based on data from the GWAS catalog (<https://www.ebi.ac.uk/gwas/>). As for the initially submitted manuscript, the date of last access was November 13, 2020. In the revision, the date of access was May 26, 2021. The origin of the known variants list is now clarified in the manuscript text (lines 85-86), as well as in the **Supplementary Table 2** legend. The revised version in the manuscript text sounds as following: Three of the signals are located ≥ 1 Mb away from the nearest known PLT variant reported in the GWAS catalog as of May 26, 2021 (note § in **Supplementary Table 2**), indicating novel PLT loci. The addition to the Supplementary Table 2 legend sounds as following: § A variant located ≥ 1 Mb away from the nearest known PLT variant reported in the GWAS catalog as of May 26, 2021.

5. Re Supplementary Table 3... it would also be helpful to know what traits were tested and were NOT significant. Could a list be provided?

A: As mentioned in Methods and in the main text, in the polygenic risk score part of the study, we tested 5,000 traits representing the majority of common human diseases (coronary artery disease, type 2 diabetes, various cancers etc) as well as various quantitative traits.

We had data on thousands of phenotypes. For example, there were more than 3,300 quantitative traits from the UK (presenting various lab measurements, various data for brain and its specific areas, ECG, measurements of body size and its different parts, metabolic rates speech reception, birth weight and so on), more than 3,400 presenting various ICD codes, 39 lists from Cancer Registry, as well as data based on CISR and NCISR codes. 5,000 traits were considered to have a matching phenotype in the other cohort, therefore the threshold was set at $1E-05$ (0.05/5,000). The great majority of the phenotypes was not significantly associated with the PLT PRS. Only significant results are presented in the manuscript, as presenting the rest would be quite laborious. In the revised version, we included a clarification paragraph in the Methods section, in the part where we describe the Polygenic risk score and phenotype correlation analyses (lines 512-522).

6. For this sentence... ““with five variants associating with hypertension, one associating with RA and one associating with MPN (Supplementary Table 6).” Are these highly pleiotropic variants, for example in ABO locus? If so I'm not sure how useful this analysis is...

A: This sentence just summarizes which variants associate with the PLT PRS correlated diseases. While the variants are pleiotropic, for the sake of the paper's content only their association with the phenotypes correlated with the PLT PRS is emphasized.

7. How many of the PRS associations are robust to exclusion of MHC region? What causal % parameter was used for the selected score? Could all variants and their weights included

in the PRS used be presented in Supplemental Table? How about a table comparing the variance in PLT explained by different score parameter choices (this would be helpful for other people assessing PRS for other quantitative traits)?

A: This is a great question. We have repeated the analysis excluding the MHC region, and the results are now presented in **Supplementary Table 5**. Of the diseases that associate with the PLT PRS, association of ankylosing spondylitis and benign prostate hyperplasia are obviously driven by the MHC. However, association of hypertension and myeloproliferative neoplasms with the PLT PRS is not dependent on the MHC, whereas association of rheumatoid arthritis with the PLT PRS depends to some extent on the MHC, albeit not entirely. A description of these results is now also included in the manuscript text (lines 125-132). The addition sounds as following: To analyze the effect of the MHC region on association of the PLT PRS with these traits, we removed the variants, representing the MCH in the polygenic risk core calculations, and recalculated the association of the PLT PRS with these traits (**Supplementary Table 5**). Exclusion of the MHC revealed that the PLT PRS association with ankylosing spondylitis, benign prostate hyperplasia and cholesterol is dependent on the MHC region. While association with rheumatoid arthritis is strongly affected by the MHC region, the MHC does not fully explain association of the PLT PRS with rheumatoid arthritis. At the same time, exclusion of the MHC strengthened the PLT PRS association with hypertension, MPV, heart rate, alkaline phosphatase, and bilirubin (**Supplementary Table 5**).

For UKB the optimal model assumed 3% causal variants, but for Iceland the infinitesimal model in LDpred fitted best. The PRS are based on 600,000 variants which is too big to include as a supplementary table, but we could include the weights in a data release together with the meta-analysis results. The variance explained for the different models (fraction of causal variants) we tested in LDpred is presented in the table below and has been added as **Supplementary Table 24**.

Fraction of variance of PLT measurements explained by PLT-PRS scores created based on different re-weighting models in LDpred.

Model ^a	UKB	Iceland
Unweighted	4.4%	7.5%
1	6.0%	9.2%
0.3	7.1%	7.6%
0.1	9.0%	6.1%
0.03	10.9%	5.5%
0.01	3.2%	5.4%
0.003	1.5%	3.9%
0.001	1.5%	4.2%
Infinitesimal	7.5%	11.9%

^aAssumed fraction of causal variants in the LDpred model used.

8. When you say a gene is implicated in PLT biology or PLT disorders... how defined? More details needed in text. I see supplementary table 7, but am not clear what sources were used. Also what does the “?” mean?

A: Implication in platelet disorders was based on information from the Online Mendelian Inheritance in Man database (OMIM, www.omim.org) and confirmed through literature search. All gene names in the column “Gene Symbol - human (HUGO / HGNC / Entrez Gene)” in the new **Supplementary Table 16** were included in the search.

Selection of genes with roles in platelet biology was also based on results of search in OMIM and published literature. Criteria included evidence of biological functions, such as animal models that exhibit a platelet-related phenotype (preferably affecting number of platelets).

With respect to the four genes in the original **Supplementary Table 7** (now **Supplementary Table 8**), which do not have the OMIM phenotype numbers but were listed as having functions in platelets (ERG, NFE2, PEAR1, and ZFPM1/FOG1), three of them are transcription factors (TFs) and cofactors of known TFs, which are involved in megakaryopoiesis and implicated in platelet disorders, e.g. GATA1 (OMIM # 300367 and 300835), FLI1 (OMIM # 617443), and RUNX1 (OMIM # 601399). Furthermore, there is supporting evidence for these three TFs (ERG, NFE2, and ZFPM1/FOG1) in megakaryopoiesis/platelet biology coming from mouse models and/or studies of human subjects. Evidence for PEAR1 is not as solid as for ERG, NFE2 and ZFPM1/FOG1, but it is still based on several studies, including a zebrafish model.

To avoid ambiguity in definition of essential platelet genes, it was decided to limit the platelet gene list (new **Supplementary Table 8**) in the revised version only to those genes that have been implicated in human platelet-related disorders (preferably where platelet count is affected) with assigned OMIM phenotype numbers. This excludes the four genes discussed above.

Corresponding corrections and clarifications were added to the revised version of the manuscript, in the text (lines 154-157) and in the legends of **Supplementary Tables 2, 8, 9** and **16**. The revised text (lines 154-157) sounds as following: Only 38 out of the 577 variants are located in or near genes implicated in platelet disorders (**Supplementary Tables 2** and **8**) defined based on information from the Online Mendelian Inheritance in Man database (OMIM, www.omim.org) and confirmed through literature search.

“?” has been removed.

9. How is mapping to genes accomplished for DEPICT and gene set enrichment analysis? It doesn't seem like you are limiting this to genes mapped either via a lead protein coding variant or cis eQTL. Are you? Also, how relevant are cis eQTLs mapped only in non hematopoietic tissues? Do you think these are likely leading you to the right target gene? I think more details/justification are needed here.

A: In DEPICT, we analyzed all 577 variants identified in our meta-analysis of PLT GWAS. We applied default settings in DEPICT where all SNPs with LD > 0.5 with respect to the PLT lead SNPs in each locus are included in the analysis, with no additional adjustments or modifications of gene mapping. For details on the DEPICT method, please refer to Pers *et al* (PMID: 25597830). This clarification has been added to the Methods section in the revised version of the manuscript (lines 527-529) and sounds as following: We analyzed all 577 variants identified in our meta-analysis of PLT GWAS. We applied default settings in DEPICT, where all SNPs with LD > 0.5 with respect to the PLT lead SNPs in each locus are included in the analysis, with no additional adjustments or modifications of gene mapping.

Even though a cis-eQTL maps only in non-hematopoietic tissue, it can still be of relevance. For example, an encoded molecule can play a role in control of platelet count (e.g.

thrombopoietin, which is expressed by kidney and liver), or in recruiting platelets and/or targeting them to specific sites (e.g. adhesion molecules in endothelium). Therefore, it can be argued that it is reasonable to analyze all eQTLs, even if they map only in non-hematopoietic tissues. This clarification has been added (lines 600-602) and sounds as following: We analyzed all tissues, defined as significant by DEPICT, including non-hematopoietic ones, since they can also be relevant to control of PLT (e.g. liver and kidney, major production sites of thrombopoietin).

10. Any interesting genes in the ones implicated by coding variants, other than those known in PLT disorders? What about genes with high expression in megakaryocytes? Could you assess this systematically using gene expression data from BLUEPRINT or other sources?

A: Of 115 signals represented by coding variants or variants highly correlated with coding variants, 85 are located in genes expressed in megakaryocytes and/or platelets, based on expression data from platelets and megakaryocytes from Kammers *et al* (PMID: 33094331) and expression data from platelets from Londin *et al* (PMID: 24524654), with 18 of them mapping to genes implicated in platelet-related disorders. The remaining 30 variants are located in 29 genes that do not have a record of expression in these two cell types. However, several of them are noteworthy: ABCA6, APOH, GCKR, TM6SF2, IRF1, and PNPLA3 that are implicated in cholesterol/lipid homeostasis. By analogy with ABCG4 (Murphy *et al*, (PMID: 23584088)), they could function both in platelet count regulation and control of cholesterol/lipid homeostasis. Alternatively, by modifying cholesterol/lipid homeostasis they might create a pro-inflammatory environment that leads to increase in platelet production. Interestingly, all these genes are expressed in liver, with three being linked to non-alcoholic fatty liver disease (see **the table below**). Further experimental evidence is required to elucidate their role in both regulation of PLT and control of lipid metabolism/homeostasis. This description has been included in the Results section of the revised version of the manuscript, where we discuss pathway and tissue enrichment analyses and search for causal genes (lines 165-176), as well as a new **Supplementary Table 11**.

Genes not expressed in platelets or megakaryocytes and harboring coding PLT variants with functions in cholesterol/lipid homeostasis and non-alcoholic fatty liver disease. Added as a new **Supplementary Table 11**.

Gene	Known functions in cholesterol/lipid homeostasis and non-alcoholic fatty liver disease	References (PMID)
ABCA6	ABCA6 belongs to the ATP-binding cassette transporter family, it is cholesterol-responsive and potentially involved in intracellular lipid transport processes. Variants in the gene have been associated with blood lipids in GWAS but not with PLT. No record of expression in platelets or megakaryocytes, although broadly expressed in other tissues, including liver (GTEx).	11478798, 24028821, 24524654, 25751400, 25961943, 33094331
APOH	APOH encodes beta-2 glycoprotein I, also known as apolipoprotein H, a single-chain plasma protein and a component of circulating plasma lipoproteins. Beta-2 GPI has been implicated in a variety of physiologic pathways, including blood coagulation, hemostasis, and the production of antiphospholipid antibodies characteristic of antiphospholipid syndrome. Variants in the gene have been associated with blood lipids in previous GWAS. Restricted expression towards liver (GTEx), no records of expression in platelets or megakaryocytes.	15507263, 24097068, 24524654, 29507422, 30275531, 30698716, 33094331

GCKR	The protein encoded by GCKR belongs to the GCKR subfamily of the Sugar Isomerase family of proteins. The gene product is a regulatory protein that inhibits glucokinase in liver. In addition to diabetes, variants in the gene have been associated with non-alcoholic fatty liver disease in GWAS and multiple variants were found to associate with blood lipids. Expressed in several tissues with very abundant expression in liver (GTEx) and no record of expression in platelets and megakaryocytes.	17463246, 19936222, 24524654, 25692341, 25961943, 29385134, 33094331
IRF1	The protein encoded by IRF1 is a transcriptional regulator and tumor suppressor, serving as an activator of genes involved in immune responses and associated with cancers. It also plays a role in transcriptional regulation of APOL1, a component of high-density lipoprotein particles. Variants in the gene have been associated with blood lipids in previous GWAS. Expressed in various tissues including liver (GTEx) with no record of expression in platelets and megakaryocytes.	24524654, 29083408, 29507422, 29599126, 30275531, 32671843, 33094331, 33476326
PNPLA3	The protein encoded by PNPLA3 is also known as adiponutrin and associates with endoplasmic reticulum and lipid droplet membranes. It is an enzyme with lipase activity towards triglycerides and retinyl esters, and acyltransferase activity on phospholipids and may be involved in regulation of energy homeostasis. Variants in the gene have been associated with non-alcoholic fatty liver disease and blood lipids in GWAS. Expressed in liver and several other tissues (GTEx) with no record of expression in platelets or megakaryocytes.	11431482, 21423719, 22719876, 23535911, 24524654, 26690388, 28334899, 29385134, 29507422, 29935383, 30275531, 33094331, 33339817
TM6SF2	TM6SF2 is a multi-pass membrane protein. Its inhibition is associated with reduced secretion of triglyceride-rich lipoproteins and increased cellular triglyceride concentration and lipid droplet content, whereas its overexpression reduces liver cell steatosis. A missense variant in TM6SF2 (Glu167Lys), which has been associated with increased risk for non-alcoholic fatty liver disease and with blood lipids in a GWAS induces increased hepatocyte fat content by reducing APOB particle secretion. Expressed mainly in small intestine and liver (GTEx), with no record of expression in platelets or megakaryocytes.	24524654, 24531328, 24927523, 25961943, 31406127, 33094331

11. Minor points on the figures- I would list significance threshold used for plotted variants in Fig 1. legend, and there are some issues with image quality in Fig. 2.

A: The significance threshold has been added to **Fig.1** legend (lines 867-868). The revised version sounds as following: **Fig. 1.** Overview of association of the PLT variants with other quantitative traits (significance criteria: $p \leq 3.6 \times 10^{-6}$ (**Methods**)). The image quality of **Fig.2** has been improved.

Updated Fig. 2.

Reviewer #2 (Remarks to the Author):

Mikaelsdottir and colleagues report 577 independent associations in a large well-conducted analysis of 2 populations (Iceland and UK Biobank) along with downstream analyses in regards to polygenic risk scores (PRS), disease associations, eQTLs and pathway enrichments. Multiple testing was adjusted based on functional category. Due to recent large GWAS in the UK Biobank alone for PLT/other cell counts (Vukovic et al) and trans-ethnic GWAS in ~746K, most of the signals in the current paper are not novel, with 4 being reported as novel for PLT. There are some major and a few minor comments for the authors to consider.

1) Discussion should acknowledge that many of these variants are known in the most recent large work and cite Chen et al. Overall there is a lack of discussion of Chen et al. and Vukovic et al. as they covered many of the topics in the current work from different approaches (PRS, genetic correlation with traits, Mendelian variants/bleeding associated variants, etc). Chen et al. is cited but not really discussed and Vukovic et al. is not cited.

A: In the revised version, we have added a discussion of the publications by Chen *et al* (PMID: 32888493) and Vukovic *et al* (PMID: 32888494) that highlights their findings and approaches in comparison to our discoveries (lines 298-314).

Specifically, if our manuscript is compared to the study by Chen *et al*, it becomes clear that although we work on similar topics, the approaches are different, as well as questions addressed, and most of the results are not overlapping, contradicting or redundant. Chen *et al* do an impressive job addressing issues of ethnicity with focus on hematopoietic tissues, whereas we did not restrict our analyses to hematopoietic tissues only, for reasons discussed in our response to Major Comment 2 below. Furthermore, while Chen *et al* look at chromatin states in hematopoietic cell types, we look at the consequences in form of altered gene expression with speculation on how the differential expression of the genes in our dataset could potentially lead to observed outcomes (e.g. diseases associated with the PLT PRS). With regards to the polygenic risk score/polygenic trait score part, Chen *et al* restricted these analyses to variant-trait associations that reached genome-wide significance ($p < 5 \times 10^{-9}$) in trans-ethnic MR-MEGA meta-analysis, while we constructed our PRS based on 600,000 variants representing the whole genome without exclusion of any specific region. Chen *et al* tested their score in prediction of hematological disorders, while we tested ours in terms of prediction of various diseases and traits, not only hematopoietic ones.

The same is true when our study is compared to the impressive work and interesting results presented by Vukovic *et al* in their study. Their cohort includes 563,085 Europeans, which is of a similar size to ours. They explored associations with 29 blood cell phenotypes. The only eQTLs considered were the ones in blood cell types, as opposed to multiple tissues in our study, and these data were used for a different purpose than in our study. Majority of findings in the core gene analysis are related to hematopoiesis and/or hematologic disorders. Like Chen *et al*, they also looked at chromatin states in hematopoietic cell populations and how the variants may affect human transcription factor motifs, while we were interested to analyze consequences in terms of altered gene expression in our dataset with regards to outcomes, e.g. the PLT PRS associated diseases. Vukovic *et al* analyzed clinical impact of rare variants, while we evaluated association of all 577 genome-wide significant variants with all traits associated with the PLT PRS. They constructed PRS with 135-689 variants depending on the trait and used them to evaluate portability of the PRS across European populations and to look at association with rare blood disorders, while we constructed ours based on 600,000 variants representing the whole genome and analyzed associations with phenotypes representing the majority of common diseases and traits. They looked if multiple sentinel at a single locus could underlie associations with complex disease, where they took blood trait

loci with 2+ sentinels and overlapped those regions with colocalization results for 18 common human diseases, which is also a different approach from ours. Furthermore, the results of this colocalization with 18 common diseases, which are presented in Fig. 5E in their paper, are not specific for platelets, but rather represent colocalization with at least one locus associated with any of the blood cell counts.

As mentioned above, a paragraph summarizing these points have been added to the **Discussion** section of the revised version.

2) One of my biggest concerns is that the expression QTL analysis does not include megakaryocytes or platelets. By focusing on traditional GTeX and data from the Icelanders there is a major flaw. The importance of eQTL data in specific cell types is known, and demonstrated in platelets perhaps best by the characterization of a functional SNP in GRK5 that is highly platelet specific as supported by multiple experiments, eQTL analysis and colocalization in different tissues (see Rodriguez et al. PMID 32649586 for more details). There are now several platelet cis-eQTL datasets available, recognizing there may not be perfect overlap in typed/imputed variants to the current study. Most important to check is the eQTL dataset of Kammers et al. (PMID 33094331) which encompassed n=290 platelet samples (and also included megakaryocyte eQTL data on a smaller number of subjects). I suggest you expand the scope of your eQTL analyses to include platelet/MK eQTL data. As demonstrated by the example of GRK5 this approach could be a highly powerful approach, and if some of your findings seem platelet-specific you may also explore megakaryocyte epigenetics data (as Chen et al. showed the enrichment of PLT/MPV signals in MEP lineage cells). I recognize the Kammers et al. paper may have emerged while the authors completed or submitted their work (though there are a few smaller earlier works on platelet cis-eQTL as well).

A: As acknowledged by the reviewer, Kammers *et al* paper (PMID: 33094331) was emerging at the time when our manuscript was about to be submitted. Platelet and megakaryocyte eQTL data reported in this paper have now been analyzed and significant eQTLs identified are now included in data presented in **Supplementary Table 15** and referred to in the manuscript text (lines 190-192, 550-551, 600, 606-607). The IPA analyses have been repeated with these data included, and their results are presented in the updated **Supplementary Tables 17-22**, with corrections in the text where applicable (lines 198-201, 211-213, 217-222, 629, 643-648).

While we agree that megakaryocytes and platelets are of primary importance in the eQTL analysis, we note that tissues other than platelets and megakaryocytes were defined by our DEPICT analysis as significantly enriched for our set of the 577 genome-wide significant PLT variants. Moreover, essential molecules, such as thrombopoietin, are mainly produced elsewhere, not in platelets. Therefore, it was reasonable and justifiable to look at GTeX, which includes data for *e.g.* liver and kidney, major production sites of thrombopoietin, and the Icelandic RNA sequencing dataset, which includes data for whole blood (*i.e.* cell types genetically related to platelets and enriched as per the DEPICT results) available for about 15,000 individuals with very precise genotyping information. This has been added to the text (lines 600-602), and the addition sounds as following: We analyzed all tissues, defined as significant by DEPICT, including non-hematopoietic ones, since they can also be relevant to control of PLT (*e.g.* liver and kidney, major production sites of thrombopoietin).

The study by Rodriguez *et al* (PMID: 32649856) was just emerging when our manuscript was undergoing the final in-house review before the submission.

3) The authors briefly mention there are 4 novel signals in the study (annotated in TableS2). It may be worth describing those briefly (two intergenic, 1 intronic to CAMTA1, one coding variant in ABCA6).

A: After taking Vuckovic *et al* (PMID: 32888494) paper into consideration, the signals, which are located ≥ 1 MB from previously reported in GWAS and potentially representing novel PLT loci, are three (based on data from the GWAS catalog accessed on May 26, 2021). A paragraph describing potential genes in each of the regions has now been added to the Results section of the revised version (lines 85-108).

rs77542162 is a missense variant in ABCA6. We found that it is also associated with blood lipids, alkaline phosphatase, and height (**Supplementary Table 6**). There are no previously reported associations with platelet traits (PLT or MPV) in the 2MB window surrounding the SNP, whereas it has been associated with blood lipids (for instance, see van Leeuwen *et al* (PMID: 25751400) and Surakka *et al* (PMID: 25961943)), and some other variants in the area (about 0.5 MB from rs77542162) are associated with height. The variant is in the gene ABCA6 that belongs to the ATP-binding cassette transporter family, it is cholesterol-responsive and potentially involved in intracellular lipid transport processes (see Kaminski *et al* (PMID: 11478798) and Gai *et al* (PMID: 24028821)). While the exact mechanisms of ABCA6 involvement in control of these traits are yet to be uncovered, there is a known example of ABCG4, a related transporter that regulates cholesterol efflux and platelet production/number. For details, see Murphy *et al* (PMID: 23584088).

rs7808461 on chr7 and rs2118446 on chr2 have no previous reports of association with PLT either, although variants nearby have been reported in association with MPV, which is the only trait besides PLT that we detected associations with in our study. While rs2118446 is correlated with previously reported MPV variants rs2258404 and rs2577595 (Chen *et al* (PMID: 32888493) and Vuckovic *et al* (PMID: 32888494), respectively) with $R^2 > 0.8$, correlation of rs7808461 with the reported variants rs3218477, rs3218455, rs145530953 and rs3218502 (Aistle *et al* (PMID: 27863252), Chen *et al* (PMID: 32888493) and Vuckovic *et al* (PMID: 32888494)) is $R^2 < 0.2$. In addition, it should be noted that while PLT and MPV are genetically related, association with one trait does not automatically mean association with the other one. For example, of 577 variants associated with PLT in our study only 33% (191 variants) associate also with MPV.

For rs7808461 on chr7, we did not observe significant cis-eQTLs in the tissues that were tested, nor in other tissues in GTEx. The variant is, however, located near (about 75 Kb) *ACTR3B*, which encodes a member of the actin-related proteins and may have a regulatory role in the actin cytoskeleton and induce cell-shape change and motility. The gene is expressed in platelets and megakaryocytes (expression data for platelets and megakaryocytes from Kammers *et al* (PMID: 33094331) and expression data for platelets from Londin *et al* (PMID: 24524654), although its precise role in platelets and megakaryocytes is yet to be determined. Another gene in proximity is *XRCC2* (less than 10 kb away), a member of the *RAD51* gene family, which is involved in homologous recombination repair of DNA damage. It does not seem to be expressed in platelets and megakaryocytes according to data from Kammers *et al* and Londin *et al*, making it less convincing as a candidate gene, since the variant's associations, which we detected, are platelet-specific (PLT and MPV).

Similar to rs7808461, rs2118446 associated only with platelet traits in our data. For rs2118446, we found one cis-eQTL that involves gene expression of *GCC2* in the adipose tissue (**Supplementary Table 15**). *GCC2* (GRIP and coiled-coil domain containing 2) is a peripheral membrane protein localized to the trans-Golgi network and is required for endosome-to-Golgi transport and maintenance of Golgi structure (Luke *et al*

(PMID: 12446665) and Derby *et al* (PMID: 17488291). According to expression data from Kammers *et al* and Londin *et al*, the gene is also expressed in platelets and megakaryocytes, although no significant cis eQTLs involving rs2118446 and *GCC2* were detected in these two cell types. The exact role of the gene in platelets and/or megakaryocytes is not known.

On a quick check of Chen *et al*. sentinel variants (accounting for different genome builds), 3 of these signals had a GWAS significant MPV signal relatively nearby to the current PLT signals. Thus, these may reflect similar signals as noted by others and the authors here due to the PLT/MPV consistent relationship at most loci. The 4th locus an *ABCA6* coding variant had nearby associations (3kb) with red cell traits in Chen *et al*. but interestingly not PLT/MPV.

While there is a consistent relationship between PLT and MPV, as noted by the reviewer, association of a SNP with one trait does not automatically translate into association with the other trait. Of the 577 variants associated with PLT in our study, only one third (191) also associate with MPV. Secondly, it should be pointed out that variants, located near our PLT variant rs7808461 and reported to associate with MPV (PMID: 27863252, 32888493, 32888494), correlate with rs7808461 with $R^2 < 0.2$, thus very likely representing a different signal.

As for the *ABCA6* missense variant, rs77542162, we replicated its previously reported association with total cholesterol (PMID: 25961943, 29507422, 30275531) and report a novel association with PLT ($\text{Effect}_{\text{comb}} = -2.96 \times 10^3 / \mu\text{l}$, $P_{\text{comb}} = 8.7 \times 10^{-11}$, where $\text{Effect}_{\text{Ice}} = -2.93 \times 10^3 / \mu\text{l}$, $P_{\text{Ice}} = 7.7 \times 10^{-3}$ and $\text{Effect}_{\text{UK}} = -2.97 \times 10^3 / \mu\text{l}$, $P_{\text{UK}} = 3.3 \times 10^{-9}$ with $P_{\text{het}} = 0.98$). No association with MPV was observed for this variant in our study. For an example of an ABC transporter involved both in efflux of cholesterol and regulation of platelet production, see Murphy *et al* (PMID: 23584088).

These points have now been added to the revised version of the manuscript (lines 90-91, 96-101, 234-237).

4) There is not much discussion of the MPN association. It is interesting to note the TERT SNP association here. However, other genes traditionally affecting PLT like JAK2 and MPL are also known to be associated with MPN in clinical cases. I believe there may be more to discuss and unpack here. Were there rare/functional variants in JAK2 and MPL in the dataset and were they not associated with MPN?

A: MPN association was not discussed *per se*, but it is definitely noteworthy. It could be considered a positive control/proof of principle for the PLT PRS, since in our study we used the total cohort of myeloproliferative neoplasms. The cohort was used without further splitting into essential thrombocythemia, polycythemia vera and primary myelofibrosis due to very small sizes of cohorts representing each of these diseases. It could therefore be speculated that the PLT PRS association with MPNs is expected and is potentially driven by the essential thrombocythemia subset of the MPN cohort (which could not be addressed due to the cohort sizes).

Our dataset includes variants with minor allele frequencies (MAF) down to 0.05%. There is a missense variant in *MPL*, rs12731981, with MAF 1.63% in Iceland and 3.21% in the UK. There is also a rare intronic variant in *JAK2* rs12004239 with MAF 0.8% in Iceland and 0.92% in the UK. However, neither seems to have an effect on PLT that would likely have pathologic consequences. On par with this prediction, we did not observe association of either variant with MPN given the set p-value threshold.

All variants associated with PLT in both populations, those of them correlated with coding variants, their minor allele frequencies and all significant associations with MPN are presented in **Supplementary Tables 2, 7 and 9**.

5) Hematologic disorders mentioned here seem to involve blood cancers (white cell cancers would not necessarily be expected). What is the scope of disorders examined? (perhaps a list of codes?) Were bleeding and platelet disorders included? What about venous/arterial/cerebro vascular CVD? There is a long-running hypothesis about MPV increase being associated with increased CVD risk (and to some extent high PLT being associated with CVD) though the evidence is mixed. Astle et al. found CVD MR association in the opposite direction to that expected. It is unclear to me if these other disease etiologies would have been addressed in the current analysis. If so, the lack of positive findings may be worth some discussion.

A: As mentioned in **Methods** and in the main text, in the polygenic risk score part of the study, we tested 5,000 traits representing the majority of common human diseases (coronary artery disease, type 2 diabetes, various cancers etc) as well as various quantitative traits.

Hematologic diseases other than those listed in **Supplementary Table 4** were not analyzed for several reasons. First of all, some of cohorts were too small. Secondly, some other cohorts were defined using different criteria in the two populations and as a result were not comparable. For yet another cases, such as thrombocytopenia, there was no information on whether diagnosis was primary or secondary thrombocytopenia or either one. We performed analysis of only those hematologic diseases where cohorts were defined in the same way, that is, data coming from the Cancer Registry of Iceland and the UK and diagnosis confirmed by pathologists. These points have now been added to the **Methods** section.

We had data for thousands of phenotypes. For example, there were more than 3,300 quantitative traits from the UK (presenting various lab measurements, various data for brain and its specific areas, ECG, measurements of body size and its different parts, metabolic rates speech reception, birth weight and so on), more than 3,400 presenting various ICD codes, 39 lists from Cancer Registry, as well as data based on CISR and NCISR codes. 5,000 traits were considered to have a matching phenotype in the other cohort, therefore the threshold was set at $1E-05$ ($0.05/5,000$). The great majority of the phenotypes was not significantly associated with the PLT PRS. Only significant results are presented in the manuscript, as presenting the rest would be quite laborious. In the revised version, we included a clarification paragraph in the Methods section, in the part where we describe the **Polygenic risk score and phenotype correlation analyses** (lines 512-522).

Of note, several diseases reached meta p-values of association with the PLT PRS that was just above the set threshold. Since these results are considered not significant given the set threshold, they were not included. These diseases are presented in the table below, among which there are coronary artery disease and pure hypercholesterolemia:

Diseases with meta p-values close to the threshold

Disease	Iceland (cases/controls)	UK (cases/controls)	Effect _{meta}	CI	P _{meta}	P _{het}
Coronary artery disease	19,129/125,489	28,110/380,455	0.024	(0.014, 0.035)	1.1E-05	0.58
ICD10 N39	10,184/132,482	19,252/389,401	0.019	(0.010, 0.029)	9.0E-05	0.55
ICD10 E780	1,203/140,923	33,184/375,469	0.021	(0.011, 0.031)	5.5E-05	0.21

Intracerebral hemorrhage with about 1,000 cases in each cohort was non-significant ($p > 0.05$). The same was also true for venous thromboembolism represented by 2,650 cases in Iceland and 6,956 cases in the UK.

As acknowledged by the reviewer, there is mixed evidence with regards to MPV or PLT increase associating with increased CVD risk. According to this, both increase in MPV and increase in PLT are associated with increased risk of CVD, although in general PLT and MPV are inversely related. In other words, if increase in MPV is correlated with increased risk of CVD, PLT should be decreased. However, it could be speculated that this association with CVD might be based on a subset of those few variants that associate with PLT and MPV with the same direction of effect (17 identified in our study, marked with § in **Supplementary Table 6**).

Associations with CVD observed in our study were just above the significance threshold at the best, as mentioned above. Potentially, they could be strengthened if the polygenic risk scores were calculated using a subset of variants, such as those associating with PLT and MPV with the same direction of effect. This is, however, outside of the scope of our study, as we aimed at analyzing associations of the PLT PRS constructed with data representing the whole genome, not just specific regions or functional subsets.

Of note, although our findings for coronary artery disease (CAD) did not reach the set significant threshold, they are consistent with the results presented by Astle et al (PMID: 27863252) who reported a weak inverse association between risk of CAD and MPV. The inverse relationship between risk of CAD and MPV would imply that risk of CAD is in turn positively correlated with PLT, which is in agreement with the observation in our study.

This point has now been added to the Discussion section of the revised version (lines 289-297) and sounds as following: At the same time, genetic sharing by PLT and cardiovascular diseases, such as coronary artery disease (CAD), venous thromboembolism, and intracerebral hemorrhage did not reach the applied significance threshold of 1×10^{-5} , although each of these cohorts included from about 1,000 to 28,110 cases. Of note, although our findings for coronary artery disease (CAD) did not reach the set significance threshold ($P_{\text{meta}} = 1.1 \times 10^{-5}$, $\text{Effect}_{\text{meta}} = 0.024$), they are consistent with the results presented by Astle *et al* [12] who reported a weak inverse correlation between risk of CAD and MPV, opposite to what is expected (see e.g. [58]). This inverse relationship between risk of CAD and MPV implies that risk of CAD is positively correlated with PLT, which is in agreement with the observations in our study.

6) The association of PLT with MPV is expected as the authors acknowledge. This is long known as the authors cite to several older papers. It is also illuminated by more recent genetic correlation analyses (i.e., Eicher et al. and Chen et al.). It is unclear how/if red cell correlations were examined. It is mentioned in the Discussion as nothing found but then why are red cell traits not listed in Table S3? Two papers are cited as recent – however, the most up to date and critical is Chen et al. 2020 – see their genetic correlation matrices (their Figure S6B) between blood cell traits where they show stronger PLT to WBC correlation and weak PLT to RBC traits.

A: We tested counts of all blood cell types available, including red blood cells (RBC). Results of the PLT PRS association with RBC as well as the cohort size, representing RBC, are presented in the bottom line of **Table 1** in the manuscript. With regards to **Supplementary Table 3**, there are listed only cohorts representing PLT and the ones that are significantly correlated with the platelet count polygenic risk score (see the table legend). As mentioned above and in the manuscript, 5,000 phenotypes, including major human diseases and traits, were included in the analysis. Only the ones associated with the PLT PRS with $p < 1E-05$

(0,05 corrected with the 5,000 phenotypes) underwent further analyses for which data are presented. A clarification has been added (lines 381-382) and sounds as following: See **Supplementary Table 3** for details on cohorts that represent traits associated with the PLT polygenic risk score.

With regards to correlation matrices, stronger correlation of PLT with WBC and weaker between PLT and RBC has been reported before, in the Japanese population in 2018 (PMID: 29403010) and in Europeans in 2019 (PMID: 30858613). Therefore since the study by Chen *et al*, which was just emerging at the time of our submission, is not the first report of this discovery and does not substantially change or challenge these results, their paper was not cited in the original submission but has now been added to the revised version.

7) Table S4 – Inherited platelet disorders – what is your reference? (ISTH Tier 1 Platelet/bleeding genes/UK PanelApp as rather current) To my knowledge SH2B3 does not belong in inherited platelet disorders. CD36 arguably does not as well since most reported mutations are mild with no effect on bleeding diatheses or hemostasis.

A: For the diseases, the source was the Online Mendelian Inheritance in Man database (www.omim.org) in addition to literature search. While it is correct that variants in *SH2B3* do not cause inherited platelet disorders *per se*, a somatic mutation in this gene was found in a patient with essential thrombocythemia (PMID: 20404132) and a germ-line mutation was discovered in a patient with MDS/MPN-ring sideroblasts thrombocytosis (PMID: 31173385). With respect to this notion, the column is now called “Platelet-related disorder”. As for CD36, although the mutations may have mild effects, carriers are reportedly presented with macrothrombocytopenia, that is platelet count and platelet size are affected (PMID: 2316511).

Information on the source was added to the revised version of the manuscript (in the text (lines 154-157) and in the legends of **Supplementary Tables 2, 8, 9** and **16**). The revised text (lines 154-157) sounds as following: Only 38 out of the 577 variants are located in or near genes implicated in platelet disorders (**Supplementary Tables 2** and **8**), with platelet disorders defined based on information from the Online Mendelian Inheritance in Man database (OMIM, www.omim.org) and confirmed through literature search. See also our response to Comment 1 by Reviewer 1.

8) Table S7 for platelet biology and platelet disorders seems somewhat incomplete. What is the source? (as above some sources suggested). Further, for platelet function/aggregation you may require an updated search. For instance the very same GRK5 SNP mentioned above associated here with PLT is not given which has evidence of strength similar to PEAR1.

A: Please refer to our response to Comment 8 by Reviewer 1.

9) Caution is needed in interpreting the PLT PRS as potentially suggesting causal links. Given the numerous loci involved and their complex biology these PRS associations may be driven by non-causal relationships as well as chance and other factors. In particular I find the Hypertension relationship to be one I am skeptical of given the modest effect size and modest association in Icelanders ($p=0.017$). The OR are rather weak in Table S6 by contrast with those of RA/MPN.

A: We absolutely agree with your point regarding causality, and we are not claiming causality either. Our main point is that there is shared genetic basis between PLT and these phenotypes, with no speculations on what is the cause and what is the consequence. While we admit that association with hypertension is modest in Iceland, it should still be considered a significant result, since it replicates the significant association observed in the UK. Weaker ORs may indicate that genetic sharing between PLT and hypertension is less substantial than, for instance, sharing between PLT and MPN.

Minor comments

1) A lot of the later analyses are exploratory data analyses (enrichment, IPA, master regulators). While these can be illuminating there is also some circularity in these types of analyses (i.e., biases of existing databases/literature), and the fact that gene expression levels in specific cell types may drive many of these enrichments, they may be non-specific and categories with a high degree of overlap may not provide granular detail or insights. E.g., the enrichment of spleen associations is not surprising given that the spleen is the primary site of platelet destruction and any spleen or platelet expression dataset will nearly always show a high degree of overlap. At the end of the day I find myself somewhat lost in very large tables of gene lists and there is not a lot of surety about what is truly meaningful (without further experimentation or support). However, as mentioned above and below a few areas may be more interesting (steatosis, cancer, MPN) while others may not be (HTN).

A: It is a fair comment on some circularity in the analyses. However, there is also complementarity that should be acknowledged: DEPICT deals with GWAS data relating them to tissue and gene set enrichment, whereas the core in the IPA analysis are eQTL data that are explored in the context of how differential expression of genes in the dataset could potentially be linked to certain outcomes/functions/pathways.

While enrichment of spleen is not surprising, it can be viewed as a positive control/ proof of principle: it would be surprising if data from GWAS of PLT did not show enrichment with regards to this organ essential to platelet biology.

It is also a fair comment that studies of this type generate a huge amount of data, long gene lists and so on, and often their meaning might be debatable or not obvious. However, such studies can also be viewed as generation of knowledge/information that can be considered for further functional analyses by researchers who work with *in vitro* and/or *in vivo* experimental models. Alternatively, researchers, who work with *in vivo* and/or *in vitro* experimental models, find in these results some support for their observations that comes from studies of human subjects. Throughout the text we do emphasize that further experimental analysis is required to elucidate and/or confirm functions and potential mechanisms.

With regards to areas of special interest, please refer to our responses to Major Comment 4 and Minor Comment 4.

2) Table S5 the note that this is with respect to PLT increasing allele is somewhat buried. The table might be easier to interpret if you also included in the Effects of the PLT trait as well (reinforces the increasing aspect and also allows side by side comparison of effect sizes).

A: A column with the PLT effects has been added to the table (new Supplementary Table 6), as well as a clarification to the table legend that sounds as following: Effects are expressed

in standard deviations and presented with respect to the PLT increasing allele for which the effect is presented in STD (PLT Emeta (in STD) column).

3) In regards to Hepatic Steatosis is interesting to note that some of the predictive models that have been built in the literature include PLT as a covariate. It would be worthwhile to cite these. Likewise, you cannot wholly claim that is a novel link to this paper.

A: It should be noted that in the original version of the manuscript, we did not claim a novel link to hepatic steatosis. Moreover, after adding eQTLs from the platelet and megakaryocyte datasets (Kammers *et al* (PMID: 33094331)) and rerunning the IPA analyses, the Benjamini-Hochberg corrected p-value for hepatic steatosis became 0.062, that is above the set significant threshold of Benjamini-Hochberg corrected p-value ≤ 0.05 .

However, we point out in the original and revised versions of the manuscript that our study demonstrates an association of the PLT PRS with liver enzymes alkaline phosphatase and gamma-glutamyl transpeptidase, not found in previous studies, which had applied a polygenic risk score (cited in the manuscript), and therefore it is one of the first to provide evidence for genetic links between PLT and these liver enzymes.

4) In re: to cancer and platelets beyond MPN, it may be worthwhile to read some of the works of Thomas Wurdinger on platelet transcriptomes as biomarkers for cancer types and consider citing some of that work

A: We did not observe association with cancers (other than MPNs) in the PLT PRS part of the study, and our thoughts on association with MPN are presented in our response to Major Point 4. Notably, we were not looking at platelet transcriptome *per se*, but rather at gene expression from the PLT-associated loci in various tissues, defined as significantly enriched by DEPICT analyses, where only 26 of 235 eQTL genes were found to be restricted to platelets and/or megakaryocytes. Therefore, the purpose of the suggested literature review in the context of our manuscript is not very clear.

5) There are some attempts being made to capture published Polygenic Risk Scores to make them more transparent and reproducible due to known issues transferring these PRS across populations and ongoing research in that area. I suggest the authors consider if accepted depositing their PRS in such a resource (or making a clear Supplemental table of the PRS included and PRS calculations)

A: The PRS will be deposited if the manuscript is accepted.

Reviewer #3 (Remarks to the Author):

Review: Platelet count: Sequence variants, its genetic associations with other traits and potential regulatory mechanisms

This manuscript by Mikaelsdottir et al. presents the results of a large-scale meta-analysis of platelet count including summary data from GWAS of more than half a million European people. After identifying a large number of independent variants (4 are claimed as novel), the authors characterized potential mechanisms of action via coding variant and cis- expression quantitative trait locus (cis-eQTL) analyses and Data- driven Expression Prioritized Integration for Complex Traits (DEPICT). In addition they employed the Ingenuity Pathway Analysis (IPA) to look for enrichment of effects by pathway. Finally, they compared the predictive value of a polygenic risk score for platelet count among an array of additional available phenotypes. Overall, this paper represents a well-powered analysis coupled with hypothesis-generating downstream functional analyses that are likely to be of interest to the field.

Major comments:

1. This article needs a “table 1” summarizing the demographic information and platelet related measurements in each dataset for all included individuals.

A: These data are presented in **Supplementary Table 3**.

2. The authors do not mention inclusion and exclusion criteria of subjects. Platelet counts could be greatly affected by variety of disease conditions and medications.

A: Please refer to our response to Comment 1 by Reviewer 1.

3. The effect of relatedness in the association tests from the genotypically imputed 1st and 2nd degree Icelanders should be explored more- it would be helpful to have one more column in supplemental table 2 to list the p-value from genotyped-Icelander-only, as a sensitivity analysis.

A: We have calculated the effect estimate and p-values for the Icelandic analysis restricted to genotyped individuals (Iceland Genotyped only). The comparison with results, which include both genotyped and familiarly imputed Icelanders (Iceland all), are presented in **the table below and can be seen in the plot** of effects for the 577 PLT variants following the table. We do not observe any bias in the effect estimates, weighted linear regression between the effect estimates with and without the ungenotyped individualst included gives a slope of 1.01. The changes in the p-values are not big either, on average the Z-score statistic only increases by 3% by including the ungenotyped individuals, probably because they are close relatives of genotyped individuals that have PLT measurements. If considered necessary by the Reviewer, the plot and/or the table can be added as a supplementary material.

PLT association results for all Icelanders vs. only genotyped. Effects are expressed in $N \times 10^3/\mu\text{l}$.

rsName	Chr	Pos	EA/OA	Iceland All		Iceland Genotyped only	
				Effect	Pval	Effect	Pval
rs263527	chr1	2241549	C/T	-2.07	2.3E-09	-2.09	8.5E-09
rs1417986	chr1	7737674	T/A	-1.61	5.5E-06	-1.55	3.1E-05
rs10864368	chr1	8858254	C/T	1.13	0.001	1.19	0.001
rs11121529	chr1	10211630	G/C	-1.98	2.5E-04	-2.12	1.8E-04
rs2236055	chr1	11982204	G/A	-2.29	3.2E-11	-2.23	6.4E-10
rs66530629	chr1	24705677	A/G	-1.10	0.007	-1.03	0.015
rs9438901	chr1	25258485	G/A	0.97	0.046	0.96	0.058
rs909832	chr1	25427534	G/C	-2.45	3.7E-12	-2.51	9.7E-12
rs56214942	chr1	25565468	A/G	1.32	1.4E-04	1.28	4.2E-04
rs182050989	chr1	26936054	T/C	2.81	0.011	3.31	0.004
rs2236074	chr1	27884873	A/G	-1.19	9.8E-04	-1.27	7.6E-04
rs157198	chr1	28877395	C/T	-1.20	0.025	-1.32	0.019
rs34101571	chr1	30739047	A/G	5.17	0.002	5.97	5.6E-04
rs35343437	chr1	36307804	T/G	1.04	0.003	0.95	0.009
rs7529794	chr1	39937698	T/G	1.88	8.6E-07	1.70	2.2E-05
rs12731981	chr1	43338669	A/G	3.75	0.006	4.68	0.001
rs75139539	chr1	43375311	T/C	2.99	0.002	3.69	2.6E-04
rs17853159	chr1	45345193	A/G	1.79	0.006	1.72	0.011
rs11211124	chr1	45496737	C/T	-3.00	5.2E-12	-3.08	1.3E-11
rs10789481	chr1	45917911	G/C	1.52	9.3E-06	1.51	2.8E-05
rs140436199	chr1	87636754	A/G	-2.42	6.9E-05	-2.41	1.6E-04
rs79898419	chr1	91122004	G/A	1.12	0.009	0.86	0.056
rs17501512	chr1	91546924	G/C	-0.92	0.008	-1.01	0.005
rs945631	chr1	92960610	A/G	-2.60	1.6E-04	-2.22	0.002
rs547866	chr1	93259496	T/C	-0.95	0.007	-1.02	0.006
rs4477285	chr1	94411372	G/A	-1.36	8.0E-05	-1.49	3.5E-05
rs333947	chr1	109928142	A/G	1.15	0.018	1.58	0.002
rs2999157	chr1	112690571	G/A	1.83	1.1E-07	2.04	1.7E-08
rs3767812	chr1	117612998	A/G	2.09	3.4E-06	2.15	5.2E-06
rs61819435	chr1	150539152	G/C	2.06	1.4E-08	2.45	1.3E-10
rs4521985	chr1	154151047	A/G	1.58	1.4E-05	1.72	6.3E-06
rs78261031	chr1	156142679	A/G	-2.02	0.016	-1.88	0.033
rs1342442	chr1	156496907	G/A	1.88	1.3E-07	1.91	3.2E-07
rs12041331	chr1	156899922	A/G	2.35	8.7E-04	2.15	0.004
rs6425522	chr1	171976942	T/C	3.26	1.7E-16	3.23	5.9E-15
rs10174	chr1	185297982	G/A	-0.76	0.031	-0.82	0.026
rs1124025	chr1	199010689	C/G	-1.69	2.6E-05	-1.88	8.8E-06
rs5780225, rs751257259, rs763749135, rs796130745 rs2802813	chr1	204292903	CTGTTAG/C	-1.21	0.001	-1.22	0.002
	chr1	204936906	C/T	0.90	0.046	0.93	0.048
rs1172155	chr1	205251113	T/C	-1.92	9.7E-04	-2.12	5.1E-04
rs1668871	chr1	205268009	C/T	1.93	7.0E-08	1.93	2.5E-07
rs1891058	chr1	213764039	G/A	0.97	0.007	0.97	0.010
rs6696074	chr1	225774988	C/T	-1.56	6.4E-06	-1.55	1.9E-05
rs138994074, rs76006554	chr1	226351602	CA/C	-1.60	5.4E-04	-1.98	4.8E-05
rs4846914	chr1	230159944	G/A	-0.78	0.027	-0.85	0.022
rs2758994	chr1	236540467	I/T	1.16	0.003	1.34	0.001
rs12239046	chr1	247438293	T/C	-0.77	0.030	-0.73	0.049
rs41315846	chr1	247549001	C/T	3.68	6.4E-26	3.79	5.5E-25
rs56043070	chr1	247556467	A/G	-7.54	1.5E-27	-7.83	4.4E-27

rs1339847	chr1	247875992	A/G	2.32	4.0E-05	2.28	1.2E-04
rs3811444	chr1	247876149	T/C	-2.70	1.4E-13	-2.90	3.3E-14
rs35330522	chr2	12736934	A/G	0.88	0.011	1.03	0.004
rs72781667	chr2	24002797	G/A	-1.25	0.012	-1.38	0.008
rs1260326	chr2	27508073	T/C	2.43	2.6E-11	2.57	1.4E-11
rs655029	chr2	31254972	G/A	-3.67	9.3E-22	-3.73	9.8E-21
rs6544047	chr2	36921087	A/T	1.19	6.8E-04	1.20	0.001
rs7563723	chr2	37819920	C/T	-1.35	2.4E-04	-1.49	1.0E-04
rs10181075	chr2	42915859	G/T	-0.72	0.044	-0.82	0.027
rs1430083	chr2	43221340	T/A	1.49	0.003	1.42	0.007
rs149290349	chr2	43224818	A/G	-4.14	1.2E-07	-3.95	1.4E-06
rs4952782	chr2	45846957	T/A	-0.96	0.011	-1.03	0.009
rs10168349	chr2	46133768	C/G	1.00	0.005	0.96	0.010
rs6545465	chr2	54985201	T/C	1.33	0.004	1.52	0.002
rs58966835	chr2	66423642	C/T	-4.45	0.029	-4.26	0.047
rs939136	chr2	66450399	G/C	-1.33	3.7E-04	-1.52	9.7E-05
rs10048745	chr2	68735005	A/G	2.12	2.3E-07	2.18	3.4E-07
rs3771529	chr2	69937708	A/G	-2.96	2.6E-12	-2.85	1.2E-10
rs2301984	chr2	74493928	G/A	1.26	0.012	1.50	0.004
rs11682055	chr2	85411200	G/A	-0.79	0.023	-0.79	0.030
rs147348486	chr2	85539177	A/G	2.95	0.050	3.21	0.042
rs2118446	chr2	108529352	T/C	1.74	9.5E-07	1.87	5.5E-07
rs76496105	chr2	109690090	G/C	-1.98	0.002	-2.34	5.2E-04
rs7562480	chr2	111177655	C/T	0.76	0.027	0.72	0.045
rs1045267	chr2	111429464	A/G	2.31	5.6E-09	2.38	9.5E-09
rs6734238	chr2	113083453	G/A	1.74	4.4E-07	2.00	3.1E-08
rs4849842	chr2	120255090	T/A	1.73	4.7E-06	1.62	4.1E-05
rs145365565	chr2	159503699	A/C	-2.67	1.2E-07	-2.54	1.4E-06
rs76774368	chr2	159748003	T/C	5.61	1.1E-05	5.93	8.4E-06
rs12052715	chr2	159820864	C/G	2.87	1.1E-14	3.01	1.0E-14
rs114821641	chr2	159858447	T/C	11.88	4.1E-06	10.55	8.9E-05
rs7585866	chr2	191831529	G/A	0.81	0.028	0.91	0.018
rs979020	chr2	197305025	C/T	-1.07	0.002	-1.08	0.003
rs7560328	chr2	201300114	A/C	-1.33	1.8E-04	-1.30	4.6E-04
rs72932729	chr2	202785544	C/T	-1.09	0.001	-1.00	0.005
rs1047891	chr2	210675783	A/C	-1.21	0.002	-1.60	9.1E-05
rs17572109	chr2	218229211	A/G	1.63	2.1E-05	1.65	3.9E-05
rs115504855	chr2	218229854	G/A	-2.00	0.040	-2.24	0.029
rs10207991	chr2	224898426	T/C	-3.66	2.0E-08	-3.46	4.3E-07
rs116778355	chr2	224951925	T/A	-3.35	0.012	-2.95	0.036
rs11676298	chr2	226427015	G/C	2.90	5.1E-12	2.89	5.5E-11
rs55664157	chr2	233366008	A/T	-2.36	2.7E-06	-2.27	1.7E-05
rs114266592	chr2	234981357	C/T	-2.36	0.039	-2.29	0.056
rs78909033	chr2	240571486	A/G	2.30	1.5E-06	2.42	1.5E-06
rs9810259	chr3	12226691	G/C	-1.84	2.4E-07	-1.86	6.8E-07
rs7618405	chr3	18209017	A/C	-2.74	4.5E-11	-2.98	8.3E-12
rs62240975	chr3	18444681	A/G	-1.38	3.6E-04	-1.35	8.8E-04
rs1388786	chr3	27330731	G/A	-2.20	3.4E-09	-2.10	7.9E-08
rs2371108	chr3	27715527	T/G	1.07	0.003	1.29	5.2E-04
rs13084317	chr3	39146367	A/G	1.20	9.1E-04	1.34	4.3E-04
rs397949584, rs57383920	chr3	47207585	TA/T	-1.53	7.1E-04	-1.66	4.8E-04
rs71617297	chr3	56690065	G/T	2.25	0.011	1.97	0.033
rs2046823	chr3	56744983	A/G	1.56	6.4E-05	1.53	1.8E-04
rs17288922	chr3	56817359	A/G	-4.82	3.1E-27	-5.00	1.0E-26

rs17288936	chr3	56822572	T/C	-5.83	1.7E-25	-5.77	6.0E-23
rs75665326	chr3	56831053	A/T	-3.07	8.2E-08	-3.11	2.2E-07
rs17825630	chr3	56916027	A/G	2.79	4.4E-09	2.85	1.1E-08
rs6786088	chr3	58329348	C/T	-1.84	3.9E-07	-2.00	1.6E-07
rs7611020	chr3	69807676	G/A	1.44	5.0E-05	1.49	6.1E-05
rs9809116	chr3	72348128	G/A	1.52	1.8E-05	1.50	5.3E-05
rs7610102	chr3	101527959	A/G	-0.74	0.031	-0.91	0.012
rs167924	chr3	107660990	A/G	-0.73	0.042	-0.75	0.044
rs1143671	chr3	121928439	T/C	0.87	0.012	0.74	0.042
rs3804749	chr3	123114156	I/T	2.43	2.2E-11	2.70	1.5E-12
rs17295246	chr3	123386874	A/G	-1.48	2.8E-04	-1.58	2.0E-04
rs2165252	chr3	124626515	I/C	0.89	0.015	0.67	0.080
rs73199529	chr3	124837573	T/G	3.85	0.004	3.53	0.011
rs12629965	chr3	128703815	G/A	-0.86	0.031	-1.00	0.017
rs557662044	chr3	129419788	C/T	-28.47	2.2E-12	-29.10	1.0E-11
rs1347209	chr3	136907855	G/T	-0.88	0.011	-1.07	0.003
rs900400	chr3	157080986	C/T	0.82	0.025	1.03	0.007
rs77232317	chr3	167747119	A/G	0.78	0.044	0.91	0.026
rs73167972	chr3	169148751	G/A	1.74	0.013	1.82	0.014
rs9860749	chr3	179013232	G/A	2.09	7.0E-08	2.20	6.2E-08
rs10937159	chr3	184034513	A/C	1.12	0.001	1.22	7.8E-04
rs562344339	chr3	184060596	C/T	-17.19	4.3E-07	-16.58	2.6E-06
rs952982	chr3	184371784	T/G	7.28	3.1E-10	7.83	1.0E-10
rs78565404	chr3	184372454	T/C	10.13	1.0E-44	10.06	1.6E-40
rs6141	chr3	184372478	C/T	-3.80	5.1E-28	-3.88	8.4E-27
rs572476245	chr3	184373314	G/T	-18.42	6.5E-35	-19.09	5.5E-34
rs376419231,rs55827759	chr3	184373595	TGGAA/T	-3.68	2.6E-06	-4.08	6.3E-07
rs34623301	chr3	184375252	A/G	4.72	1.1E-28	4.80	3.7E-27
rs147646497	chr3	184387795	A/G	6.53	3.6E-09	6.20	8.7E-08
rs11705701	chr3	185826521	A/G	-1.67	1.7E-06	-1.62	1.0E-05
rs9829114	chr3	196791752	A/G	-1.15	0.001	-1.35	2.7E-04
rs35537543	chr4	3218574	G/A	-10.95	5.1E-04	-11.99	3.1E-04
rs13108218	chr4	3442204	A/G	1.77	8.2E-07	1.57	3.0E-05
rs11734099	chr4	6889708	A/G	2.20	5.7E-07	2.49	6.0E-08
rs62291089	chr4	6908608	G/C	-1.19	0.005	-0.69	0.122
rs6815294	chr4	7040622	G/A	-1.45	2.8E-05	-1.54	2.1E-05
rs2315552	chr4	17625671	A/G	0.76	0.035	0.66	0.080
rs138541467	chr4	38652151	TA/T	-0.77	0.025	-0.91	0.011
rs58408429	chr4	56903658	C/T	-1.91	1.4E-05	-1.85	5.4E-05
rs11735092	chr4	87305079	C/T	0.94	0.007	0.86	0.019
rs6532796	chr4	99121091	A/G	-1.01	0.008	-1.01	0.011
rs4699156	chr4	105114278	A/C	-0.92	0.017	-0.98	0.016
rs113693454	chr4	105286606	A/T	-2.41	0.042	-2.47	0.045
rs6533483	chr4	109966833	A/G	-2.25	2.1E-10	-2.19	3.0E-09
rs58583086	chr4	119635207	A/G	-1.53	1.7E-05	-1.41	1.4E-04
rs17755079	chr4	123749526	T/I	-1.62	0.002	-1.31	0.019
rs60179902	chr4	140915737	C/A	-1.34	9.4E-05	-1.13	0.002
rs1512281	chr4	144513749	G/A	-1.57	5.6E-06	-1.40	1.1E-04
rs1429139	chr4	147362952	T/A	2.04	6.0E-07	1.90	8.5E-06
rs28666858	chr4	151495230	T/C	-1.63	2.5E-06	-1.51	3.2E-05
rs112969588	chr4	156770941	C/T	-1.53	0.008	-1.40	0.021
rs7705526	chr5	1285859	A/C	2.72	7.4E-14	2.96	7.5E-15
rs2853677	chr5	1287079	G/A	2.03	7.5E-09	2.22	1.6E-09
rs35179196	chr5	62235390	A/AT	0.95	0.007	0.96	0.009

rs10940080	chr5	66704896	G/A	1.24	4.4E-04	1.34	2.8E-04
rs4976137	chr5	67925107	A/G	1.10	0.002	1.04	0.006
rs34651	chr5	72848178	C/T	-1.33	0.041	-1.11	0.104
rs2307111	chr5	75707853	C/T	0.80	0.025	0.91	0.016
rs7706078	chr5	76444374	T/C	-0.96	0.006	-0.96	0.008
rs34968964	chr5	76665143	C/G	-5.81	0.023	-6.53	0.016
rs34592828	chr5	76701084	A/G	-7.47	7.6E-15	-7.03	2.4E-12
rs353937	chr5	77812245	A/T	-1.16	7.9E-04	-1.12	0.002
rs56363865	chr5	78489919	C/T	-2.62	1.8E-11	-2.46	1.5E-09
rs565934142	chr5	88776849	A/G	13.36	0.004	12.20	0.011
rs114694170	chr5	88884379	C/T	8.72	6.1E-45	8.92	4.7E-43
rs1158464	chr5	89042972	A/G	1.95	1.1E-06	2.16	2.7E-07
rs10074585	chr5	90959181	G/T	1.27	0.002	1.05	0.013
rs57486725	chr5	111725319	G/T	-3.41	1.2E-06	-3.49	2.0E-06
rs1047440	chr5	123346140	T/C	1.06	0.002	0.97	0.008
rs11950562	chr5	132316836	C/A	-2.00	7.0E-09	-2.02	2.5E-08
rs2070722	chr5	132488794	C/A	2.00	3.3E-08	2.04	7.7E-08
rs329117	chr5	134524410	T/C	-0.74	0.036	-0.74	0.047
rs589153	chr5	139677963	G/A	-1.07	0.004	-0.95	0.015
rs449454	chr5	142153497	A/G	-2.22	3.3E-10	-2.57	3.8E-12
rs4704727	chr5	156953056	T/G	0.89	0.016	1.11	0.004
rs6863275	chr5	158769734	C/A	-1.94	0.002	-2.10	0.002
rs6556405	chr5	159208094	C/T	-1.40	6.6E-04	-1.42	9.4E-04
rs4538653	chr5	160168887	G/C	1.86	1.1E-06	2.11	1.3E-07
rs6861931	chr5	160208949	T/C	-1.63	0.004	-1.70	0.004
rs10587485,rs58403685	chr5	177310990	C/CAG	3.30	6.4E-04	2.75	0.006
rs11960863	chr5	178187400	G/C	-2.15	2.8E-05	-1.96	2.5E-04
rs34164888	chr6	25521693	A/C	2.08	4.3E-07	1.98	4.1E-06
rs214053	chr6	25527735	C/T	-1.92	2.9E-08	-1.81	6.0E-07
rs129128	chr6	26125114	C/T	-1.38	0.010	-1.52	0.007
rs9404952	chr6	29836388	A/G	1.55	6.7E-06	1.59	9.9E-06
rs12665339	chr6	30633455	G/A	2.73	2.4E-10	2.93	7.3E-11
rs2442730	chr6	31351380	C/A	2.22	1.4E-09	2.25	4.5E-09
rs1050538	chr6	31356822	G/T	2.98	1.3E-10	3.24	2.4E-11
rs9266658	chr6	31379867	A/G	3.68	8.6E-19	3.87	4.6E-19
rs11575845	chr6	31724609	G/C	-4.34	1.2E-05	-4.70	5.1E-06
rs9296095	chr6	33574746	C/T	6.07	3.8E-45	5.81	3.8E-38
rs210143	chr6	33579153	T/C	-5.66	5.7E-50	-5.68	4.1E-46
rs10807137	chr6	34215249	C/T	-1.18	0.011	-1.00	0.038
rs2814983	chr6	34623430	G/A	-1.70	0.001	-1.67	0.003
rs76976387	chr6	36553586	A/G	7.61	1.8E-10	8.11	7.9E-11
rs9394951	chr6	43383015	C/T	-1.42	4.3E-05	-1.53	2.6E-05
rs186369938	chr6	43770246	C/G	3.47	2.6E-05	3.36	1.0E-04
rs11755026	chr6	47352650	A/G	-1.42	0.015	-1.25	0.042
rs7772202	chr6	47653843	A/G	1.71	1.9E-05	1.74	3.2E-05
rs614570	chr6	52454453	T/C	1.00	0.004	0.75	0.038
rs6423287	chr6	52798089	G/T	1.27	3.0E-04	1.30	4.2E-04
rs112273618,rs34208219	chr6	109296454	A/AAAG	-1.46	2.4E-05	-1.57	1.5E-05
rs9374170	chr6	110391788	G/A	0.74	0.032	0.91	0.012
rs35548455	chr6	116399351	T/C	1.63	0.010	1.68	0.011
rs71686340,rs71698790	chr6	134725909	T/TAAAGG	1.92	6.1E-06	1.74	9.6E-05
rs6916729	chr6	134887604	T/A	-4.95	3.3E-12	-4.80	1.2E-10
rs2210366	chr6	135094070	A/G	-4.62	1.8E-31	-4.86	1.0E-31
rs6930223	chr6	135103065	G/T	-2.30	2.4E-11	-2.30	1.9E-10

rs9483788	chr6	135114363	C/T	6.50	1.2E-61	6.67	2.0E-59
rs723388	chr6	135207584	G/A	-2.70	3.1E-14	-2.75	1.3E-13
rs381500	chr6	164057356	A/C	2.12	6.5E-10	2.32	9.5E-11
rs35069969	chr7	2479055	C/G	1.17	0.001	1.12	0.003
rs2260230	chr7	2783352	A/I	1.33	5.5E-04	1.35	8.0E-04
rs7789916	chr7	13987932	G/A	0.89	0.017	0.96	0.014
rs67956034	chr7	18189259	T/C	-1.55	1.5E-04	-1.76	4.1E-05
rs2710804	chr7	36044919	C/T	0.93	0.009	0.97	0.009
rs13236163	chr7	37403010	C/T	-1.22	0.008	-1.21	0.013
rs11762008	chr7	44864065	G/A	-2.00	8.2E-09	-2.21	1.0E-09
rs3735485	chr7	44969742	A/G	-1.25	0.009	-1.24	0.013
rs6592965	chr7	50360284	A/G	1.37	9.1E-05	1.35	2.3E-04
rs4947490	chr7	55092845	A/G	-1.01	0.007	-1.03	0.009
rs13247874	chr7	73596112	T/C	-1.06	0.014	-0.77	0.086
rs8565	chr7	76000956	T/C	0.91	0.023	0.82	0.049
rs11764390	chr7	80586889	A/G	0.79	0.022	0.74	0.041
rs35574453, rs397976140, rs566083700, rs4385401	chr7	80605492	GA/G	2.05	0.005	1.80	0.019
rs62482241	chr7	100501843	C/T	2.62	3.7E-09	2.75	3.2E-09
rs342293	chr7	106731773	G/C	-3.57	6.0E-25	-3.77	1.9E-25
rs12706108	chr7	116872215	C/T	-1.37	6.8E-04	-1.49	3.9E-04
rs689341	chr7	123763920	A/C	-1.05	0.004	-1.21	0.002
rs79947009	chr7	123776519	G/T	4.26	4.1E-10	4.60	1.0E-10
rs140553648	chr7	129612295	CGCGGGCG G/C	2.35	0.002	2.41	0.002
rs11556924	chr7	130023656	T/C	1.51	3.3E-05	1.35	3.8E-04
rs62471615	chr7	131062196	C/A	1.41	2.7E-04	1.26	0.002
rs7806221	chr7	135634031	C/T	-1.38	1.2E-04	-1.50	6.3E-05
rs73164936	chr7	135983054	G/A	2.11	0.001	1.98	0.003
rs75511207	chr7	140181500	C/T	-2.48	2.3E-05	-2.71	1.0E-05
rs7808461	chr7	152684513	T/G	0.99	0.004	0.95	0.008
rs1153998	chr7	158827968	C/T	2.32	0.009	1.58	0.087
rs36056437	chr8	8935355	T/G	-1.19	5.0E-04	-1.21	7.3E-04
rs200119611	chr8	21995826	GT/G	-2.52	0.039	-2.63	0.039
rs11779638	chr8	22582695	C/A	1.88	1.9E-05	2.04	9.0E-06
rs6985703	chr8	55872574	T/G	-0.68	0.048	-0.64	0.075
rs61709988	chr8	65976651	C/T	-1.93	9.4E-08	-2.10	2.7E-08
rs6993770	chr8	105569300	T/A	-3.94	1.3E-21	-3.96	4.1E-20
rs28455756	chr8	124881909	T/C	0.73	0.045	0.66	0.086
rs7010394	chr8	125331278	T/C	-1.02	0.003	-0.93	0.011
rs62523770	chr8	129646924	T/G	1.53	3.7E-04	1.35	0.003
rs1158570	chr8	130318827	T/C	-0.70	0.041	-0.63	0.080
rs35858667	chr8	143920720	A/G	2.36	0.002	2.39	0.003
rs11993233	chr8	143928115	G/A	2.47	3.7E-12	2.51	1.8E-11
rs139204327, rs3831137, rs71312798, rs9406914	chr9	273145	TTG/T	-1.11	0.005	-0.80	0.052
rs10970979	chr9	277776	T/C	1.03	0.020	1.10	0.018
rs10970979	chr9	334337	G/A	-1.18	0.002	-0.88	0.027
rs34881325	chr9	2622134	T/C	1.05	0.003	0.92	0.013
rs62540578	chr9	4741387	G/C	-1.96	2.4E-08	-2.26	8.2E-10
rs118088097	chr9	4762085	A/G	-6.85	2.1E-43	-6.81	3.0E-39
rs10974771	chr9	4762737	G/C	-3.81	6.0E-20	-4.06	1.6E-20
rs409950	chr9	4763368	A/C	3.73	1.3E-17	3.71	4.6E-16
rs10815071	chr9	4763484	G/A	-3.94	4.3E-30	-4.10	1.2E-29
rs12005199	chr9	4763491	A/G	7.70	5.8E-100	7.95	2.2E-97
rs368418	chr9	4768336	A/G	-3.25	3.4E-12	-3.29	1.9E-11

rs117857686	chr9	4807471	T/C	4.29	0.001	3.93	0.005
rs79075814	chr9	4808971	T/C	-5.52	4.0E-25	-5.72	1.3E-24
rs10974808	chr9	4840380	G/A	8.30	6.7E-77	8.44	1.6E-72
rs117935834	chr9	4887853	G/A	5.12	9.1E-05	5.15	1.7E-04
rs12338005	chr9	4898136	A/G	-1.54	6.6E-05	-1.38	6.5E-04
rs2381194	chr9	4959331	A/G	-2.28	8.8E-06	-2.27	2.5E-05
rs10815146	chr9	5015901	T/A	1.69	1.0E-06	1.67	3.9E-06
rs12004239	chr9	5057822	A/C	9.49	8.1E-07	10.21	3.9E-07
rs191064596	chr9	5271995	T/C	11.53	5.5E-06	11.62	1.1E-05
rs3731211	chr9	21986848	T/A	-2.60	1.4E-11	-2.54	2.8E-10
rs10757287	chr9	22143571	T/A	-2.46	8.1E-06	-2.56	9.0E-06
rs741917	chr9	35707057	T/C	-1.12	0.005	-1.18	0.005
rs10973700	chr9	38196120	G/C	1.26	2.6E-04	1.41	1.1E-04
rs12000252	chr9	70423047	G/A	-1.20	5.1E-04	-1.21	8.2E-04
rs142550358	chr9	88777771	T/TTC	-4.81	2.8E-12	-5.13	1.1E-12
rs772407	chr9	90798568	C/A	1.40	4.8E-05	1.24	5.7E-04
rs112679102	chr9	91167740	T/C	-1.74	0.008	-1.68	0.014
rs1853427	chr9	92806161	C/T	0.93	0.026	0.88	0.043
rs10990535	chr9	96328727	T/C	2.82	6.9E-12	2.81	7.0E-11
rs10820727	chr9	96495705	A/G	-1.57	0.001	-1.36	0.008
rs114266868	chr9	97951319	C/T	1.67	3.7E-06	1.54	4.2E-05
rs10817007	chr9	110392967	G/T	2.16	2.2E-04	2.28	2.0E-04
rs10116799	chr9	113422027	T/C	1.70	0.001	1.80	9.3E-04
rs2900177	chr9	120735367	C/T	-1.67	7.5E-06	-1.69	1.5E-05
rs10986338	chr9	124428953	G/A	0.83	0.022	0.71	0.061
rs13289095	chr9	128704210	T/G	-2.26	3.1E-06	-2.28	7.6E-06
rs60757417	chr9	132989049	G/C	-4.39	4.0E-11	-4.52	8.5E-11
rs150813342	chr9	132989126	T/C	-27.53	7.0E-91	-27.28	5.9E-82
rs13284142	chr9	133001148	G/C	2.24	9.2E-11	2.03	2.3E-08
rs8176746	chr9	133255935	T/G	-1.45	0.038	-1.77	0.016
rs2520099	chr9	134053017	C/T	1.62	2.7E-05	1.66	4.1E-05
rs7912035	chr10	3665123	T/C	-2.08	2.2E-05	-2.32	5.6E-06
rs34346558	chr10	11862835	C/A	1.96	7.6E-06	1.78	1.0E-04
rs943188	chr10	14542302	T/C	1.08	0.004	1.34	5.7E-04
rs802171	chr10	16814686	A/G	1.15	7.9E-04	1.07	0.003
rs7085742	chr10	17206829	C/G	1.38	5.6E-05	1.30	3.0E-04
rs11014291	chr10	24909758	C/T	-1.45	5.3E-05	-1.50	6.3E-05
rs866919	chr10	30224354	C/T	-1.79	3.4E-07	-1.77	1.3E-06
rs61848370	chr10	49069530	T/G	-1.80	4.7E-06	-1.92	2.8E-06
rs10740059	chr10	62049768	A/G	0.72	0.041	0.74	0.044
rs224033	chr10	62762354	C/G	-0.69	0.048	-0.64	0.078
rs10761731, rs71476393	chr10	63267850	T/A	3.87	5.2E-29	4.00	2.0E-28
rs186167801	chr10	63453066	T/G	-6.17	0.002	-5.15	0.014
rs10762859	chr10	79343614	A/T	0.80	0.021	1.16	0.001
rs116052829	chr10	79404390	T/C	2.13	3.9E-04	1.86	0.003
rs11202654	chr10	88098060	G/A	1.30	0.013	1.39	0.011
rs2068888	chr10	93079885	A/G	-1.66	1.7E-06	-1.91	1.4E-07
rs11190127	chr10	99512225	A/C	-0.99	0.006	-1.17	0.002
rs71016384	chr10	102469616	AG/A	2.28	9.5E-07	2.81	7.9E-09
rs911547	chr10	103879663	G/A	2.08	2.0E-05	2.14	2.7E-05
rs10886430	chr10	119250744	G/A	-1.20	0.021	-0.91	0.093
rs11604127	chr11	196944	T/C	5.27	7.4E-40	5.31	3.0E-37
rs9704108	chr11	308065	T/C	-2.85	2.3E-06	-2.82	7.3E-06
rs12806645	chr11	5677225	A/G	0.94	0.009	0.72	0.056

rs10769966	chr11	8825494	A/C	-1.61	2.7E-06	-1.74	1.3E-06
rs2645029	chr11	9790689	G/A	1.38	0.009	1.48	0.008
rs10840453	chr11	10643354	T/C	-1.06	0.006	-0.99	0.014
rs602126	chr11	32889926	C/T	1.50	5.0E-04	1.55	6.1E-04
rs35741412	chr11	47410213	A/G	3.25	3.8E-04	3.29	5.9E-04
rs3817334	chr11	47629441	T/C	-1.81	1.9E-07	-1.77	1.2E-06
rs10750866	chr11	57637306	G/A	2.60	9.5E-11	2.49	2.9E-09
rs174560	chr11	61814292	C/T	3.10	6.6E-18	3.17	3.3E-17
rs477895	chr11	64281440	C/T	-2.08	2.3E-05	-2.04	7.0E-05
rs11227261	chr11	65695372	G/C	-1.04	0.004	-1.02	0.006
rs667555	chr11	65857091	A/C	-1.90	2.5E-07	-2.04	1.1E-07
rs11602052	chr11	69197107	C/G	1.87	1.6E-07	1.72	4.2E-06
rs6592656	chr11	76657010	G/C	0.98	0.005	0.87	0.016
rs585721	chr11	77842592	C/T	1.15	0.001	1.18	0.001
rs556562	chr11	85955397	A/C	-1.02	0.027	-1.25	0.009
rs646809	chr11	95101707	C/T	-1.34	0.001	-1.32	0.003
rs7934719	chr11	108471137	T/C	1.62	6.4E-06	1.81	1.6E-06
rs73000929	chr11	114082900	A/G	-4.08	3.9E-07	-4.37	2.1E-07
rs73000965	chr11	114111599	A/T	-1.05	0.004	-0.99	0.009
rs964184	chr11	116778201	G/C	-1.63	0.002	-1.72	0.002
rs77953286	chr11	117043946	C/G	-3.51	4.4E-05	-3.45	1.2E-04
rs4938637	chr11	119204613	A/G	5.72	4.3E-13	5.86	1.3E-12
rs2155380	chr11	119209327	G/A	3.40	3.0E-19	3.45	2.6E-18
rs35929108	chr11	119315038	A/G	3.16	1.5E-06	3.50	3.8E-07
rs6589810	chr11	120363199	A/G	1.25	2.8E-04	1.14	0.002
rs4570592	chr11	126417890	G/A	1.10	0.001	1.38	1.3E-04
rs7925737	chr11	128164127	T/C	1.92	1.4E-04	1.76	8.7E-04
rs4937333	chr11	128460625	T/C	1.69	9.2E-07	1.60	9.8E-06
rs11221442	chr11	128707729	C/G	-1.20	0.002	-1.32	9.2E-04
rs2268607	chr11	128797533	C/T	-1.94	2.9E-04	-1.68	0.003
rs10774375	chr12	614736	A/G	-0.95	0.007	-1.17	0.002
rs34038797	chr12	630843	G/C	-1.48	2.2E-05	-1.82	6.5E-07
rs216311	chr12	6019277	T/C	-1.20	6.6E-04	-1.40	1.5E-04
rs7306706	chr12	6106468	G/A	0.80	0.022	0.81	0.025
rs10849413	chr12	6176701	A/G	2.19	2.1E-10	2.36	7.0E-11
rs887477	chr12	6336816	A/C	1.31	1.4E-04	1.61	8.0E-06
rs28999107	chr12	6383934	T/G	-1.82	2.0E-07	-1.91	2.0E-07
rs2364482	chr12	6392965	G/T	1.83	6.9E-05	1.95	4.9E-05
rs12820720	chr12	8000056	T/C	-0.76	0.032	-0.72	0.054
rs3093733, rs72525724	chr12	12719550	CA/C	1.01	0.011	1.20	0.004
rs76639840	chr12	22457358	C/G	-7.61	4.4E-06	-7.52	1.5E-05
rs35024086, rs398098094	chr12	29282572	TA/T	-2.72	3.3E-15	-2.57	1.2E-12
rs7960662	chr12	40085265	A/G	-1.22	5.4E-04	-1.16	0.002
rs17442910	chr12	40156900	C/T	-10.03	0.038	-11.65	0.022
rs146762874	chr12	46542717	C/G	2.93	7.9E-06	3.10	6.7E-06
rs11168249	chr12	47814585	T/C	-2.22	1.4E-10	-2.25	5.6E-10
rs73109811	chr12	47818936	T/C	3.12	1.3E-15	3.18	7.4E-15
rs113736796	chr12	47819937	G/C	3.23	8.7E-05	3.50	4.7E-05
rs113543437	chr12	49258948	A/G	-2.43	1.3E-04	-2.33	4.5E-04
rs35082997, rs397713263	chr12	50506605	A/AGT	-2.25	4.8E-10	-2.15	1.2E-08
rs7133314	chr12	51270728	T/C	2.53	3.3E-06	2.70	2.0E-06
rs79977579	chr12	54300776	A/C	4.66	2.3E-14	4.86	2.9E-14
rs10876550	chr12	54318524	G/A	-3.36	5.0E-22	-3.34	6.5E-20
rs10783794	chr12	56593395	G/A	1.75	8.9E-07	1.81	1.2E-06

rs7138821	chr12	56796206	C/A	2.04	4.0E-08	2.07	9.6E-08
rs1716505	chr12	64611299	G/C	2.89	7.3E-15	2.88	1.2E-13
rs146785420	chr12	64646112	A/ATTAT	2.20	2.4E-05	2.45	7.3E-06
rs117543282	chr12	64666533	A/G	-2.97	0.003	-3.38	0.001
rs76982746	chr12	64669405	A/G	-7.12	5.7E-06	-7.41	6.4E-06
rs113825134	chr12	77826960	A/G	-1.16	0.004	-1.26	0.002
rs35124423, rs398098391	chr12	93411631	T/A/T	1.71	2.9E-04	1.50	0.002
rs11107928	chr12	95182175	A/C	2.30	0.001	2.51	7.3E-04
rs6606735	chr12	109051638	T/C	2.71	1.6E-14	2.79	5.2E-14
rs187790190	chr12	111187301	A/G	-4.97	0.007	-5.58	0.004
rs4766453	chr12	111254320	T/C	-1.77	3.0E-06	-1.56	9.3E-05
rs3809272	chr12	111362454	A/G	-5.54	8.0E-50	-5.49	7.0E-45
rs575760658	chr12	111420391	G/A	-6.99	6.3E-15	-6.56	2.9E-12
rs72650673	chr12	111447506	A/G	38.68	2.7E-09	37.53	3.6E-08
rs146378570	chr12	111481102	A/G	-8.75	6.1E-11	-9.63	7.7E-12
rs117532831	chr12	111599646	A/G	-5.72	4.8E-23	-5.82	6.4E-22
rs148019457	chr12	111603681	T/C	-6.24	2.8E-07	-7.02	3.3E-08
rs11065987	chr12	111634620	G/A	5.97	1.9E-61	6.06	7.3E-58
rs147237662	chr12	112118750	T/C	-8.10	0.001	-8.31	0.001
rs116999150	chr12	112479345	G/A	-3.95	2.0E-04	-4.09	2.6E-04
rs35429	chr12	115118062	G/A	-1.56	9.2E-06	-1.32	3.2E-04
rs7310409	chr12	120987058	A/G	-1.03	0.004	-1.10	0.003
rs11553699	chr12	121779004	G/A	-4.91	1.6E-23	-5.41	7.8E-26
rs1270594	chr12	123208418	A/T	-1.90	2.3E-06	-1.69	6.0E-05
rs35817718	chr13	32567803	A/C	1.42	6.9E-05	1.57	2.6E-05
rs61963266	chr13	40596899	A/G	1.69	1.0E-04	1.71	1.6E-04
rs9590569	chr13	41010554	C/T	-1.95	4.6E-07	-1.91	2.1E-06
rs7983902	chr13	46671509	G/C	-1.02	0.004	-1.00	0.006
rs480243	chr13	50847016	G/A	0.86	0.014	0.91	0.013
rs670180	chr13	70662479	T/A	-1.56	8.9E-06	-1.43	1.0E-04
rs4148445	chr13	95245105	C/T	-3.91	2.3E-08	-3.96	5.9E-08
rs4773860	chr13	95248987	C/T	-1.98	1.6E-08	-1.80	8.0E-07
rs11841319	chr13	109840279	T/C	-3.52	3.5E-09	-3.43	3.8E-08
rs336248	chr13	109859213	G/C	1.50	8.2E-04	1.50	0.001
rs544012	chr13	110226292	T/G	-1.58	4.7E-05	-1.57	1.1E-04
rs750598	chr13	110376631	A/G	-0.90	0.014	-1.04	0.006
rs7319994	chr13	113355025	G/I	-2.47	0.001	-2.40	0.002
rs4907618	chr13	113360700	A/G	0.85	0.016	0.71	0.054
rs34723659, rs565104997	chr14	30903115	G/GCT	-0.94	0.018	-0.81	0.051
rs2934701	chr14	50666569	T/A	-0.82	0.017	-0.72	0.045
rs8011233	chr14	53121039	C/A	-1.95	1.7E-05	-2.01	2.2E-05
rs11627485	chr14	65020976	C/T	1.67	1.3E-06	1.69	2.5E-06
rs11158588	chr14	65333158	G/A	-1.86	4.6E-05	-1.95	4.2E-05
rs11622135	chr14	68001175	A/G	-2.94	1.6E-10	-3.18	3.9E-11
rs194730	chr14	68840768	G/A	-1.56	5.7E-05	-1.53	1.6E-04
rs7149929	chr14	68864058	G/A	1.05	0.012	1.23	0.005
rs117672662	chr14	68958750	C/T	-18.52	3.2E-04	-18.43	6.0E-04
rs10220411	chr14	68985371	G/A	1.20	0.003	1.30	0.002
rs116735454	chr14	69087518	C/T	5.86	1.1E-09	5.86	5.2E-09
rs4083463	chr14	81389979	A/G	-1.11	0.002	-1.19	0.002
rs36084521	chr14	93050053	G/T	3.19	1.9E-11	3.33	1.8E-11
rs17580	chr14	94380925	A/T	1.93	0.045	1.91	0.056
rs7148436	chr14	100707574	C/G	-3.12	1.0E-13	-3.26	1.2E-13
rs1555405	chr14	100710432	A/G	-3.45	3.2E-20	-3.85	7.0E-23

rs4906212	chr14	102515258	T/C	-2.08	3.8E-07	-1.83	2.1E-05
rs2146430	chr14	102605488	G/T	-2.78	4.7E-14	-2.68	3.6E-12
rs10138008	chr14	102752567	G/C	-0.96	0.008	-1.02	0.006
rs2297066	chr14	103100498	G/C	3.10	1.0E-13	3.30	3.6E-14
rs744153	chr14	103110107	G/C	-1.52	5.7E-04	-1.73	1.7E-04
rs2497296	chr14	104178439	T/C	1.58	2.7E-05	1.40	3.5E-04
rs7146643	chr14	105291055	C/T	1.93	9.8E-08	1.75	3.6E-06
rs10681907, rs397750208	chr15	38986960	CTTAAA/C	2.01	4.7E-09	1.90	1.2E-07
rs17687755	chr15	41937640	C/G	0.83	0.028	0.94	0.017
rs139974673	chr15	43735687	C/T	5.78	3.7E-08	6.50	3.0E-09
rs68191015	chr15	50096143	G/T	1.09	0.004	1.15	0.004
rs1158246	chr15	56893771	G/A	-2.22	1.3E-07	-2.68	9.7E-10
rs11071720	chr15	63049797	T/C	-3.42	3.8E-20	-3.49	2.8E-19
rs138843544	chr15	63071291	C/T	2.45	0.017	2.23	0.037
rs367966675, rs749807805	chr15	64591430	AAAAC/A	-1.61	0.009	-1.65	0.011
rs141601939	chr15	64804508	C/CT	3.02	0.002	2.96	0.004
rs1684036	chr15	64843126	T/C	-0.86	0.015	-0.87	0.019
rs149678861	chr15	64882295	G/A	-8.34	1.6E-05	-7.38	2.6E-04
rs1719263	chr15	64883993	T/G	3.13	7.4E-11	3.27	6.5E-11
rs12591119	chr15	75063603	G/A	-1.29	0.007	-1.39	0.006
rs8028409	chr15	90963192	T/A	-2.17	6.7E-06	-1.79	3.7E-04
rs4965426	chr15	98704812	A/G	-2.12	6.6E-05	-2.20	7.9E-05
rs140249978, rs866153759	chr16	448906	A/ACT	2.75	1.4E-06	2.99	5.1E-07
rs1298104	chr16	4386444	A/G	1.25	0.033	1.01	0.099
rs9937661	chr16	4970513	C/T	-1.24	3.3E-04	-1.20	8.8E-04
rs34890846, rs59246045	chr16	8947636	ATTG/A	-1.17	0.009	-1.29	0.006
rs2021511	chr16	11251046	T/C	1.97	6.1E-07	1.84	8.9E-06
rs151234	chr16	28494339	C/G	4.33	4.8E-16	4.47	9.5E-16
rs8050500	chr16	31393250	C/T	1.04	0.003	1.23	7.1E-04
rs56088754	chr16	53124307	G/A	0.98	0.011	0.73	0.072
rs3114409	chr16	68698146	C/A	1.19	0.002	1.08	0.008
rs4888387	chr16	75355857	T/G	-0.82	0.019	-0.90	0.014
rs2738502	chr16	78536644	G/C	-0.87	0.019	-0.81	0.037
rs57652769	chr16	79720079	T/C	1.64	2.3E-05	1.65	4.9E-05
rs12445050	chr16	81837364	T/C	-1.52	0.008	-1.19	0.047
rs8056420	chr16	85378035	G/A	-1.01	0.024	-0.96	0.041
rs4783185	chr16	85382020	A/G	1.80	5.9E-04	2.33	2.0E-05
rs9934875	chr16	85398690	G/C	0.74	0.035	0.60	0.102
rs1049868	chr16	85673027	C/T	1.47	1.1E-04	1.42	3.3E-04
rs7200918	chr16	87854513	C/T	1.29	4.6E-04	1.32	5.6E-04
rs17175830	chr16	88491756	A/G	1.58	5.1E-05	1.92	2.3E-06
rs13332145	chr16	89014667	G/A	1.11	0.011	1.04	0.023
rs7225843	chr17	2098531	C/T	-1.31	0.001	-1.35	0.002
rs7213347	chr17	2249963	G/C	2.34	6.2E-10	2.46	4.9E-10
rs1985205	chr17	3912626	T/C	-1.07	0.006	-0.90	0.026
rs141336258	chr17	4921551	C/T	-4.65	0.001	-5.85	9.9E-05
rs56337033	chr17	4932332	T/C	-12.50	0.002	-11.85	0.004
rs2243103	chr17	4936105	G/C	5.22	1.1E-16	5.19	2.5E-15
rs238241	chr17	4947777	A/G	-4.61	3.6E-10	-4.36	1.4E-08
rs141179182	chr17	4947784	A/C	-5.64	0.023	-7.18	0.005
rs150324550	chr17	7364266	AT/A	-1.39	0.002	-1.30	0.005
rs112843870, rs34157498, rs397817394	chr17	7854407	T/TA	1.60	7.8E-06	1.38	2.5E-04
rs9907984	chr17	29399133	G/A	-2.08	2.6E-06	-2.33	5.0E-07
rs8065958	chr17	29465512	T/A	2.42	2.0E-12	2.64	2.0E-13

rs1048317	chr17	31376984	T/C	-1.28	3.4E-04	-1.51	4.7E-05
rs79007502	chr17	35553286	C/T	5.25	2.5E-05	6.10	3.0E-06
rs7503168	chr17	35558885	G/A	3.05	1.9E-10	3.29	5.5E-11
rs201192867	chr17	35572827	C/CCT	4.68	3.6E-05	4.48	1.6E-04
rs16971207	chr17	35614725	T/A	3.79	0.004	2.64	0.052
rs1112173	chr17	37399966	T/C	1.00	0.004	1.13	0.002
rs375677	chr17	44031606	A/C	1.95	8.2E-06	1.79	9.1E-05
rs186330160	chr17	44365997	T/C	-13.07	0.003	-15.00	0.001
rs25552, rs34603233	chr17	44377654	T/TGAGCCC CTG	-1.86	1.5E-07	-1.98	8.2E-08
rs150568286	chr17	44517387	A/G	-15.01	2.3E-10	-15.77	1.7E-10
rs745804299	chr17	44765782	A/G	-26.21	5.5E-20	-25.52	1.4E-17
rs150497606	chr17	57389065	A/G	-3.19	9.2E-05	-3.31	1.0E-04
rs2632516	chr17	58331728	C/G	1.43	3.0E-05	1.66	4.0E-06
rs16943520	chr17	59422422	T/C	-1.30	7.9E-04	-1.17	0.004
rs8178824	chr17	66228657	T/C	5.23	0.002	4.87	0.005
rs77542162	chr17	69085137	G/A	-2.93	0.008	-2.85	0.013
rs2034309	chr17	74691926	C/T	-1.38	5.2E-04	-1.42	5.9E-04
rs10541233	chr18	9617853	CTG/C	-1.17	0.006	-1.31	0.003
rs11082304	chr18	23141009	G/T	3.16	4.1E-20	3.10	6.9E-18
rs74997723	chr18	44416285	G/C	2.90	7.3E-09	2.77	1.2E-07
rs4890487	chr18	44650207	A/C	-1.38	8.0E-05	-1.32	3.3E-04
rs718515	chr18	46276334	G/A	0.74	0.030	0.96	0.007
rs72969820	chr18	59677958	T/C	-1.46	0.005	-1.52	0.005
rs17758695	chr18	63253621	T/C	-6.08	1.1E-11	-5.64	1.5E-09
rs1865761	chr18	69872156	C/T	-0.68	0.049	-0.93	0.010
rs113205391	chr18	69950261	G/C	-5.19	2.9E-04	-5.99	6.7E-05
rs142316985, rs367640262	chr18	75196910	C/CAATT	-2.46	0.002	-2.27	0.006
rs12985107	chr19	1163597	G/A	1.03	0.009	1.06	0.010
rs8100043	chr19	2005645	G/A	1.72	2.2E-04	1.87	1.2E-04
rs8106212	chr19	6802560	T/C	-11.26	0.001	-11.44	0.002
rs34536443	chr19	10352442	C/G	-3.78	7.2E-06	-3.70	2.7E-05
rs113906245	chr19	10473649	T/C	-3.69	0.020	-4.06	0.014
rs34855805	chr19	15272771	A/C	1.33	0.002	1.58	5.8E-04
rs11086023	chr19	16089568	T/C	2.53	3.5E-11	2.60	7.4E-11
rs57843631	chr19	16095202	T/C	-14.61	8.2E-30	-14.64	1.2E-27
rs34353978, rs397859699	chr19	17053576	A/AG	-0.98	0.022	-1.11	0.013
rs59922977	chr19	17141573	TC/T	1.04	0.003	0.94	0.010
rs6512220	chr19	17732986	C/T	1.06	0.002	0.92	0.011
rs188247550	chr19	19285807	T/C	-5.76	0.009	-6.27	0.006
rs7249692	chr19	19559879	T/C	1.50	5.2E-05	1.42	2.4E-04
rs3841260	chr19	19645264	A/AGCC	3.37	1.9E-06	3.47	2.7E-06
rs45522544	chr19	19654690	T/C	10.12	2.8E-05	10.99	1.4E-05
rs117137505	chr19	20179339	T/G	9.01	0.018	9.51	0.015
rs10402931	chr19	32581163	G/A	-1.07	0.003	-1.16	0.003
rs12975577	chr19	33264458	T/C	-1.02	0.003	-1.01	0.005
rs897764	chr19	35068720	T/C	-2.12	7.8E-04	-1.97	0.003
rs6510469	chr19	35181105	T/G	-1.12	0.002	-1.11	0.003
rs2733737	chr19	35554501	T/C	0.72	0.046	0.90	0.017
rs11668070	chr19	38258273	G/A	-0.80	0.020	-0.90	0.013
rs12983010	chr19	38738449	G/A	3.31	8.5E-07	3.19	5.4E-06
rs12721051	chr19	44918903	G/C	-1.46	7.4E-04	-1.25	0.006
rs3803906	chr19	45212718	G/A	-1.66	1.3E-05	-1.93	1.2E-06
rs73036520	chr19	45246226	C/G	-3.67	2.3E-19	-3.60	3.2E-17
rs1005165	chr19	45405792	T/C	-0.95	0.022	-0.97	0.025

rs59310453, rs796764477	chr19	45812099	AT/A	0.80	0.019	0.80	0.026
rs3865444	chr19	51224706	A/C	-1.58	1.2E-05	-1.79	2.0E-06
rs626283	chr19	54173307	C/G	1.46	2.8E-05	1.68	4.4E-06
rs892090	chr19	55027704	T/G	1.03	0.027	1.21	0.013
rs45541434	chr19	55182101	T/C	-8.12	4.5E-10	-7.56	3.2E-08
rs147881000	chr19	55186344	A/G	-11.08	0.009	-12.85	0.003
rs34548043	chr19	58420528	TA/T	-1.22	4.2E-04	-1.15	0.001
rs190391173	chr20	1488179	C/T	-53.10	1.3E-55	-52.54	9.8E-50
rs11906768	chr20	1943420	C/T	-3.93	1.8E-24	-3.93	1.5E-22
rs6055955	chr20	8623534	C/T	1.23	3.4E-04	1.07	0.003
rs80054178	chr20	31706879	C/T	8.43	1.3E-16	9.11	1.0E-17
rs6060986	chr20	31838494	A/G	-1.61	1.5E-05	-1.84	2.3E-06
rs4812447	chr20	40643980	G/A	0.77	0.026	0.75	0.038
rs6103669	chr20	44188205	A/G	-0.95	0.012	-1.27	0.001
rs7265567	chr20	50471092	T/C	1.05	0.005	1.17	0.003
rs55708816	chr20	56415068	A/G	1.56	0.006	1.46	0.015
rs463312	chr20	59022915	C/A	-5.22	4.4E-12	-5.75	2.6E-13
rs11471957	chr20	59023277	CAA/C	2.33	1.3E-08	2.59	1.7E-09
rs4812056	chr20	59098895	C/A	1.63	5.5E-05	1.81	2.0E-05
rs4809330	chr20	63718234	A/G	-0.79	0.027	-0.55	0.140
rs2142218	chr21	15059123	C/T	-0.99	0.028	-0.76	0.105
rs2070513	chr21	33981573	T/G	0.72	0.049	0.79	0.039
rs111527738	chr21	34887027	G/A	-3.51	0.018	-3.40	0.028
rs2242892	chr21	35023677	A/G	-2.95	4.2E-06	-3.21	1.7E-06
rs75967349	chr21	35028144	G/C	-2.70	0.001	-2.96	6.5E-04
rs7280028	chr21	35044543	C/T	-0.91	0.032	-0.87	0.050
rs17285189	chr21	38474017	T/G	2.04	4.7E-05	1.95	1.9E-04
rs9974653	chr21	39369920	C/A	-1.06	0.005	-1.16	0.003
rs183393610	chr22	17769676	A/G	3.78	0.030	3.93	0.031
rs1059196	chr22	19724571	T/C	-1.67	3.0E-06	-1.67	7.7E-06
rs2238784	chr22	19985036	A/G	1.67	5.5E-06	1.63	1.9E-05
rs45462093	chr22	27791408	G/A	1.58	1.5E-04	1.51	5.0E-04
rs5763928	chr22	30282541	G/A	-1.90	3.0E-06	-2.03	1.7E-06
rs855791	chr22	37066896	A/G	1.33	1.4E-04	1.45	7.6E-05
rs2013837	chr22	38875612	G/A	0.97	0.005	1.16	0.001
rs5758880	chr22	42704850	T/C	0.97	0.009	0.86	0.026
rs1883314	chr22	42978801	G/C	1.81	1.6E-07	1.87	2.1E-07
rs738409	chr22	43928847	G/C	-1.68	3.6E-05	-1.67	8.7E-05
rs61431587	chr22	49766569	G/T	1.26	0.050	1.33	0.047
rs75107793	chr22	50190508	A/G	7.02	4.3E-33	6.73	5.0E-28
rs73187269	chr22	50224992	T/C	-2.59	6.4E-08	-2.66	1.2E-07
rs8138438	chr22	50642329	G/A	0.98	0.005	0.99	0.007

Plot showing effects of the 577 PLT variants expressed as platelet count (PLT, in $N \times 10^3/\mu l$) with 95% confidence intervals, in data including both genotyped and familially imputed Icelanders (x-axis) vs. only genotyped Icelanders (y-axis).

4. The author use the meta-analysis results to calculate the PLT PRS- additional details are warranted here. How good is the PLT PRS? AUC? What is the correlation between PLT and PLT PRS?

A: We did not use the meta-analysis results to calculate the PLT-PRS. To avoid overfitting due to sample overlap, we used the PRS GWAS for Iceland to create the PLT-PRS in the UKB dataset, and the PRS GWAS for UKB to calculate the PLT-PRS for the Icelandic dataset (described in the **Methods** section). The resulting PLT-PRSs explain 11.9% and 10.9% of the variance in the Icelandic and UKB datasets respectively.

The variance explained for the different models (fraction of causal variants) that we tested in LDpred is presented in the table below and has been added as **Supplementary Table 24**.

Fraction of variance of PLT measurements explained by PLT-PRS scores created based on different re-weighting models in LDpred.

Model^a	UKB	Iceland
Unweighted	4.4%	7.5%
1	6.0%	9.2%
0.3	7.1%	7.6%
0.1	9.0%	6.1%
0.03	10.9%	5.5%
0.01	3.2%	5.4%
0.003	1.5%	3.9%
0.001	1.5%	4.2%
Infinitesimal	7.5%	11.9%

^aAssumed fraction of causal variants in the LDpred model used.

5. The authors identified 24 traits that are correlated with PLT PRS, how does this compare with the phenotypic correlation with the original measured PLT?

A: The table below presents results of the phenotype correlation between PLT measurements and the 24 traits in the UKB dataset, for which we have more complete and unbiased phenotype information and which is larger and hence better powered. For comparison we have repeated the phenotype correlation test using PLT measurement adjusted for the PLT-PRS to remove some of the genetic factor (See Major Comment 7).

The phenotype correlation between the 24 traits and PLT measurements on the one hand and PLT measurements adjusted for the PLT PRS on the other hand, in the UKB dataset.

Trait	Platelet measurements UK			Platelet measurements adjusted for PLT-PRS UK		
	Beta	P	R2	Beta	P	R2
Ankylosing spondylitis	0.0047	1.3E-27	0.009	0.0042	3.4E-19	0.006
Hypertension	0.0012	2.8E-65	0.001	0.0012	6.0E-61	0.001
RA	0.0034	1.6E-51	0.005	0.0033	9.6E-44	0.004
MPN	0.0125	1.4E-214	0.147	0.0127	2.7E-212	0.145
BPH	-0.0004	0.0003	0.000	-0.0007	2.6E-08	0.000
MPV	-0.0082	4.3e-25333	0.227	-0.0080	2.3e-20449	0.191
WBC	0.0048	1.5e-7301	0.078	0.0048	1.3e-6606	0.071
Lymphocyte count	0.0029	9.0e-2517	0.028	0.0028	2.36e-2133	0.024
Basophil count	0.0018	1.4e-1096	0.011	0.0019	2.4e-1036	0.010
Eosinophil count	0.0019	6.2e-1157	0.013	0.0018	3.6e-885	0.010
Monocyte count	0.0027	7.7e-2231	0.025	0.0026	2.6e-1881	0.021
Neutrophil count	0.0043	1.5e-5695	0.062	0.0044	4.3e-5343	0.058
Triglycerides	0.0015	3.9e-632	0.008	0.0015	1.2e-600	0.007
Total cholesterol	0.0015	2.8e-662	0.008	0.0016	1.5e-672	0.008
Non-HDL cholesterol	0.0017	1.8e-764	0.010	0.0018	3.6e-781	0.010
Heart rate	0.0023	6.5e-1522	0.018	0.0025	9.7e-1579	0.019
Mean arterial pressure	0.0011	1.9e-347	0.004	0.0012	1.7e-340	0.004
GGTP	0.0013	7.1e-450	0.005	0.0012	2.5e-377	0.005
AP	0.0017	1.1e-794	0.010	0.0017	3.8e-779	0.009
Bilirubin	-0.0028	9.3e-2290	0.027	-0.0031	4.4e-2421	0.028
CRP	0.0027	2.1e-2035	0.024	0.0028	1.1e-2093	0.025
Creatinine	-0.0009	2.3E-236	0.003	-0.0009	1.9E-221	0.003
Height	-0.0016	3.3e-776	0.009	-0.0016	3.1e-662	0.008
Weight	0.0000	0.38	0.000	0.0001	0.0011	0.000

6. Given the likely high correlation of phenotypes, this portion of the study would be better powered if significance was achieved through permutation, rather than using bonferonni for multiple test corrections.

A: As there is extensive correlation both between the genetic variants and the study individuals that are related, in particular in the Icelandic dataset but also to some extent in the UK Biobank dataset, permutation test would not be valid.

7. The authors also ran the single variant association test for the 577 PLT SNPs on the identified 24 traits. It would be interesting to also run the regression adjusting the original PLT measures in order to see the genetic effect separately from the phenotypic correlation.

A: See Major Comment 5 and the table in our response to that comment where we present phenotype correlation of PLT measurements with the 24 traits in the UKB dataset and a separate result for the PLT measurements adjusted for the PLT-PRS . We caution though

that the PLT-PRS only captures a part of the genetic factor (in the UKB dataset it explains 10.9% of the PLT variance), hence these results will not prove a good estimate of the phenotypic correlation in the absence of genetic factors.

8. Although authors state there is not significant heterogeneity in Icelander and UK Biobank, some p values are large in Icelanders but small in UK Biobank (UKB). For example, rs964184 (supplementary table 2) has p value of 1.9E-3 in Icelander and 1.8E-25 in UKB. Are such significant meta-analysis results driven by UKB data?

A: The effective sample size of the UKB PLT dataset is much larger than for the Icelandic PLT dataset as can be seen from the association results for the two dataset. Although the Icelandic dataset includes 270,211 individuals, about half of them are not genotyped directly and they do not contribute much to the statistical power (as is shown in response to Major Comment 3 above). And as the genotyped Icelandic individuals are very related, 139,479 individuals correspond to about 1/3 of the Icelandic population, and the effective sample size is much less than the number of genotyped individuals would indicate. This is reflected in a large genotypic inflation factor for the Icelandic genotyped individuals (PLT- λ_{gc} = 2.65). In contrast the UKB PLT dataset includes 397,495 mostly unrelated individuals (for UKB PLT- λ_{gc} = 1.51). So it is fair to say that association results are mostly driven by the UKB dataset. However, even though the P values are much less significant for the Icelandic dataset, the effect estimates between the Icelandic and UKB results are consistent as can be seen the heterogeneity test presented in Supplementary table 2 in the P_{het} column.

9. The authors should provide figures demonstrating that their test statistics are behaving appropriately given the complex relatedness structures and genotype imputation procedures which may introduce bias- the correction factor for case-control phenotypes which were found to correlate with the PLT PRS are given, but it would be useful to have more information about any inflation seen in the primary GWAS/Meta analyses.

A: The plots below show the excess of real signals with large χ^2 values in the datasets, especially in the UKB dataset. But for lower χ^2 values the statistics look well calibrated although slightly conservative for the Icelandic dataset. We have added the estimated correction factors for the QT phenotypes to **Supplementary Table 23**. The qq-plots are now included as **Supplementary Figure 3**. The corresponding clarifications have now been added to the text (lines 450-452) and sound as following: The estimated correction factors for the phenotypes, which were found to correlate with the PLT PRS, are shown in **Supplementary Table 23**. The Q-Q plots are presented in **Supplementary Figure 3**.

Q-Q plots of the genome-wide association results. A) A Q-Q plot of the adjusted Chi²-statistics for association with PLT values for all tested variants for the UK Biobank (red dots) and the Icelandic (blue dots) datasets, respectively. The equiangular line (green line) is included in the plot for reference purpose. B) A subset of Figure A) showing the Q-Q plot for variants with low Chi² values.

A)

B)

Minor comments:

1. Filtering criteria of imputation information over 0.8 and 0.7 were applied to Iceland and UK Biobank dataset respectively. What is the effect of these differing thresholds on low MAF (< 1~2%) in Iceland data?

A: There are about 0.66 million variants in the Icelandic data with imputation info in the range 0.7 to 0.8, and with minor allele frequency of 1-2%.

2. The 4 novel PLT loci, should be better highlighted and specifically described. Did any of these show compelling associations with related traits?

A: Please refer to our response to Comment 3 by Reviewer 2.

3. IPA method is a bit of a black box (disclosure- I'm not a fan of this method, but am trying to set aside my bias here) and provides a network association rather than real causality. A formal causality test would be appropriate and potentially informative for the highlighted networks of interest.

A: It is not unusual to be skeptical towards IPA, but upon close acquaintance with its methodology and available features it becomes much less of a black box and can be very helpful in providing biological insights into data (especially into expression data). Causality is not claimed, instead it helps to gain insight into how differential expression of genes in our dataset can potentially lead to outcomes, for which we identified associations in other parts of our study. Since we aimed at trying to understand biological mechanisms and pathways through which the identified PLT loci could affect platelet count, it was a reasonable approach, complementary to other parts of the study and turned out to be supportive of our findings from e.g. the PLT PRS association part.

REVIEWERS' COMMENTS:

Reviewer #2 (Remarks to the Author):

The authors have provided a thorough and detailed response and I believe they adequately met the review comments provided.

I also complement the authors on making their PRS available, and the results available on their DeCode site to enable future scientific endeavors.

Reviewer #3 (Remarks to the Author):

The authors should be commended on a highly responsive revision.

I have only two minor points:

1. The concern regarding variant quality filtering is why filter two datasets using different imputation qualities (0.8 for Iceland and 0.7 for UKB)? This approach could lead to loss of variants with low MAF in Icelandic dataset, since often less frequent variants have lower imputation quality.
2. The sensitivity analysis plot on p.32 is great and it should be included in supplementary materials.